# Engineering twin structures and substitutional dopants in ZnSe$_{0.7}$Te$_{0.3}$ anode material for enhanced sodium storage performance

Jingui Zong[1], Yazhan Liang[1], Fan Liu[1], Mingzhe Zhang[1,2], Kepeng Song [1]✉, Jinkui Feng [2], Baojuan Xi[1]✉ & Shenglin Xiong[1]✉

Compared with lithium-ion batteries (LIBs), sodium-ion batteries (SIBs) are an alternative technology for future energy storage due to their abundant resources and economic benefits. Constructing various defects is considered to be a common viable means of improving the performance of sodium storage. However, it is of significance to thoroughly scrutinize the formation mechanism of defects and their effects and transition during the charge–discharge process. Here, twin structures are introduced into ZnSe$_{0.7}$Te$_{0.3}$ nanocrystals by doping of Te heteroatoms. The Te dopants are visualized to locate in the lattices of ZnSe by spherical aberration electron microscopy. The formation of twin structures is thermodynamically promoted by Te heteroatoms partially replacing Se based on the theoretical calculation results. Moreover, calculation results show that with the increase of twin boundaries (TBs), the sodium diffusion energy barrier is greatly reduced, which helps the kinetics of sodium ion diffusion. In the connection, the composition and amount of TBs are optimized via tuning the doping level. The combined effect of point defects and twin structures greatly improves the sodium storage performance of ZnSe$_{0.7}$Te$_{0.3}$@C. Our work reveals the mechanism of the point defect on the twin plane defect and systematically investigates their effect on the electrochemical performance.

In recent years, sodium-ion batteries (SIBs) have made great progress, and are expected to be one of the substitutes for widely commercialized lithium-ion batteries (LIBs), due to the abundant reserves of sodium resources and similar electrochemistry and operation mechanisms[1,2]. As a typical anode material for LIBs, graphite has been widely used. In view of the limited battery capacity and structural instability of sodium-graphite intercalation compounds, it's urgent to search for high-capacity and long-life SIB anode materials to meet the needs of large-scale energy storage applications[3,4]. Thanks to their structural diversity and high theoretical capacity (about 400–600 mAh g$^{-1}$), transition metal chalcogenides (TMC) have great application potential in the anode of SIBs[5]. ZnSe, an important member of TMC, is an alloy–conversion combined anode material prospective for SIBs. It has decent electrical conductivity, better than its oxide counterpart, and a weak metal-selenium (Se) bond that facilitates the electrochemical conversion reactions[6]. But, it has some problems such as poorer electronic conductivity than carbon, sluggish kinetics, and large volume change

[1]School of Chemistry and Chemical Engineering, Shandong University, Jinan, China. [2]School of Materials Science and Engineering, Shandong University, Jinan, China. ✉e-mail: kpsong@sdu.edu.cn; baojuanxi@sdu.edu.cn; chexsl@sdu.edu.cn

in the long-term charge-discharge process, which limits its rapid charge-discharge ability and structural stability. To solve these problems, various strategies have been developed, such as combining ZnSe with carbon-based materials and constructing robust three-dimensional nanostructures. Moreover, defect engineering is demonstrated to be an effective method to reasonably design electrode materials for rechargeable batteries to achieve the improved electrochemical performance[7].

Defects in crystals can be classified into point defects, line defects, and planar defects according to the dimension of the defects. They have significant effects on the chemical properties, thermal stability, and mechanical properties of materials[8,9]. Point defects such as vacancies, substitute and interstitial atoms can increase adsorption sites, accelerate the ion diffusion, and improve the electronic conductivity of the LIB and SIB electrode materials[10], such as Co and F codoped $SnO_2$[11], $MoS_2$/C with S vacancies[12], Cu-doping cobalt embedded nitrogen-doped porous carbon (CoCu@NC)[13]. Dislocations, one kind of line defects, can prevent cracking, loss of active materials, and adverse interface reactions with electrolytes by reducing strain during the phase transition of spinel material $LiNi_{0.5}Mn_{1.5}O_4$[14]. In addition, the diffusion rate of $Na^+$ at grain boundaries (planar defects) is much faster than in the bulk phase of $Ta^{5+}$-substituted $Na_3V_2(PO_4)_3$[15]. As a member of special planar defects, twin boundaries (TBs) also often appear in crystal materials with a twin structure. The existence of twin structure is conducive to the diffusion of lithium ions in electrode materials, which helps to improve the electrochemical dynamics of batteries[8,16]. For instance, Nie et al. demonstrated that TBs promote the diffusion of lithium ions in single-crystal $SnO_2$ nanowires[17]. Wang et al. studied the formation of TBs in lithium manganate oxide, and also demonstrated TBs can enable fast lithium-ion diffusion and charging performance[18]. It follows that the defect investigations have been done thoroughly, especially in LIBs; however, more efforts should be made on the study of anode materials for SIBs. The electrical conductivity of Te is about $2 \times 10^2\,S\,cm^{-1}$, which is much higher than that of Se ($1 \times 10^{-4}\,S\,cm^{-1}$)[19,20]. In addition, Se and Te atoms are in the same main group, and the latter has a slightly larger radius than the former. This indicates that Te heteroatom doping of ZnSe may improve its electrical conductivity and potentially introduce some additional defects. The introduction of two kinds of defects into ZnSe can collectively increase adsorption sites and promote reaction kinetics, contributing to better electrochemical sodium storage. At the same time, the systematic investigations of the formation, effects, and transition of defects during electrochemistry also require further efforts.

Herein, we prepared $ZnSe_{0.7}Te_{0.3}$ nanocrystals with twin structure as anode material for SIBs using zeolitic imidazolate frameworks (ZIF-8) as template and Te heteroatom doping hybridized with thin hollow carbon structure ($ZnSe_{0.7}Te_{0.3}$@C). The doping of heterogeneous Te increased the energy of the system and lattice distortion. Alternatively, the crystal matrix introduced TB defects to alleviate this tendency and maintain the system stable. Moreover, via the composition adjustment during the synthesis, $ZnSe_{0.7}Te_{0.3}$ was determined as the optimized Te doping level with optimal TB amount. By a combination of a series of structural characterizations, the electrochemical reactions of $ZnSe_{0.7}Te_{0.3}$ with sodium ions were confirmed, which also demonstrated the transition of defects in $ZnSe_{0.7}Te_{0.3}$ during the charging/discharging. $ZnSe_{0.7}Te_{0.3}$@C electrode shows significantly superior sodium storage properties to the pristine ZnSe@C electrode, including a higher capacity ($5\,A\,g^{-1}$; $307\,mAh\,g^{-1}$ vs $118.8\,mAh\,g^{-1}$ after 1000 cycles), better rate performance ($20\,A\,g^{-1}$; $256.2$ vs $121.5\,mAh\,g^{-1}$). The good storage performance results from the promotive effect of two defect dimensions, TB (planar defect) and substitution dopant (point defect), in $ZnSe_{0.7}Te_{0.3}$. When designing the anode materials of SIBs, the defects of two dimensions are introduced at the same time, which will overcome their shortcomings from different aspects, and finally realize the comprehensive and significant improvement of sodium storage performance.

## Results and discussion
### Materials synthesis and characterization
The preparation flow chart of $ZnSe_{0.7}Te_{0.3}$@C nanocomposite with twin structure is shown in Fig. 1a. ZIF-8 is composed of zinc ions and dimethylimidazole molecules and serves as the precursor. From scanning electron microscope (SEM) and transmission electron microscopy (TEM) images in Supplementary Fig. 1, the as-prepared ZIF-8 exhibits solid dodecahedra with a side length of about 1 μm. Due to the presence of the terminal N-H functional groups, ZIF-8 is always sensitive to the mixed solution of ethanol and water[21]. In our synthesis, ZIF-8 dodecahedra grew into smaller solid nanospheres under the action of hydroxyl groups in M-aminophenol and ammonium hydroxide. At the same time, M-aminophenol and formaldehyde polymerized to form MF resin encapsulating ZIF-8 nanospheres (ZIF-8@MF), as shown in SEM and TEM images in Supplementary Fig. 2. Then, ZIF-8@MF was fully ground and mixed with Se and Te powder, followed by annealing at high temperature under the protection of $Ar/H_2$ atmosphere and $ZnSe_{0.7}Te_{0.3}$@C nanocomposite was harvested. In order to detect the role of defects, the nanocomposite of ZnSe@C was also fabricated for comparison. The morphology and structure of ZnSe@C and $ZnSe_{0.7}Te_{0.3}$@C were characterized by SEM and TEM. From images of Fig. 1b, c, $ZnSe_{0.7}Te_{0.3}$@C nanocomposite consists of typical nanobowl-like carbon with a diameter of about 60 nm and $ZnSe_{0.7}Te_{0.3}$ nanocrystals. Then, aberration-corrected high-angle annular dark-field-scanning TEM (HAADF-STEM) image (Fig. 1d) shows the low contrast of the carbon structure, demonstrating the thin shell of a few nanometers. The energy-dispersive X-ray spectrum (EDS) (Fig. 1e) mapping shows that $ZnSe_{0.7}Te_{0.3}$@C includes C, N, Zn, Se, and Te elements, and the nanoparticles are composed of Se, Te, and Zn elements, indicating the formation of $ZnSe_{0.7}Te_{0.3}$. ZnSe@C exhibits similar morphological features, composed of nanobowl-like carbon and ZnSe nanocrystals (Supplementary Fig. 3).

In order to further detect the crystallographic structure of $ZnSe_{0.7}Te_{0.3}$, high-resolution imaging was done by the HAADF-STEM technique. From the Supplementary Fig. 4, a remarkable twin structure and multiple TBs in $ZnSe_{0.7}Te_{0.3}$ nanoparticle are clearly observed. Figure 2a shows the partial HAADF-STEM image of Supplementary Fig. 4. The twin plane is analyzed to be (111) plane with a spacing of 0.34 nm (Fig. 2a), and the twinning direction can be determined to be [11-2]. In contrast, no twin structure is present in ZnSe@C, as shown in Supplementary Fig. 5. Figure 2b-d shows the corresponding atom mapping of Zn, Se, and Te. The lattices of Zn and Se are relatively complete, while Te atoms are randomly dispersed and the distribution doesn't accord with the periodic lattice structure, indicating the doped Te atoms. The structure of mixed ZnSe atom mapping is shown in Fig. 2e. Figure 2f-h shows that Te heteroatoms are successfully doped into the ZnSe lattices. Furthermore, it can be seen that partial Se atoms are missing and replaced by Te atoms marked by the white circle. The absence of Se atoms is demonstrated by the white circle marked in the combined atom mapping of Se and Zn in Fig. 2e. The white circle marked in the complex atom mapping of Zn and Te in Fig. 2f shows the existence of Zn and Te atom, indicating that Te atom does not replace the Zn atom. In summary, it can be concluded that doped Te atoms partially replace Se atoms. According to the above analysis, the atom model schematic illustration of $ZnSe_{0.7}Te_{0.3}$ twin structure was described in Fig. 2i.

In order to explain the formation mechanism of the twin structure in $ZnSe_{0.7}Te_{0.3}$@C, we carried out the density functional theory (DFT)[22,23] calculations to detect the underlying impetus. According to the above spherical aberration-corrected electron microscopy results, no obvious vacancy defect was found in $ZnSe_{0.7}Te_{0.3}$@C, and the

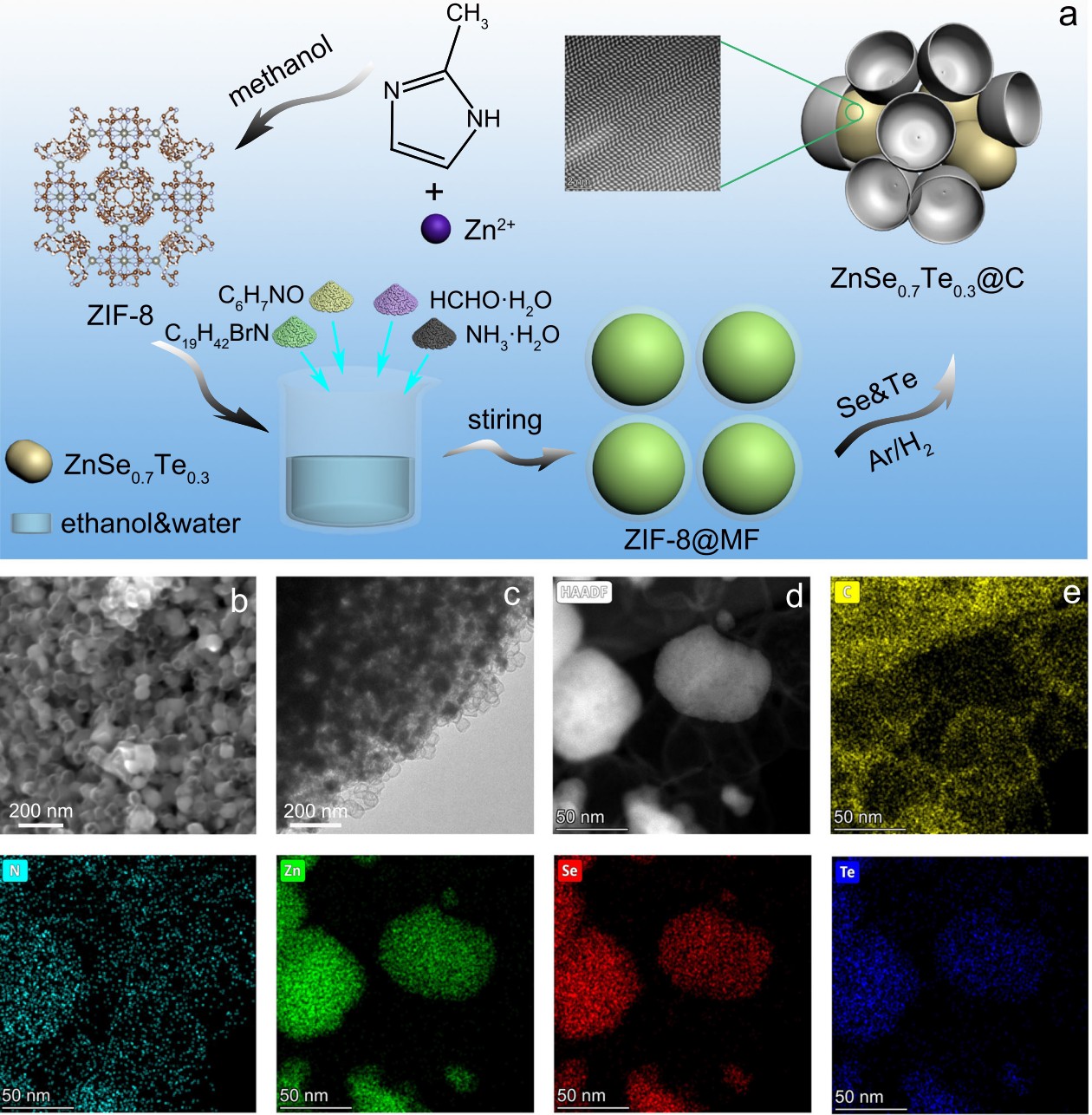

**Fig. 1 | Morphology and structural characterization of ZnSe$_{0.7}$Te$_{0.3}$@C. a** Schematic illustration of the preparation process of ZnSe$_{0.7}$Te$_{0.3}$@C. **b** FESEM and **c** TEM images of ZnSe$_{0.7}$Te$_{0.3}$@C. **d** and **e** STEM−EDX elemental mapping of ZnSe$_{0.7}$Te$_{0.3}$@C: C (yellow), N (azure), Te (blue), Se (red), and Zn (green).

arrangement of atoms was highly ordered and complete. Therefore, from the perspective of theory, Te can either replace Se atoms or occupy different interstitial sites in ZnSe, considering Te and Se are in the same group[24]. In order to further verify theoretically whether Te replaces Se atoms or occupies interstitial sites in ZnSe, we constructed the structural models of different configurations in Fig. 3a–c and then calculated the corresponding phonon spectra, average energy per atom in the final state, and defect formation energy. From Supplementary Fig. 6a, b, it can be seen that the phonon spectra of the sub 1 model with one Te atom replacing one Se atom and the sub 3 model with three Te atoms replacing three Se atoms have no virtual phonon mode, indicating that the structures are dynamically stable. However, the phonon spectrum for the int model in which Te occupies an interstitial position has a virtual phonon mode, proving that its structure is unstable, as shown in Supplementary Fig. 6c–e. The virtual

frequency may be caused by the large Te atom occupying the interstitial position, rendering a huge structural distortion and profoundly affecting the normal arrangement of the surrounding atoms. The average energy of each atom in the final state of these different configurations is negative, which indicates that they are thermodynamically stable (Supplementary Fig. 6f). Furthermore, the elastic constants of sub 1, sub 3 and int models have been calculated, and the results indicate that these structural models are mechanically stable (The calculations and result details are in Note 1 in Supplementary information). The defect formation energy of sub 1 model is 0.76 eV, which is much lower than that of sub 3 and int models (2.23 and 3.31 eV), as shown in Fig. 3f. This indicates that Te tends to replace Se atoms rather than occupy interstitial positions, which is consistent with the above atom mapping results. In addition, considering the application of the system for SIB, the ab initio molecular dynamics

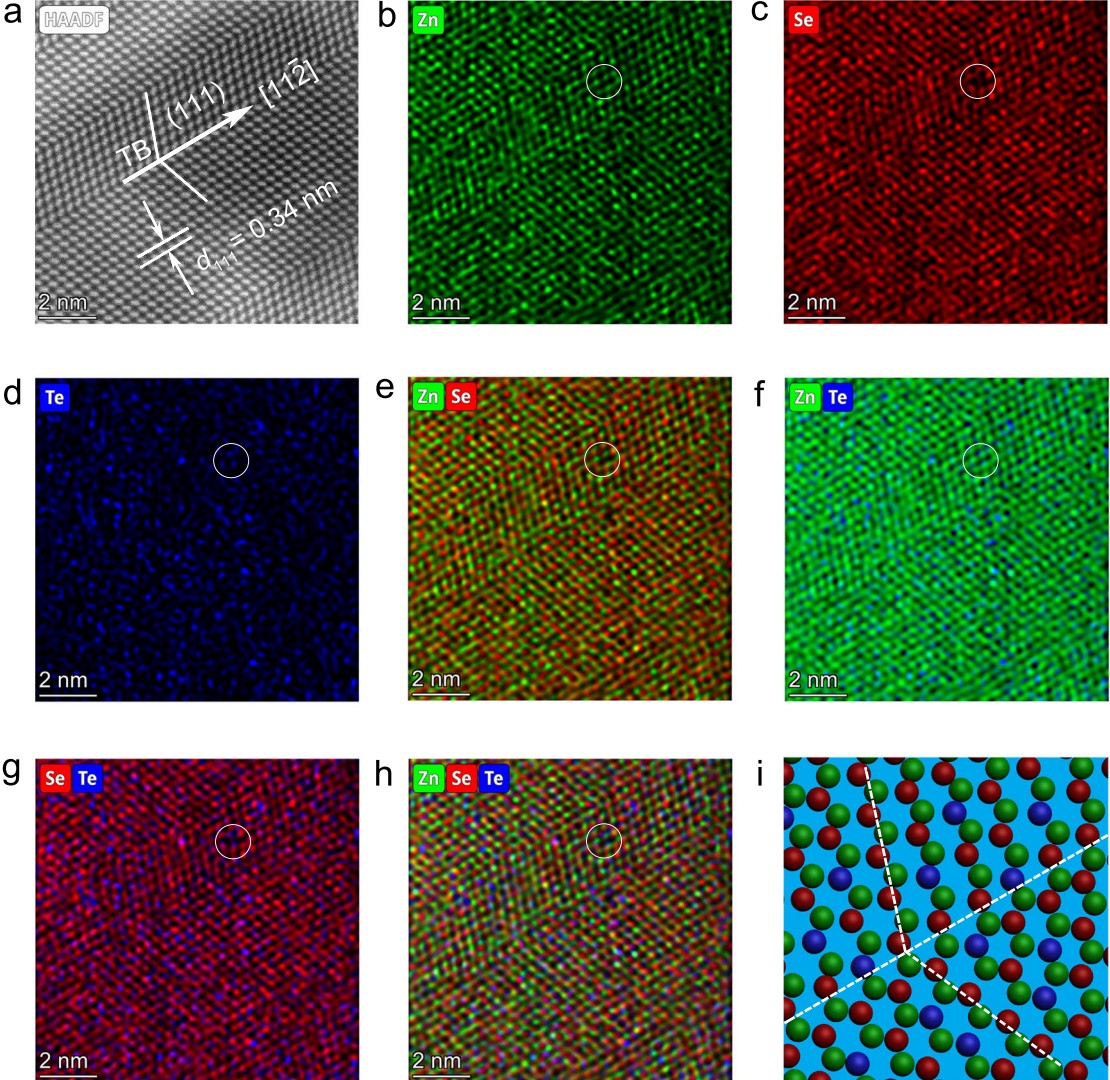

**Fig. 2 | Characterization of twin boundaries in the ZnSe$_{0.7}$Te$_{0.3}$@C. a–h** Atomically resolved HAADF−STEM image (**a**) and the corresponding EDX maps of **b** Zn, **c** Se, **d** Te, **e** Zn and Se merging, **f** Zn and Te merging, **g** Se and Te merging, and **h** Zn, Se, and Te merging. **i** Atomic model schematic illustration of the twin ZnSe$_{0.7}$Te$_{0.3}$ (Zn: green, Se: red, and Te: blue).

simulations were used to detect the behaviors of sub 1 and sub 3 at different temperatures. As shown in Supplementary Fig. 7, the results show that, at 298 and 313 K, the sub 1 and sub 3 structures do not change significantly with the increase of temperature. Figure 3d, e shows the atomic interface structure models of ZnSe$_{0.7}$Te$_{0.3}$ with TBs and ZnSe$_{0.7}$Te$_{0.3}$ without TBs based on the HAADF−STEM imaging results. The total number of atoms of ZnSe$_{0.7}$Te$_{0.3}$ models with TBs and without TBs is the same, and the ratio of Se to Te atoms is about 7:3. The average energy per atom in the final state of the ZnSe$_{0.7}$Te$_{0.3}$ with TBs is −−2.99 eV, which is lower than that of ZnSe$_{0.7}$Te$_{0.3}$ without TBs (−2.8 eV), indicating that ZnSe$_{0.7}$Te$_{0.3}$ with twin structure is more stable thermodynamically, as shown in Fig. 3g. In addition, the defect formation energy of ZnSe$_{0.7}$Te$_{0.3}$ with TBs is relatively small (0.86 eV), indicating that the twin structure is easy to form. Based on the above analyses, it's known that the introduction of Te substitute atoms will increase the energy of the ZnSe system. But, the formation of twin structure can, to some extent, stabilize the Te-doped ZnSe$_{0.7}$Te$_{0.3}$ system. So, the Te substitutional atoms thermodynamically promote the formation of twin structures in ZnSe.

The crystal phase of the samples was characterized by X-ray diffraction (XRD) measurements, as shown in Supplementary Fig. 8. Both

ZnSe@C and ZnSe$_{0.7}$Te$_{0.3}$@C show a set of diffraction peaks notably consistent with the face-centered cubic ZnSe with space group F-43m. No impurity peaks were detected in either sample, and sharp Bragg peaks indicated good crystallinity in both ZnSe@C and ZnSe$_{0.7}$Te$_{0.3}$@C. Accurate structural information of ZnSe@C and ZnSe$_{0.7}$Te$_{0.3}$@C is obtained through Rietveld refinement, and the results are listed in Supplementary Tables 1–3. It can be seen from the refinement results that the cell parameters and cell volume of ZnSe$_{0.7}$Te$_{0.3}$@C are larger than those of ZnSe@C, indicating that Te atoms are successfully doped into ZnSe (Supplementary Table 3). The occupancy of Se and Te atoms in ZnSe$_{0.7}$Te$_{0.3}$@C was 0.029 and 0.013, respectively, which was consistent with the atomic percentage of Se and Te in XPS results (Supplementary Table 2). In addition, compared with ZnSe@C, the diffraction peaks of ZnSe$_{0.7}$Te$_{0.3}$@C have different degrees of deviation toward small angles, which was caused by Te heteroatoms successfully doped into ZnSe crystal lattices. The average crystallite size of ZnSe@C and ZnSe$_{0.7}$Te$_{0.3}$@C is calculated using Scherrer's method to be 43.4 nm and 17.8 nm, respectively, which proves that the nanoparticle size of ZnSe$_{0.7}$Te$_{0.3}$@C is smaller (Supplementary Table 4). The valence state and chemical composition of ZnSe$_{0.7}$Te$_{0.3}$@C nanocomposites were further studied by X-ray

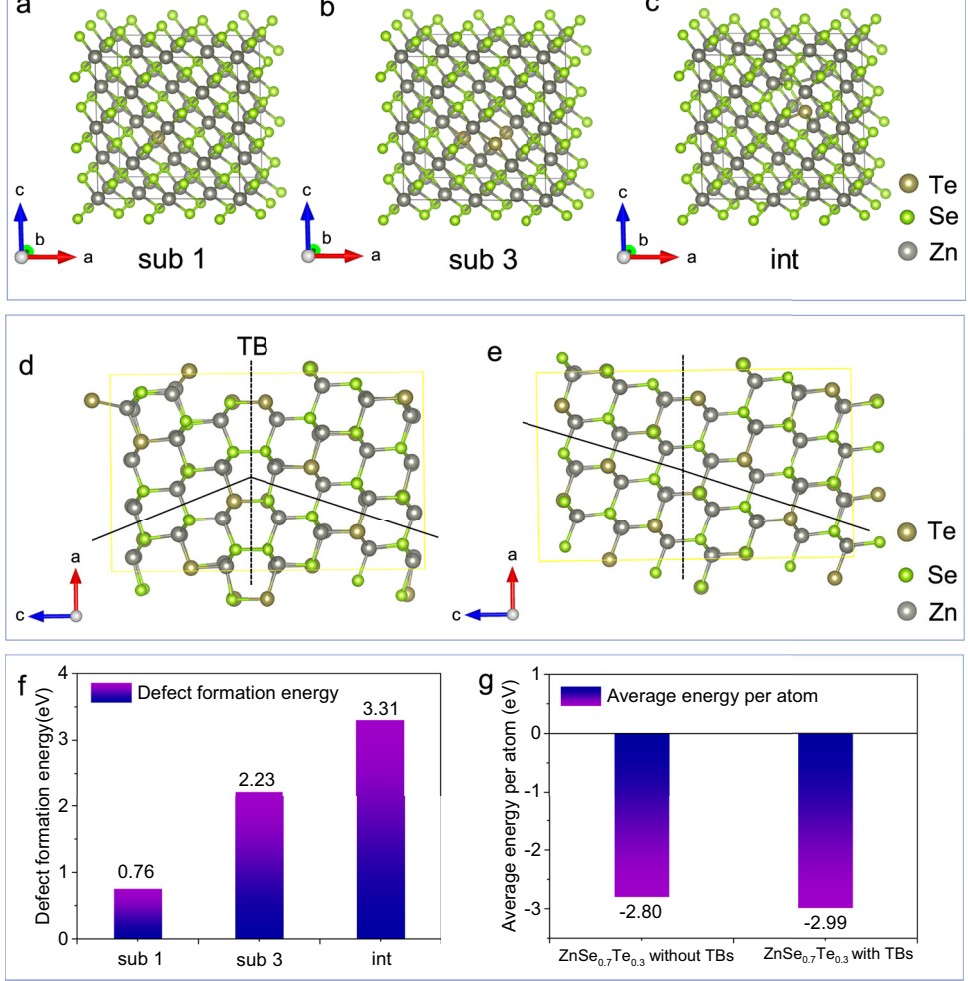

**Fig. 3 | Calculations of defect formation energy and average energy per atom in the final state. a** A phase model in which one Te atom replaces one Se atom (sub 1). **b** A phase model in which three Te atoms replace three Se atoms (sub 3). **c** A phase model with one Te atom occupying an interstitial position (int). Interface models of ZnSe$_{0.7}$Te$_{0.3}$ **d** with TBs and **e** without TBs. **f** Comparison diagram of defect formation energy of three models of sub 1, sub 3, and int. **g** Comparison diagram of average energy per atom in the final state of interface models. Source data are provided as a Source Data file.

photoelectron spectroscopy (XPS). C, N, Se, Te and Zn elements coexist in ZnSe$_{0.7}$Te$_{0.3}$@C nanocomposites, and the atomic ratio of Se and Te is about 0.7 : 0.3 (Supplementary Fig. 9a). The high-resolution C 1$s$ spectrum in Supplementary Fig. 9b involves three main peaks. One at 284.8 eV corresponds to C−C bonds, while the other two peaks at 286.4 and 288.1 eV represent C−N and C=O bonds, respectively[25]. In addition, the N 1$s$ XPS spectrum (Supplementary Fig. 9c) is deconvolved into three remarkable peaks at 398.7, 401.0, and 402.7 eV, respectively, pertaining to pyridinic N, pyrrolic N, and graphite N, indicating that the carbon is doped with N atoms[26]. The binding energy of Se 3$d_{5/2}$ (53.9 eV) and Se 3$d_{3/2}$ (54.9 eV) in the spectrum shows the presence of Se$^{2-}$ in ZnSe$_{0.7}$Te$_{0.3}$@C nanocomposites in Supplementary Fig. 9d. The high-resolution Te 3$d$ spectrum is exhibited in Supplementary Fig. 9e, where four main peaks can be observed. The peaks at 572.7 and 583.0 eV result from Te 3$d_{5/2}$ and Te 3$d_{3/2}$ orbitals, respectively, indicating the presence of Te$^{2-}$. Peaks at 576.2 and 586.7 eV are attributable to its oxide, which is caused by the oxidation of the sample surface[27]. The Zn 2$p$ XPS spectrum (Supplementary Fig. 9f) contains two main peaks at 1044.9 and 1021.9 eV, respectively, coming from Zn 2$p_{1/2}$ and Zn 2$p_{3/2}$ orbitals, a token of bivalent Zn[28].

The thermogravimetric analysis (TGA) curve of ZnSe$_{0.7}$Te$_{0.3}$@C nanocomposites in the air atmosphere is shown in Supplementary Fig. 10. It can be calculated that the content of carbon in the sample is

about 48.6%. Nitrogen adsorption–desorption measurements were carried out to study their porous profiles and specific surface area of ZnSe@C and ZnSe$_{0.7}$Te$_{0.3}$@C nanocomposites, as shown in Supplementary Fig. 11. The adsorption isotherms of them are typical type III isotherms, and the H$_3$ hysteresis appears when the relative pressure of P/P$_0$ is greater than 4.5, indicating the presence of mesoporous structures[29]. According to nitrogen adsorption–desorption measurements, the Brunauer–Emmett–Teller (BET) specific surface area of ZnSe$_{0.7}$Te$_{0.3}$@C nanocomposites is 270.6 m$^2$ g$^{-1}$, which is much greater than that of ZnSe@C (172.7 m$^2$ g$^{-1}$), pertaining to the reduced size of nanoparticles after introduction of Te atoms. Supplementary Fig. 11b shows the pore size distribution, and the pore size of both is mainly distributed in the range of 2–30 nm. The results demonstrate that ZnSe$_{0.7}$Te$_{0.3}$@C has more mesoporous pores than ZnSe@C, which is conducive to promoting the full contact between electrolyte and electrode materials and increasing the transfer rate of sodium ions.

## Electrochemical properties characterization

NaPF$_6$, as a conventional sodium electrolyte salt, has good ionic conductivity[30]. Compared with carbonate solvent, dimethoxyethane (DME) can effectively change the interface and reduce the charge transfer resistance[31]. A thin but stable sodium ion permeable solid electrolyte interface (SEI) layer is easily formed in the electrolyte

NaPF$_6$-DME, facilitating its cycling and rate performance[32]. So, 1 M NaPF$_6$ in DME was used as the electrolyte for cell testing. To evaluate the electrochemical performance of the samples, the synthesized electrodes were assembled into coin-type cells and tested in a 0.01–3 V potential window at 25 °C. To evaluate the effect of twin structures and substitute Te atoms on the sodium ion storage electrochemistry of ZnSe$_{0.7}$Te$_{0.3}$@C nanocomposites, the kinetics analyses were deployed. The diffusion process of sodium ions along and across the TBs of ZnSe$_{0.7}$Te$_{0.3}$ was studied by theoretical calculations. As shown in Fig. 4a–c, obviously, the Na$^+$ diffusion energy barrier along the TBs is 0.45 eV for ZnSe$_{0.7}$Te$_{0.3}$ with twin structures, much lower than that of the defect–free counterpart (0.66 eV). Not only that, as for the Na$^+$ diffusion across the TBs in Fig. 4d–f, ZnSe$_{0.7}$Te$_{0.3}$ with twin structures also shows a lower energy barrier of 0.70 eV than without defects (0.90 eV). Hence, the existence of twin structures is demonstrated to be favorable for reducing the diffusion energy barrier of sodium ions. The kinetics of sodium ion diffusion is improved both along and across the TBs. The existence of a twin structure accelerates the diffusion of sodium ions along different paths. To further verify the simulation conclusions, galvanostatic intermittent titration technique (GITT) curves of the charge/discharge process of ZnSe@C and ZnSe$_{0.7}$Te$_{0.3}$@C were measured to dig out their sodium ion diffusion rates (Supplementary Fig. 12). On this basis, the diffusion coefficients of sodium ions ($D_{Na+}$) were calculated in Fig. 4g, h. It is obvious that the $D_{Na+}$ of ZnSe$_{0.7}$Te$_{0.3}$@C is higher than that of ZnSe@C during the charging and discharging process, in accord with the above theoretical calculation results.

In order to theoretically investigate the effect of the number of TBs on the performance, the model of ZnSe$_{0.7}$Te$_{0.3}$ with two TBs is constructed (Supplementary Fig. 13). And the sodium ion diffusion energy barrier is calculated, as shown in Fig. 4i. Obviously, the Na$^+$ diffusion energy barrier across the TBs is 0.39 eV for ZnSe$_{0.7}$Te$_{0.3}$ with two TBs, much lower than that of the ZnSe$_{0.7}$Te$_{0.3}$ with one TB (0.70 eV). The calculation results show that with the increase of the number of TBs, the sodium diffusion energy barrier is greatly reduced, which helps the kinetics of sodium ion reactions. To further investigate the kinetics of ZnSe@C and ZnSe$_{0.7}$Te$_{0.3}$@C, electrochemical impedance spectroscopy (EIS) tests were performed. The Nyquist plots are fitted by an equivalent circuit model shown in Supplementary Fig. 14a, and the obtained values of resistance are listed in Supplementary Table 5. As shown in Supplementary Fig. 14a, the R$_{ct}$ of ZnSe$_{0.7}$Te$_{0.3}$@C is about only one-third of that of ZnSe@C, implying ZnSe$_{0.7}$Te$_{0.3}$@C has a higher charge transfer rate. The Z′–ω$^{-1/2}$ plot derives from the EIS spectra in Supplementary Fig. 14b, and the slope called Warburg coefficient σ is related to the diffusion of sodium ions in the electrode materials. Obviously, ZnSe$_{0.7}$Te$_{0.3}$@C has a slope of 26.3, much lower than that of ZnSe@C (80.4), further demonstrating the faster sodium ion diffusion of ZnSe$_{0.7}$Te$_{0.3}$@C.

To consolidate the above theoretical analyses, different samples were controlled in terms of the number of twin boundaries by tuning the doping level of tellurium. By changing the usage amount of Se and Te during the synthesis, ZnSe$_{0.8}$Te$_{0.2}$@C, ZnSe$_{0.7}$Te$_{0.3}$@C, and ZnSe$_{0.5}$Te$_{0.5}$@C were prepared. Atomic percentages of C, N, O, Zn, Se, and Te in samples of different compositions were shown by the energy spectrometer attached to the SEM, as shown in Supplementary Table 6. According to the component content in the energy spectrum, the atomic percentage of Se and Te in the sample can be determined, so as to obtain ZnSe$_{0.8}$Te$_{0.2}$@C, ZnSe$_{0.7}$Te$_{0.3}$@C, and ZnSe$_{0.5}$Te$_{0.5}$@C.

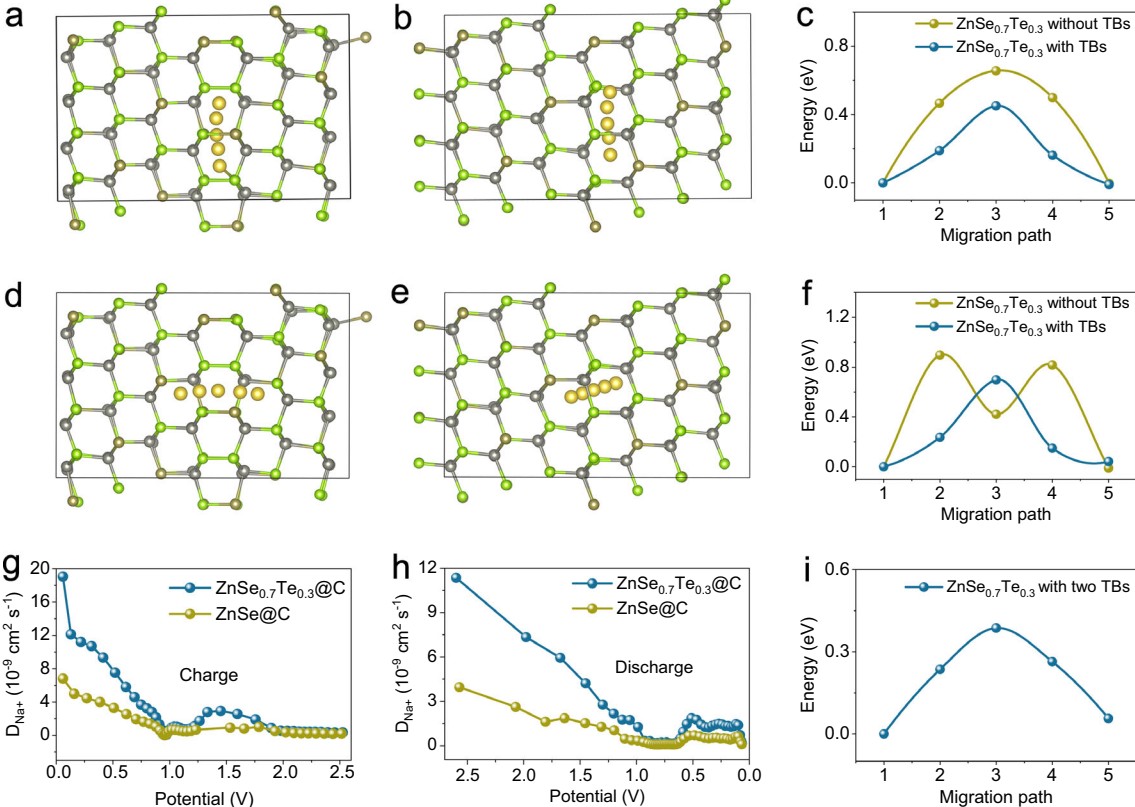

**Fig. 4 | Diffusion mechanism of sodium ions in ZnSe$_{0.7}$Te$_{0.3}$ with and without TBs.** Sodium ion diffusion models for ZnSe$_{0.7}$Te$_{0.3}$ (**a**) with and (**b**) without TBs along the TB (Se: green, Zn: gray, Te: dark yellow, and Na: yellow). The energy barrier of sodium ion diffusing (**c**) along and (**f**) across the TB for ZnSe$_{0.7}$Te$_{0.3}$ with TBs and without TBs. Sodium ion diffusion models for ZnSe$_{0.7}$Te$_{0.3}$ (**d**) with TBs and (**e**) without TBs across the TB. The diffusion rate of sodium ions during (**g**) charging and (**h**) discharging is calculated by GITT. **i** The energy barrier of sodium ion for ZnSe$_{0.7}$Te$_{0.3}$ with two TBs across the TB. Test temperature: 25(±0.5) °C with air convection. Type of electrolyte: 1 M NaPF$_6$ in dimethoxyethane. Source data are provided as a Source Data file.

In order to determine the number of twin boundaries in samples of different components, the samples were characterized by the technique of TEM. Supplementary Fig. 15a–c shows the low magnification TEM images of $ZnSe_{0.8}Te_{0.2}@C$, $ZnSe_{0.7}Te_{0.3}@C$, and $ZnSe_{0.5}Te_{0.5}@C$, in which some of the grains have some fine stripes alternating between light and dark. When these streaks are enlarged, they have a typical twin structure, as shown in Supplementary Fig. 15d–f. Therefore, the grains with twin boundary fringes of different compositions are analyzed statistically and quantitatively. The percentage of grains with twin boundary fringes in $ZnSe_{0.8}Te_{0.2}@C$, $ZnSe_{0.7}Te_{0.3}@C$, and $ZnSe_{0.5}Te_{0.5}@C$ is 4.5%, 11.4%, and 6.2%, respectively (Supplementary Fig. 15g). Compared with $ZnSe_{0.8}Te_{0.2}@C$ and $ZnSe_{0.5}Te_{0.5}@C$, the grain twin boundary ratio of $ZnSe_{0.7}Te_{0.3}@C$ is the largest, which may improve the performance the most.

Then, the rate performance of $ZnSe@C$, $ZnSe_{0.8}Te_{0.2}@C$, $ZnSe_{0.7}Te_{0.3}@C$ and $ZnSe_{0.5}Te_{0.5}@C$ is tested and shown in Fig. 5c. $ZnSe_{0.7}Te_{0.3}@C$ is capable of releasing reversible specific capacities of 351.1, 333.3, 321.7, 310.2, 294.2, 277.2, 256.2 mAh g$^{-1}$ at current densities of 0.2, 0.5, 1, 2, 5, 10 and 20 A g$^{-1}$, respectively. For comparison, $ZnSe@C$ is able to release reversible specific capacities of 245.0, 204.9, 184.5, 169.3, 149.4, 133.9, 121.5 mAh g$^{-1}$ at the same current density. $ZnSe_{0.8}Te_{0.2}@C$ releases specific capacities of 293, 277.9, 264.7, 246.2,

224.8, 206.7, 185.3 mAh g$^{-1}$, respectively, and $ZnSe_{0.5}Te_{0.5}@C$ releases specific capacities of 298.2, 270.4, 246.9, 219.9, 191.7, 170, 152.1 mAh g$^{-1}$, respectively, at the same current density. The rate performance of $ZnSe_{0.7}Te_{0.3}@C$, $ZnSe_{0.8}Te_{0.2}@C$, and $ZnSe_{0.5}Te_{0.5}@C$ is better than that of $ZnSe@C$, which is due to the collective effect of Te atom doping improving conductivity and the twin structure improving sodium ion diffusion dynamics. The rate performance of $ZnSe_{0.7}Te_{0.3}@C$ is better than that of $ZnSe_{0.8}Te_{0.2}@C$ and $ZnSe_{0.5}Te_{0.5}@C$, indicating that $ZnSe_{0.7}Te_{0.3}@C$ has the most appropriate number of TBs and Te atom doping. The rate performance of $ZnSe_{0.5}Te_{0.5}@C$ is slightly poorer than that of $ZnSe_{0.8}Te_{0.2}@C$, which is due to the excessive amount of Te atom doping. The rate performance measurements of other batteries for the $ZnSe_{0.7}Te_{0.3}@C$, $ZnSe@C$, $ZnSe_{0.5}Te_{0.5}@C$ and $ZnSe_{0.8}Te_{0.2}@C$ are shown in Supplementary Fig. 16. The charge and discharge curves of $ZnSe_{0.7}Te_{0.3}@C$ nanocomposites at different current densities are shown in Fig. 5d. When the current density increases from 0.2 to 20 A g$^{-1}$, the voltage gap changes slightly between the charge and discharge voltage platforms, indicating the smaller polarization of reactions. Supplementary Fig. 17 compares the rate properties of zinc–based selenides and tellurides with those previously reported, exhibiting better rate capability, especially at higher rates. In order to study the contribution of

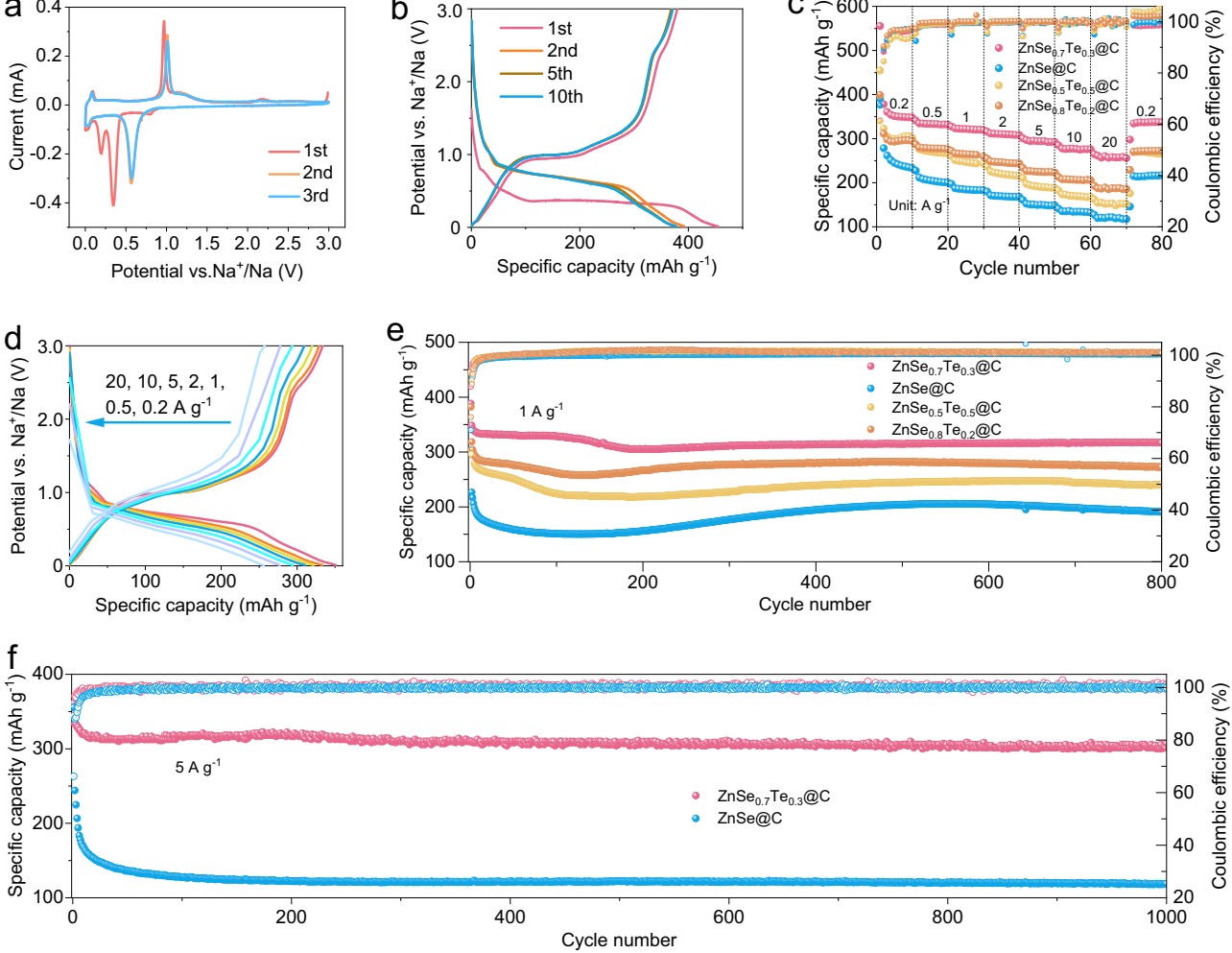

**Fig. 5 | Electrochemical characterizations of the ZnSe@C, ZnSe$_{0.5}$Te$_{0.5}$@C, ZnSe$_{0.8}$Te$_{0.2}$@C, and ZnSe$_{0.7}$Te$_{0.3}$@C nanocomposites. a** CV curves and **b** galvanostatic charge and discharge curves at a current density of 0.2 A g$^{-1}$ of ZnSe$_{0.7}$Te$_{0.3}$@C. **c** Rate performance and **e** long–term cycling at the current density of 1 A g$^{-1}$ of ZnSe@C, ZnSe$_{0.5}$Te$_{0.5}$@C, ZnSe$_{0.8}$Te$_{0.2}$@C and ZnSe$_{0.7}$Te$_{0.3}$@C

(1 A g$^{-1}$ = 1.6 C). **d** Charge and discharge curves of ZnSe$_{0.7}$Te$_{0.3}$@C at different current densities. **f** Long cycle performance at the current density of 5 A g$^{-1}$ of ZnSe@C and ZnSe$_{0.7}$Te$_{0.3}$@C. Test temperature: 25(± 0.5)°C with air convection. Type of electrolyte: 1 M NaPF$_6$ in dimethoxyethane. Source data are provided as a Source Data file.

amorphous carbon to capacity in the sample, the cycle and rate performance were tested, as shown in Supplementary Fig. 18. At a current density of $1 \, A \, g^{-1}$, amorphous carbon has only a specific capacity of $32.3 \, mAh \, g^{-1}$ after 1000 cycles (Supplementary Fig. 18a). Moreover, amorphous carbon releases reversible capacities of 58.2, 35.3, 24.8, 17.7, 12.2, 8.9, $11.5 \, mAh \, g^{-1}$ at current densities of 0.2, 0.5, 1, 2, 5, 10, and $20 \, A \, g^{-1}$ (Supplementary Fig. 18b). Obviously, it contributes little capacity in these hybrid nanocomposites.

Furthermore, Fig. 5e shows the corresponding cycle performance at the current density of $1 \, A \, g^{-1}$. The first discharge capacities of $ZnSe_{0.7}Te_{0.3}@C$, $ZnSe_{0.8}Te_{0.2}@C$, $ZnSe_{0.5}Te_{0.5}@C$, and $ZnSe@C$ are 388.4, 382.0, 445.3, and $303.4 \, mAh \, g^{-1}$, respectively. After 800 cycles, the discharge capacity of $ZnSe_{0.7}Te_{0.3}@C$, $ZnSe_{0.8}Te_{0.2}@C$, and $ZnSe_{0.5}Te_{0.5}@C$ can retain 317.4, 272.5, and $240.3 \, mAh \, g^{-1}$, respectively, significantly higher than that of $ZnSe@C$ ($191.0 \, mAh \, g^{-1}$). This is because the Te atom doping introduces twin plane defects, thereby increasing the active sites and improving the kinetics. Besides, the Te atom doping also reduces the size of the nanoparticles, increases the specific surface area of the active material in contact with the electrolyte, and makes the nanoparticles fully react with sodium ions. Compared with $ZnSe_{0.8}Te_{0.2}@C$ and $ZnSe_{0.5}Te_{0.5}@C$, $ZnSe_{0.7}Te_{0.3}@C$ has the highest specific capacity, indicating that $ZnSe_{0.7}Te_{0.3}@C$ has the optimal Te doping level and number of TBs. The cycle performance of other batteries at the current density of $1 \, A \, g^{-1}$ for smaples is shown in Supplementary Fig. 19. The stability and capacity of batteries at high currents are still significant, so a high-current cycle performance test of $ZnSe_{0.7}Te_{0.3}@C$ and $ZnSe@C$ was carried out, as shown in Fig. 5f. It can be seen that both $ZnSe_{0.7}Te_{0.3}@C$ and $ZnSe@C$ have good cyclic stability. After 1000 cycles, the discharge capacity of $ZnSe_{0.7}Te_{0.3}@C$ remains $307 \, mAh \, g^{-1}$, while that of $ZnSe@C$ is only $118.8 \, mAh \, g^{-1}$. The fine electrochemical performance of $ZnSe_{0.7}Te_{0.3}@C$ is attributed to the twin structures by Te atom doping, the optimal number of TBs, and the improved conductivity. Hence, $ZnSe_{0.7}Te_{0.3}@C$ was determined to continue other electrochemistry testing. The cycle performance of other batteries at the current density of $5 \, A \, g^{-1}$ for the $ZnSe_{0.7}Te_{0.3}@C$ and $ZnSe@C$ is shown in Supplementary Fig. 20.

Considering the promotive action of twin structures and substitional Te doping, $ZnSe_{0.7}Te_{0.3}@C$ has a great prospect as the anode material for SIBs. Cyclic voltammetry (CV) tests were conducted to study the electrochemical reactions in a potential window of 0–3 V vs. $Na^+/Na$, with a scanning speed of $0.2 \, mV \, s^{-1}$, as shown in Fig. 5a. During the first cathodic scan, an unobtrusively wide peak appears at 0.80 V, which results from the formation of the SEI film and the insertion of Na ions into the $ZnSe_{0.7}Te_{0.3}$[33,34]. Two additional strong cathodic peaks at about 0.35 and 0.19 V may be due to the conversion of $ZnSe_{0.7}Te_{0.3}$ to the metal Zn, $Na_2Se$, and $Na_2Te$, and further alloying of Zn[35–37]. During the first anodic scan, oxidation peaks at about 0.97 and 1.16 V are associated with the dealloying reaction of $NaZn_{13}$ and the oxidation of metal Zn to ZnSe, respectively[37]. The small anodic peak at about 2.18 V is related to the oxidation of $Na_2Te$ to Te[25,38]. These analyses will be further demonstrated by the XRD and HRTEM measurements below. A pair of weak redox peaks near 0 V can be attributed to the insertion/extraction of $Na^+$ for the hollow bowl-like carbon[39]. In subsequent scans, the CV curves almost overlapped, indicating good electrochemical reversibility. $ZnSe@C$ has similar CV profiles to $ZnSe_{0.7}Te_{0.3}@C$ in Supplementary Fig. 21a. For comparison, the presence of two cathode peaks at 0.34 V and 0.10 V in $ZnSe@C$ corresponds to the conversion of ZnSe to Zn and $Na_2Se$, and further alloying of Zn. The galvanostatic charge and discharge curves of $ZnSe_{0.7}Te_{0.3}@C$ and $ZnSe@C$ at the current density of $0.2 \, A \, g^{-1}$ are shown in Fig. 5b and Supplementary Fig. 21b, whose charge and discharge platforms are consistent with the redox peaks in CV curves. In addition, the initial coulomb efficiency of $ZnSe_{0.7}Te_{0.3}@C$ is 83.6%, while that of $ZnSe@C$ is 65.9%, which pertains to the fast kinetics

contributed by the twin structures and substitute Te atoms. After the first cycle, the following charge and discharge curves of $ZnSe_{0.7}Te_{0.3}@C$ basically coincide, while the curves of $ZnSe@C$ separate seriously, indicating that $ZnSe_{0.7}Te_{0.3}@C$ has better cyclic stability.

As for the electrochemistry of sodium ion reactions at $ZnSe_{0.7}Te_{0.3}@C$, it was further revealed by ex situ XRD and ex situ TEM characterizations. Figure 6a shows the ex situ XRD pattern of $ZnSe_{0.7}Te_{0.3}@C$ during the first discharge and charge process. Copper foil acting as a current collector shows strong XRD peaks at 43.3°, 50.3°, and 74.0°. The other diffraction peaks are located at 27.1°, 45.0°, 53.4°, 65.7°, and 72.3°, demonstrating the pure phase of $ZnSe_{0.7}Te_{0.3}@C$ present in the original electrode. As the discharge continues, these peak intensities of $ZnSe_{0.7}Te_{0.3}@C$ gradually decrease and then disappear completely. When discharged to 0.4 V, new weak peaks appear at 23.8° & 34.4°, assignable to the production of $Na_2Te$, and peaks at 31.8° & 36.2° are attributable to the production of $NaZn_{13}$ and Zn. When discharged to 0.01 V, a small peak appears at 37.4° due to the formation of $Na_2Se$. This demonstrates that the conversion reaction of $ZnSe_{0.7}Te_{0.3}$ to $Na_2Se$, $Na_2Te$, and $NaZn_{13}$ occurs during the discharge process. During the continuous charging process, these characteristic peaks of $Na_2Se$, $Na_2Te$, and $NaZn_{13}$ gradually weaken and disappear, indicating that $Na_2Se$, $Na_2Te$, and $NaZn_{13}$ gradually transform into ZnSe and Te. To prove it, the electrode material after fully charging was studied using the TEM technique. In Fig. 6b, we can see a large number of hollow carbon shells and several nanoparticles. Further high-resolution observation in Fig. 6c shows clear lattice fringes with 0.33 nm corresponding to the (111) crystal plane of ZnSe, and 0.32 nm from the (101) crystal plane of the Te phase, demonstrating that the final product is ZnSe and Te. The HAADF-STEM image and elemental mapping images shown in Fig. 6d and e can also verify the product. It can be seen that Zn and Se element mapping overlap very well, denoting ZnSe nanoparticles. However, Te element tends to aggregate away from ZnSe nanoparticles, further indicating the formation of Te phase. Figure 6f shows the schematic diagram of the electrochemistry process of $ZnSe_{0.7}Te_{0.3}$ during the first cycle. The electrochemical reactions for the first cycle can be summarized as follows:

Discharging:

$$ZnSe_{0.7}Te_{0.3} + 2Na^+ + 2e^- \rightarrow Zn + 0.7Na_2Se + 0.3Na_2Te$$

$$13Zn + Na^+ + e^- \rightarrow NaZn_{13}$$

Charging:

$$NaZn_{13} \rightarrow 13Zn + Na^+ + e^-$$

$$Zn + Na_2Se \rightarrow ZnSe + 2Na^+ + 2e^-$$

$$Na_2Te \rightarrow 2Na^+ + Te + 2e^-$$

The results show that the initial electrochemical reactions destroyed the $ZnSe_{0.7}Te_{0.3}$ structure, and it cannot retain the original phase. Since the active materials for the following electrochemistry are ZnSe and Te, not $ZnSe_{0.7}Te_{0.3}$ phase, what's the use of constructing $ZnSe_{0.7}Te_{0.3}$ with special defects as the electrode material? Here, we designed a comparative experiment to verify the effect on performance. In view of the low melting point of elemental Te, via the melting-diffusion method, Te nanoparticles of small size were composited with $ZnSe@C$ in the molar ratio of Se: Te = 7: 3 ($ZnSe@C/Te$). It has the same constituents as the active materials after cycling. The $ZnSe@C/Te$ material in the comparison experiment was characterized

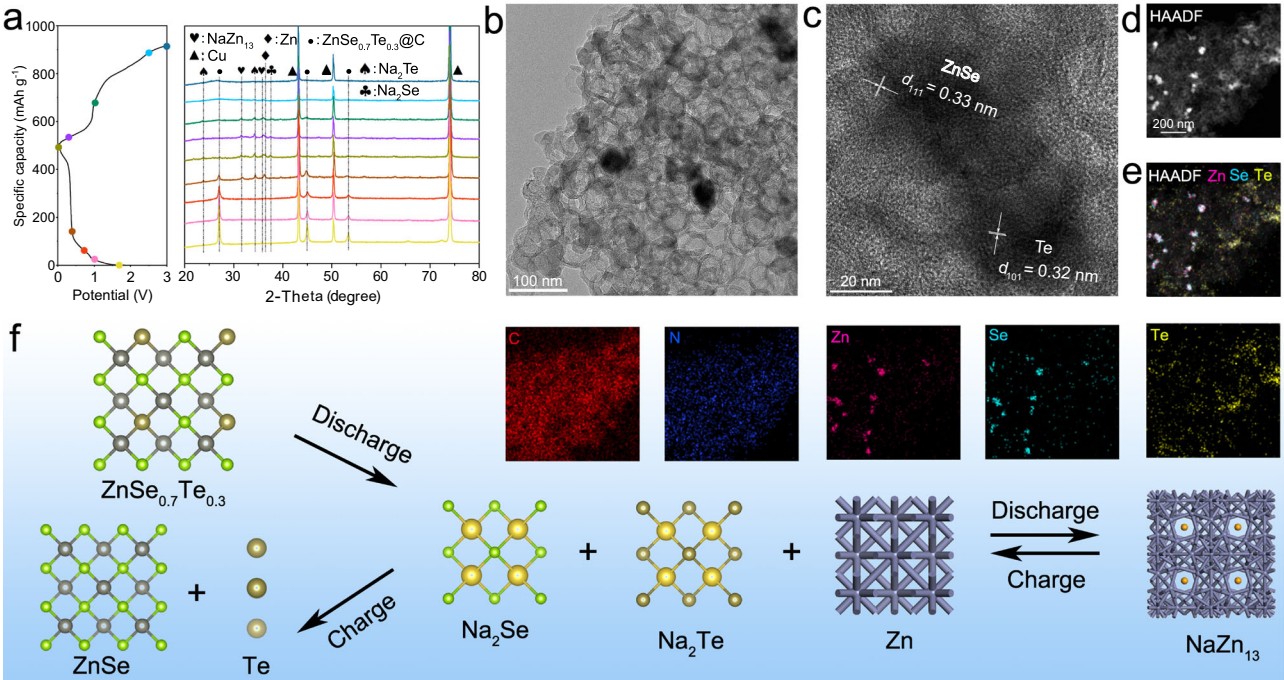

**Fig. 6 | Electrochemical reaction characterization of sodium ion with ZnSe$_{0.7}$Te$_{0.3}$@C nanocomposite. a** Ex situ XRD pattern of the ZnSe$_{0.7}$Te$_{0.3}$@C during the first charge and discharge process. **b** TEM image, **c** HRTEM image, **d** HAADF–STEM image, and **e** elemental mapping images of the ZnSe$_{0.7}$Te$_{0.3}$@C after two full discharge and charge cycles. **f** Schematic diagram of sodium ion reaction mechanism of ZnSe$_{0.7}$Te$_{0.3}$ during the first charge and discharge process. Test temperature: 25(±0.5) °C with air convection. Type of electrolyte: 1 M NaPF$_6$ in dimethoxyethane. Source data are provided as a Source Data file.

by TEM, as shown in Supplementary Fig. 22. It can be seen from the figure that Te and ZnSe nanoparticles are also coated with carbon. Supplementary Fig. 23 shows the cycling performance of ZnSe@C/Te and the parallel battery at the current density of 1 A g$^{-1}$. It can just maintain the lifetime of 339 cycles and then break. A low reversible capacity of 223 mAh g$^{-1}$ was offered, much worse than ZnSe$_{0.7}$Te$_{0.3}$@C (310.9 mAh g$^{-1}$). Moreover, ZnSe@C/Te releases reversible capacities of 292.4, 241.9, 191.4, 157.1, 129.9, 109.1, and 96.4 mAh g$^{-1}$ at current densities of 0.2, 0.5, 1, 2, 5, 10, and 20 A g$^{-1}$ (Supplementary Fig. 24), exhibiting a poor rate capability. Parallel data for ZnSe@C/Te electrode is also provided in Supplementary Fig. 24. Obviously, the initial ZnSe$_{0.7}$Te$_{0.3}$ plays a vital role in sustaining the fine electrochemical performance even though it thoroughly transformed to other materials.

In conclusion, substitutional defects and twin structures were introduced into ZnSe nanocrystals through Te heteroatomic doping. Based on HAADF–STEM, atomic mapping, and theoretical calculation, it is revealed that tellurium partially replaces selenium in ZnSe, which promotes the formation of twin structures verified by calculations. By tuning the composition of Te-doped ZnSe, the optimal composition and number of TBs are obtained for ZnSe$_{0.7}$Te$_{0.3}$. The combined effect of point defects, twin structures, and the optimal number of TBs greatly improves the sodium storage performance of ZnSe$_{0.7}$Te$_{0.3}$@C with a capacity of 307 mAh g$^{-1}$ after 1000 cycles at the current density of 5 A g$^{-1}$. Our work reveals the mechanism of action of Te substitute atoms on twin plane defects, the effect of Te dopant content on the number of twin boundaries, and its effect on electrochemical performance. This provides the theoretical basis of defect engineering for designing the anode materials of sodium-ion batteries with good performance.

## Methods
### Materials
Methanol (≥99.5%), hexadecyltrimethylammonium bromide (C$_{19}$H$_{42}$BrN, ≥99.0%), zinc nitrate hexahydrate (Zn(NO$_3$)$_2$·6H$_2$O, ≥99.0%),

formaldehyde aqueous solution (CH$_2$O, 37.0–40.0%) and ammonia solution (NH$_4$OH, 25.0–28.0%) were bought from Sinopharm Chemical Reagent Co., Ltd. 2-methylimidazole (C$_4$H$_6$N$_2$, 98%), selenium (Se, ≥99.99%), Sodium metal (Na, 99.7%), and tellurium (Te, ≥99.9%) were bought from the Aladdin. M-aminophenol (C$_6$H$_7$NO, 98%) was bought from 9ding chemical (Shanghai) limited. Sodium carboxymethyl cellulose and electrolyte (1 M NaPF$_6$ in dimethoxyethane) were bought from Duoduo Chemical Technology Co., Ltd. None of the reagents were further purified.

### Synthesis of ZIF−8
1.31 g of 2-methylimidazole and 200 mL of methanol were mixed in a 250 mL beaker and magnetically stirred until a clear solution was reached. Then, 2.38 g of zinc nitrate hexahydrate was added to the above solution until it turned milky white under magnetic agitation. The obtained mixture was left for 12 h, then centrifuged, washed with methanol, and dried at 60 °C to collect the white powder of ZIF−8.

### Synthesis of ZIF−8@MF
The above-prepared ZIF−8 powder, 0.46 g of hexadecyltrimethylammonium bromide, 0.2 mL of ammonia solution, 28 mL of ultra-pure water, and 12 mL of ethanol were magnetically stirred in a 50 mL beaker for 6 h. Then 0.07 g of m-aminophenol and 0.12 mL of formaldehyde aqueous solution were added and stirred magnetically for 12 h. Afterward, the solution was extracted and filtered, washed with ultra-pure water, and dried at 60 °C to prepare ZIF−8@MF.

### Synthesis of ZnSe$_{0.8}$Te$_{0.2}$@C, ZnSe$_{0.7}$Te$_{0.3}$@C, ZnSe$_{0.5}$Te$_{0.5}$@C, ZnSe@C, and amorphous carbon
0.2 g of ZIF−8@MF, 0.2 g of Te powder, and 0.12 g of Se powder were ground evenly and then heated in an Ar/H$_2$ mixed atmosphere at 800 °C for 2 h at a heating rate of 4 °C/min to obtain ZnSe$_{0.8}$Te$_{0.2}$@C. 0.2 g of ZIF−8@MF, 0.2 g of Te powder, and 0.1 g of Se powder were ground evenly and then heated in Ar/H$_2$ mixed atmosphere at 800 °C for 2 h at a heating rate of 4 °C/min to obtain ZnSe$_{0.7}$Te$_{0.3}$@C. 0.2 g of

ZIF-8@MF, 0.2 g of Te powder, and 0.07 g of Se powder were ground evenly and then heated in an $Ar/H_2$ mixed atmosphere at 800 °C for 2 h at a heating rate of 4 °C/min to obtain $ZnSe_{0.5}Te_{0.5}$@C. 0.2 g of ZIF-8@MF and 0.15 g of Se powder were ground evenly and then heat-treated under the same conditions to obtain ZnSe@C. 0.32 g of ZnSe@C and 0.14 g of Te powder were ground evenly and then heated in an Ar atmosphere at 480 °C for 1 h at a heating rate of 4 °C/min to obtain ZnSe@C/Te. The black suspension was obtained by fully soaking $ZnSe_{0.7}Te_{0.3}$@C in aqua regia and ultrasonic treatment. Afterward, the solution was extracted and filtered, washed with ultra-pure water, and dried at 60 °C to prepare amorphous carbon.

## Structural characterization

The crystal structures of samples such as ZnSe@C and $ZnSe_{0.7}Te_{0.3}$@C were characterized by X-Ray Diffractometer (XRD, Bruker D8 Advance, Germany, Cu Kα radiation, λ = 1.5418 Å) in the 2θ range from 10° to 80°. The morphology, composition, and microstructure of the samples were characterized by a field emission scanning electron microscope (FESEM, GeminiSEM 300, Carl Zeiss Microscopy Ltd.) coupled with energy-dispersive X-ray spectroscopy (EDS) and a transmission electron microscope (TEM, Talos F200X, accelerating voltage of 200 kV). High-angle annular dark-field-scanning TEM (HAADF-STEM) imaging was done on a Thermo Scientific Spectra 300 equipped with a spherical aberration correction system, and the microscope was operated at 300 kV. The components and valence states of the $ZnSe_{0.7}Te_{0.3}$@C were measured by X-ray photoelectron spectroscopy (XPS, Thermo ESCALAB 250 Xi) with a monochromatic Al Kα X-ray source (hν = 1486.6 eV). And all binding energies were calibrated using C 1 s signals at 284.8 eV. Thermogravimetric analysis (TGA) of the sample was carried out on a Simultaneous Thermal Analyzer (TGA/DSC3 + ) in an air atmosphere at a heating rate of 20 °C/min. Nitrogen adsorption–desorption measurements were carried out on the Autosorb 6B instrument at 70 K.

## Electrochemical measurements

The coin cells (CR2032) were assembled in a glove box filled with argon gas (with $H_2O$ and $O_2 < 0.1$ ppm) to evaluate the electrochemical performance of the samples. A two-electrode system was used in the battery test, in which the prepared composite electrode was used as the working electrode, and sodium foil was used as the counter electrode. The assembled 2032 coin battery consists of a positive and negative stainless steel housing, working electrode, separator, electrolyte, two round stainless steel (16 mm × 0.05 mm), and a stainless steel spring. The active material, acetylene black, and adhesive agent (sodium carboxymethyl cellulose) were dispersed in ultra-pure water at a mass ratio of 7:1.5:1.5 and fully ground in agate mortar to prepare the slurry. The slurry was then cast on a copper foil with a four-sided preparation device (SZQ, Guangzhou Demanyi Instruments Co., Ltd.) and dried in a vacuum oven at 60 °C for 12 h. The dried electrode sheet was cut into a circular working electrode with a diameter of 10 mm by a manual slicing machine (MSK-T10, Shenzhen Kejing Technology Co., Ltd.) without further calendering. The active material loading on the electrode was approximately 0.68 mg cm⁻². The mass basis of the battery refers to the mass of the active material on the electrode. A single clean copper foil with a diameter of 10 mm has a mass of about 6.8 mg. The sodium foil was drilled into a circular sheet with a diameter of 14 mm and a thickness of approximately 0.6 mm. 1 M $NaPF_6$ in dimethoxyethane was used as the electrolyte. The amount of electrolyte added per button cell is approximately 180 μL. A circular glass fiber (GF/F, Whatman) with a diameter of 19 mm acts as the separator for the battery and is about 0.7 μm thick. The Land battery test system (LAND-CT2001A, Wuhan, China) was used to conduct galvanostatic charge/discharge tests at various current densities within the voltage range of 0.01–3 V vs Na⁺/Na. All battery tests were carried out in a constant temperature chamber of 25(±0.5) °C with air convection. The

galvanostatic intermittent titration technique (GITT) was tested on a Land battery testing system (LAND-CT2001A, Wuhan, China) with a potential window of 0.01–3 V vs Na⁺/Na. The cyclic voltammetry (CV) curves of the samples were measured on an electrochemical workstation (Shanghai Chenhua electrochemistry workstation, CHI760D) with a sweep speed of 0.2 mV s⁻¹ and a voltage range of 0–3 V. Electrochemical impedance spectroscopy (EIS) measurements were done on an electrochemical workstation (Shanghai Chenhua electrochemistry workstation, CHI760D), and the amplitude of the AC voltage was set at 5 mV at 100 kHz to 0.01 Hz. The nature of the added signal is potentiostatic, and the number of data points per decade of frequency is 12. The applied quasi-stationary potential is the open circuit potential, and the open circuit voltage application time is 2 s. The electrochemical data provided in the manuscript belong only to a specific battery. The electrodes used for ex situ XRD measurements are obtained by disassembling a charge-discharged battery in a glove box filled with argon gas. Then, the electrodes were placed in a sealed bag in the glove box, and the exposure time of the electrodes in the air was about 5 s before the XRD test. Temperature environment: 25(±2) °C.

## DFT calculations

All calculations were performed on the Vienna ab initio simulation package (VASP)[40] within the frame of density functional theory (DFT)[22,23] with a cutoff energy of 450 eV. The exchange correlation interaction of electrons was described by the generalized gradient approximation (GGA) of Perdew–Burke–Ernzerhof (PBE) functional[41], and the interaction between electrons and ions was described by the projected augmented wave (PAW) method[42]. The Brillouin zone was sampled by a Monkhorst–Pack k-point mesh[43] of 2 × 2 × 2 grid for bulk phase models and 2 × 3 × 1 grid for interface models. In addition, DFT-D3 method[44,45] was used to explain the presence of van der Waals forces within the system. The structural optimizations were done, and the total energy converged within $10^{-5}$ eV. The final force of each ion was below 0.02 eV/Å.

The defect formation energy of structural models was also calculated with the chemical potential of the components, and the calculation formula was as follows:

$$\triangle H_D = E_D - E_h + \sum n_i \mu_i \qquad (1)$$

$$\mu_i = E_i / N_i \qquad (2)$$

where $\triangle H_D$ is the defect formation energy, $E_D$ is the total energy of the supercell with the defect ($D$), $E_h$ is the total energy of the ZnSe supercell, $\mu_i$ is the chemical potential of element Te and Se. $n_i$ is the number of Te and Se atoms that were removed or added to form the system. $E_i$ is the total energy of the solid structure of Se and Te. $N_i$ is the total number of atoms in a solid structure of Se and Te. The chemical potentials of Te and Se were obtained by the solid structure models of Se and Te after structure optimization, as shown in Supplementary Fig. 25.

$$E_{per\ atom} = E_{total} / N_{atom} \qquad (3)$$

$E_{per\ atom}$ is the average energy per atom in the final state, $E_{total}$ is the total energy of the structural model, and $N_{atom}$ is the total number of atoms in the structural model.

## Ab initio molecular dynamics (AIMD) simulation

The ab initio molecular dynamics (MD) simulations were carried out via VASP, with a 300 eV cutoff energy and $10^{-4}$ eV energy convergence. Nose-Hoover thermostat[46–48] was employed in order to control the system at finite temperatures 298 K and 313 K. The time step was 2 fs, and each simulation lasted for 20 ps.

## Data availability
The data supporting the findings of this study are available within this article and its Supplementary Information file, or from the corresponding author upon request. Source data are provided with this paper.

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

## Acknowledgements

This work was supported by the National Natural Science Foundation of China (U21A2077), the Natural Science Foundation of Shandong Province (ZR2022JQ08, ZR2024MB003), Postdoctoral Innovation Project of Shandong Province (SDCX-ZG-202303012), the Taishan Scholars Program of Shandong Province (tsqn202211028), and the Instrument Improvement Founds of Shandong University Public Technology Platform (ts20230209).

## Author contributions

S.L.X. and B.J.X. conceived and designed the research. J.G.Z., S.L.X. and B.J.X. performed the experiments and the characterization of the materials. J.G.Z., S.L.X. and B.J.X. analyzed the data and wrote the manuscript. K.P.S. conducted the electron microscopy experiments and analyzed the data. J.G.Z., Y.Z.L., F.L., M.Z.Z., K.P.S., J.K.F., B.J.X. and S.L.X. have discussed the results and commented on the manuscript. The authors also want to thank Dr. D. Qi from the Electron Microscopy Centre of Shandong University for the help with TEM experiments.

## Competing interests

The authors declare no competing interests.
