## [Transparent Peer Review file · Nature Communications]

Engineering twin structures and substitutional dopants in ZnSe_{0.7}Te_{0.3} anode material for enhanced sodium storage performance

Corresponding Author: Professor Shenglin Xiong

Version 0:

Reviewer comments:

Reviewer #1

(Remarks to the Author)

The paper entitled "Twin structures and substitutional dopants in ZnSe_{0.75}Te_{0.3}: the effect and transition during sodium ion electrochemistry" by Zong et al. discussed the formation of defect structures ZnSe_{0.75}Te_{0.3} by doping Te atom into it and electrochemical characterizations. The work is interesting but there are issues to address before acceptance of the manuscript. I failed to understand the true significance of the DFT results.

Comments:

- 1) Few sentences are difficult to understand e.g. "As most previously reported,..."; "Wang et al ...TBs and in depth understood the role in lithium manganate oxide?". It may be reconstructed.
- 2) "...ZIF-8 exhibits...a size of about 2um." 2um? or typo?
- 3) DFT references are missing when it is first time used.
- 4) "In theory, Te can either replace Se atoms or occupy different interstitial sites in ZnSe". Is there any support of this statement? Why not other different defect types are not possible in theory?
- 5) "The bulk phase formation energy of Se atom positions partially by Te is 1.09 eV, which is much lower than that of Te atoms occupying different interstices (1.45 eV, 1.84 eV), ...". The equation used to calculate the formation energy contains energy of the ZnSe_{0.7}Te_{0.3}, ZnSe and Te. Is it the energy of the free Te atom or energy taken from bulk Te? To mimic the experimental condition better, isn't it better to use chemical potential of the constituents for the formation energy?
- 6) The Kpoints used in the calculation is missed.
- 7) Reference needed for functionals used in the DFT calculation.
- 8) How do authors confirm that the structure used here is not a saddle point but is a true optimized one? Do the system mechanically, thermodynamically and dynamically stable in the DFT simulation? I have not found anything in support of it in the manuscript. Without proving it through DFT simulation what is the meaning of DFT results of a stable structure? I suggest to include these DFT studies that may give you a firm basis of the stability of the structure used here.
- 9) It is expected that at some points theoretical findings should be reflected in experimental results. So a connection between them is established.

Reviewer #2

(Remarks to the Author)

The authors investigate a doping and nano-structuring strategy for the sodium battery anode material ZnSe. They dope the material with some tellurium in place of selenium in order to induce the formation of twin boundaries, which they argue improve the material's properties and electrochemical performance for Na-ion batteries. The authors did a tremendous amount of materials characterization on the as-prepared materials, but the investigation of the mechanisms causing superior performance are not strongly supported with the experimental evidence provided. Additionally, the final composition that is presented is not properly justified. Additionally, the work is not addressed towards the broad scientific readership of this journal. Overall, the manuscript is not suited for publication in Nature Communications and must be rejected at this time. The following comments and suggestions are provided in the spirit of improving the quality of this work.

Abstract

-The abstract is very vague and it is unclear which electrochemical system these materials are targeted being used for.

-The impact and significance of this work and how it will effect the field or how it is relevant to a broader scientific community is also missing

Introduction

-The introduction still does not do a good job of establishing which battery or electrochemical system these materials are being targeted for.

-The introduction is not written for a general scientific audience.

-Are the various defect mechanism covered in the introduction as improving the diffusion rate of Na⁺ ions applicable to all types of Na⁺ active materials? Or only specific structures? Or only specific materials with certain types of reaction mechanisms (i.e. alloying, intercalation, etc)?

-You mention that TM chalcogenides have achieved a lot of attention , but you do not state to what end they have been researched for? Cathodes? Anodes? Providing clarification will give readers in the battery audience better context for comparison.

-The statements made about Te doping in the introduction need a reference or should be rephrased as there is no support for the claims made.

-The results of the new Te-doped material are summarized at the end of the introduction, but the results of a baseline ZnSe material should also be provided for comparison to show how significant the improve was.

-Were other twin-boundary inducing dopants besides Te tried for this material? How did the authors come to settle on Te for doping and on the specific composition that they chose. How are the authors sure that all of the improvements came from the twin boundary formation rather than alternative effects induced by the chemical or Na-storage properties of Te.

- The introduction does not do a good job of framing the significance of Na-ion batteries and how this work will improve our understanding towards designing better Na-ion anode materials. This would also provide context to the broader scientific community.

Materials Synthesis and Characterization

-The conclusions drawn from the HAADF-STEM showing the low contrast of the carbon structure is not immediately clear from Fig. 1d, further explanation on how to interpret the image to come to this conclusion should be provided.

-Would adding in additional Te beyond the Se_{0.7}Te_{0.3} composition induce more twin boundaries and further improve performance? This point ties back to how was this specific composition decided, were a series of materials tried? Were there some calculations run that suggested that this composition would induce the optimal amount of twin boundaries?

-Were base ZnSe and ZnSe_{0.7}Te_{0.3} tried without the carbon nanostructuring to remove the possibility of some synergistic effects?

-Were different types of Te orderings tested, such as Te clusters, in the DFT calculations to see which would be the lowest energy? From the manuscript it looks like only two Te orderings were tried, but were uniform.

-In Figure 3, it does not look like there is any Se vacancy to account for Te in the interstitial sites, this would not properly charge balance and possibly cause the formation energy produced by the calculation to be higher than it should be as there would be additional atoms in the unit cell that should not be there.

-Why is the ZnSe_{0.7}Te_{0.3} material with twin boundaries showing formation energy of -2.8eV when it is stated as 1.09 eV earlier in the same paragraph. It is not clear what is different about these two calculations that would merit the same material showing two different formation energies.

-For the X-ray diffraction data, the analysis is too general and qualitative. LeBail or Rietveld refinement can be performed to show the change in the unit cell from the undoped and doped material and scherrer analysis can be performed to indicate difference in particle sizes. Just stating that the peaks moved to lower angles is insufficient because that can also be caused from a difference in sample height in the instrument itself.

-All references should be added after the period, to avoid confusion with exponents.

-The XPS results indicate the formation of TeO₂, but if there is a significant amount of TeO₂, then this could cause a significant deviation in the sample composition. How was the sample composition verified?

-The TGA results show that there is a lot of carbon in the samples - is the carbon also contributing to the specific capacity of these materials?

-The Te-doped GITT curve looks much shorter than the undoped samples, were the material loadings different? If not it would seem the specific capacity of the doped material is much lower than the undoped material, which contradicts what is stated later in the section on the electrochemical results.

- The diffusion coefficients primarily look lower for the doped material in the 0.5-2.0V, has the conversion reaction already occurred by this point? How much sodium is in the material within this potential range?
- For the band gap calculations, the DOS diagrams in the supporting information show states that are < 1eV for both materials which does not match what is shown in Figure 4i - can the authors comment on this discrepancy.
- The statement that Te improves the electrical conductivity of the material should be removed as just because a material has a smaller bandgap that does not mean it has superior electronic conductivity, it just means the energy required for the material to conduct electrons is lower.
- The authors do not provide an equivalent circuit for how they interpreted their impedance spectra. An equivalent circuit should be provided so the readers have context on how the authors are interpreting their data.
- More information on how the CV peaks are being assigned should be provided. The reference listed (21) does not have any mention of Te, so how are they assigned the Te related reactions?
- Details on the cell testing should be provided in the main body of the manuscript.
- What were the specific capacities on the first cycle? Please add this information to the main body of the text.
- It is mentioned that adding Te to the material increases the specific surface area, but I did not see where the authors conducted surface area measurements.
- Is Figure 6a ex-situ XRD of the electrode? This point should be noted more clearly. Why was ex-situ XRD only conducted after 2 full cycles? Conducting it at various points along the 1st charge discharge curve would be more instructive as to the conversion reaction mechanism and at what voltages different parts of the reaction occur. As it is, the proposed reaction schemes have little data to back them up.
- For the comparative experiment, were the Te nanoparticles also carbon coated? Also, the cycle life of the ZnSe@C/Te material is closer to 350 in the figure (maybe 330-340, but the authors state in text that it is 300).
- Overall there is not enough characterization to justify the proposed reaction mechanism by the authors.

Discussion

- This section should be renamed to conclusion.

Other

- Can the current densities also be provided as C-rates? That would be helpful in comparison to other materials.
- Chemical suppliers and purities should be provided in the methods section.
- More parameters/details should be provided for the structural characterization methods as they were for XRD.
- The electrolyte chosen for cell testing is a bit unusual, some context for this choice would be helpful.
- There is no information on galvanostatic cycling parameters in the methods section.
- What was the voltage range and scan rate for the cyclic voltammetry?
- What was the frequency range and perturbation voltage for EIS measurements?
- What type of electrochemical workstation was used?
- More references should be added in the DFT calculations section (for the PBE functional, for example).
- In Figure 3, a legend should be provided for what color each atom is.
- In Figure 5, cycling curves of the ZnSe base material would be helpful for comparison.
- Supporting information Figure 3b looks like it was cutoff or the figure was not formatted correctly.

Version 1:

Reviewer comments:

Reviewer #1

(Remarks to the Author)

Authors have partly answered my queries and still need some clarification for this work. Following are the points need to address:

1. In fig 6c why there is discontinuity in frequency at R and U points?

Please provide zoom version (lets say -50 Hz to 200 Hz frequency range) of the phonon spectra (Fig 6 c) (specifically Gamma, T and U points). If there is any negative frequency exist, what is its origin and implication? At U point there is large negative frequency so what is the reliability of the calculation?

2. "From Supplementary Fig. 6a and 6b, it can be seen that the phonon spectra of sub 1 model with one Te atom replacing one Se atom and sub 3 model with three Te atoms replacing three Se atoms have no virtual phonon mode, indicating that the models are mechanically and dynamically stable. "

3. Mechanical stability is usually depicted through the calculation of elastic stiffness constant. How authors conclude mechanical stability from phonon spectra should be prescribed in details. And mechanical stability of the structure should be analyzed in details.

4. How chemical potential of Se and Te is calculated is not clear. It should be clearly stated as it involves in formation energy which is related to stability.

5. Since the authors are showing the application of the system for Sodium Ion Batteries (SIBs), so they should also see how these structures behave in room temperature and some higher temperatures through the study of molecular dynamics.

6. In Fig 3, the authors may clearly mark the points which shows the presence of Te in sub 1 and sub 3.

DFT methodology is used in this work but no reference of it. Original reference for DFT should be given.

7. The authors should provide the original references of DFT, software and the functional used in the calculation.

8. "The authors may also explain the reason behind the choice of Monkhorst-Pack k-point mesh of $2 \times 2 \times 2$ grid for bulk phase model and $2 \times 3 \times 1$ grid for interface model in the study". This is not addressed. I failed to understand why along Y direction, 3 points are used for bulk but 2 points for interface. Why? Have you done k point convergence test?

Reviewer #2

(Remarks to the Author)

The authors have adequately responded to each of the reviewer comments and put together a complete study on the effects of Tellurium doping into ZnSe as an anode material for Sodium-ion batteries. The only remaining comment the reviewer has is for the authors to alter the title to be more impactful - something along the lines of explaining the study was performed with the goal of tailoring the properties of the material for performance as a sodium-ion battery anode material. Once this task has been completed, this work is suitable for publication in Nature Communications.

Version 2:

Reviewer comments:

Reviewer #1

(Remarks to the Author)

Authors have adequately incorporated the suggestions made by the reviewer and answered the queries. In my opinion the work may be acceptable now for publication.

Manuscript: NCOMMS-24-22711

Title: Twin structures and substitutional dopants in ZnSe_{0.7}Te_{0.3}: the effect and transition during sodium ion electrochemistry

The authors greatly appreciate reviewers' insightful comments and careful review on our manuscript (NCOMMS-24-22711). This paper has been revised carefully according to the comments of the reviewers. The responses are listed point-by-point in the following contents, and revisions have been highlighted by red color in the revised manuscript. Following are our responses and detailed explanation towards these comments from the reviewers.

Responses to Reviewers:

To Reviewer #1:3-14

To Reviewer #2:15-56

Response to reviewers' comments:

Reviewer #1:

General comments: The paper entitled "Twin structures and substitutional dopants in ZnSe_{0.7}Te_{0.3}: the effect and transition during sodium ion electrochemistry" by Zong et al. discussed the formation of defect structures ZnSe_{0.7}Te_{0.3} by doping Te atom into it and electrochemical characterizations. The work is interesting but there are issues to address before acceptance of the manuscript. I failed to understand the true significance of the DFT results.

Our response: Thank you very much for your comments and suggestions for improving our manuscript. According to the reviewer's suggestions, we have supplemented and modified the calculation part. We hope you can get the significance of the DFT calculations after revision. The detailed revision is as follows:

"In order to explain the formation mechanism of twin structure in ZnSe_{0.7}Te_{0.3}@C, we carried out the density functional theory (DFT) calculations to detect the underlying impetus²². According to the above spherical aberration corrected electron microscopy results, no obvious vacancy defect was found in ZnSe_{0.7}Te_{0.3}@C, and the arrangement of atoms was highly ordered and complete. Therefore, for the perspective of theory, Te can either replace Se atoms or occupy different interstitial sites in ZnSe, considering Te and Se are in the same group²³. In order to further verify theoretically whether Te replaces Se atoms or occupies interstitial sites in ZnSe, we constructed the structural models of different configurations in Fig. 3a–3c and then calculated the corresponding phonon spectra, average energy per atom in the final state, and defect formation energy. From Supplementary Fig. 6a and 6b, it can be seen that the phonon spectra of sub 1 model with one Te atom replacing one Se atom and sub 3 model with three Te atoms replacing three Se atoms have no virtual phonon mode, indicating that the models are mechanically and dynamically stable. However, the phonon spectrum for int model in which Te occupies interstitial position has a tiny virtual phonon mode, proving that its structure is somewhat unstable, as shown in Supplementary Fig. 6c. This may be because larger Te occupies interstitial positions causing gigantic structural distortion, profoundly affecting the normal arrangement of surrounding atoms. The average energy of each atom in the final states of these different configurations is negative, which indicates that they are thermodynamically stable (Supplementary Fig. 6d). The defect formation energy of sub 1 model is 0.76 eV,

which is much lower than that of sub 3 and int models (2.23 and 3.31 eV), as shown in Fig. 3f. This indicates that Te tends to replace Se atoms rather than occupy interstitial positions, which is consistent with the above atom mapping results. Figure 3d and 3e show the atomic interface structure models of $\text{ZnSe}_{0.7}\text{Te}_{0.3}$ with TBs and $\text{ZnSe}_{0.7}\text{Te}_{0.3}$ without TBs based on the HAADF–STEM imaging results. The total number of atoms of $\text{ZnSe}_{0.7}\text{Te}_{0.3}$ models with TBs and without TBs is the same, and the ratio of Se to Te atoms is about 7:3. The average energy per atom in the final state of the $\text{ZnSe}_{0.7}\text{Te}_{0.3}$ with TBs is ~ -2.99 eV, which is lower than that of $\text{ZnSe}_{0.7}\text{Te}_{0.3}$ without TBs (-2.8 eV), indicating that $\text{ZnSe}_{0.7}\text{Te}_{0.3}$ with twin structure is more stable thermodynamically, as shown in Fig. 3g. In addition, the defect formation energy of $\text{ZnSe}_{0.7}\text{Te}_{0.3}$ with TBs is relatively small (0.86 eV), indicating that the twin structure is easy to form. Based on the above analyses, it's known that the introduction of Te substitute atoms will increase the energy of ZnSe system. But, the formation of twin structure can, to some extent, stabilize Te doped $\text{ZnSe}_{0.7}\text{Te}_{0.3}$ system. So, the Te substitutional atoms thermodynamically promote the formation of twin structures in ZnSe.”

The last part is to calculate the sodium diffusion energy barrier of the $\text{ZnSe}_{0.7}\text{Te}_{0.3}$ without TBs, $\text{ZnSe}_{0.7}\text{Te}_{0.3}$ with TBs and $\text{ZnSe}_{0.7}\text{Te}_{0.3}$ with two TBs. The sodium diffusion energy barrier of the $\text{ZnSe}_{0.7}\text{Te}_{0.3}$ with TBs is lower than that of the $\text{ZnSe}_{0.7}\text{Te}_{0.3}$ without TBs, which indicates that the existence of the twin structure promotes the sodium ion diffusion kinetics. The sodium diffusion energy barrier of the $\text{ZnSe}_{0.7}\text{Te}_{0.3}$ with two TBs is lower than that of the $\text{ZnSe}_{0.7}\text{Te}_{0.3}$ with TBs, which indicates that more TBs are beneficial to improve the diffusion kinetics of sodium ions.

1: Few sentences are difficult to understand e.g. "As most previously reported,..." ; "Wang et al ...TBs and in depth understood the role in lithium manganese oxide?". It may to be reconstructed.

Our response: We appreciate the reviewer's suggestions. According to the reviewer's suggestions, we had rewritten these sentences in the revised manuscript. The detailed revisions are as follows:

Line 13-15, Page 4: “Wang et al. studied the formation of TBs in lithium manganese oxide, and also demonstrated TBs can enable fast lithium-ion diffusion and charging

performance¹⁸”.

Line 21-23, Page 4: “The introduction of two kinds of defects into ZnSe can collectively increase adsorption sites and promote reaction kinetics, contributing to better electrochemical sodium storage”.

2: “..ZIF-8 exhibits...a size of about 2um.” 2um? or typo?

Our response: Many thanks to the helpful comments. ZIF–8 exhibits solid dodecahedra with a side length of about 1 μm , as shown in Supplementary Fig. 1. We have revised it in the revised manuscript. The detailed revisions are as follows:

Page 5-6: “From scanning electron microscope (SEM) and transmission electron microscopy (TEM) images in Supplementary Fig. 1, the as–prepared ZIF–8 exhibits solid dodecahedra with a side length of about 1 μm ”.

3: *DFT references are missing when it is first time used.*

Our response: We thank the reviewer for pointing out this missing information.

According to the reviewer’s suggestions, we have added the DFT references when it is first time used. We have added the reference in the revised manuscript. The detailed revisions are as follows:

Line 17-19, Page 7: “In order to explain the formation mechanism of twin structure in $\text{ZnSe}_{0.7}\text{Te}_{0.3}@C$, we carried out the density functional theory (DFT) calculations to detect the underlying impetus²²”.

Page 25:

22. Lei, Y. et al. Understanding the charge transfer effects of single atoms for boosting the performance of Na-S batteries. *Nat. Commun.* **15**, 3325 (2024).

4: *"In theory, Te can either replace Se atoms or occupy different interstitial sites in ZnSe". is there any support of this statement? Why not other different defect types are not possible in theory?*

Our response: Thank you for your good question.

Based on published literature reports, heteroatoms doping into crystals in the form of

substitution, or interstitial atoms (*Nat. Commun.* **2021**, 12, 3085; *J. Mater. Chem. A*, **2020**, 8, 8383-8396; *J. Phys. Chem. C*, **2010**, 114, 10221-10228). Se and Te are in the main group and have similar chemical properties. According to XRD results in Supplementary Fig. 7, ZnSe_{0.7}Te_{0.3}@C maintains the maternal phase from ZnSe with the same standard PDF card, confirming that Te atoms partially replace Se atoms or occupy interstitial positions in the ZnSe lattice. If the Te atom occupies the position of the Zn atom (anions to replace cations), the drastic lattice distortion is supposed to happen and even destroy the maternal lattice. Other phase structures may be generated. Furthermore, according to HAADF-STEM imaging results in Fig. 2b-2h, partial Se atoms are missing and replaced by Te atoms marked by white circle. In summary, it can be concluded that doped Te atoms partially replace Se atoms. In addition, from the experimental point of view, Se and Te powder react with hydrogen at high temperature to produce H₂Se, and H₂Te, respectively. Then, the Zn-containing polymer reacts with H₂Se and H₂Te at the same time to produce ZnSe_{0.7}Te_{0.3}@C composites. In order to support of this statement, we have cited the relevant references on page 7 in the revised manuscript.

Line 19-23, Page 7: "According to the above spherical aberration corrected electron microscopy results, no obvious vacancy defect was found in ZnSe_{0.7}Te_{0.3}@C, and the arrangement of atoms was highly ordered and complete. Therefore, Te can either replace Se atoms or occupy different interstitial sites in ZnSe, considering Te and Se are in the same group²³".

Page 25-26:

23. Liu, B. et al. N-doped carbon modifying MoS₂ nanosheets on hollow cubic carbon for high-performance anodes of sodium-based dual-ion batteries. *Adv. Funct. Mater.* **31**, 2101066 (2021).

5: *The bulk phase formation energy of Se atom positions partially by Te is 1.09 eV, which is much lower than that of Te atoms occupying different interstices (1.45 eV, 1.84 eV), ...". The equation used to calculate the formation energy contains energy of the ZnSe_07_Te_0.3, ZnSe and Te. Is it the energy of the free Te atom or energy*

taken from buld Te? To mimic the experimental condition better, isn't it better to use chemical potential of the constituents for the formation energy?

Our response: Many thanks to the helpful comments and valuable suggestions. In the previous calculation, it's the energy of free Te atom, which is common used in reported calculations. Based on your suggestion, we have re-calculated the formation energy of defects by using chemical potential of the constituents. In order to make this part of the calculation more accurate, we also expand the cell based on the previous one and added the calculation of replacing three Se atoms in ZnSe with three Te atoms. These revisions have added in the revised manuscript. The detailed revision is as follows:

Page 7-8: “In order to further verify theoretically whether Te replaces Se atoms or occupies interstitial sites in ZnSe, we constructed the structural models of different configurations in Fig. 3a–3c and then calculated the corresponding phonon spectra, average energy per atom in the final state, and defect formation energy. From Supplementary Fig. 6a and 6b, it can be seen that the phonon spectra of sub 1 model with one Te atom replacing one Se atom and sub 3 model with three Te atoms replacing three Se atoms have no virtual phonon mode, indicating that the models are mechanically and dynamically stable. However, the phonon spectrum for int model in which Te occupies interstitial position has a tiny virtual phonon mode, proving that its structure is somewhat unstable, as shown in Supplementary Fig. 6c. This may be because larger Te occupies interstitial positions causing gigantic structural distortion, profoundly affecting the normal arrangement of surrounding atoms. The average energy of each atom in the final states of these different configurations is negative, which indicates that they are thermodynamically stable (Supplementary Fig. 6d). The defect formation energy of sub 1 model is 0.76 eV, which is much lower than that of sub 3 and int models (2.23 and 3.31 eV), as shown in Fig. 3f.”

In the “Methods” part, the detailed information has been added as follows:

Page 23: “The defect formation energy of structural models was also calculated with the chemical potential of the components and the calculation formula was as follows:

$$\Delta H_D = E_D - E_h + \sum n_i \mu_i$$

where ΔH_D is the defect formation energy, E_D is the total energy of the supercell with the defect (D), E_h is the total energy of the ZnSe supercell, μ_i is the chemical potentials of element Te and Se. n_i is the numbers of Te and Se atoms that were removed or added to form the system.

$$E_{\text{per atom}} = E_{\text{total}}/N_{\text{atom}}$$

$E_{\text{per atom}}$ is the average energy per atom in the final state, E_{total} is the total energy of the structural model, and N_{atom} is the total number of atoms in the structural model.”

Fig. 3 | Calculations of defect formation energy and average energy per atom in the final state. a A phase model in which one Te atom replaces one Se atom (sub 1). **b** A phase model in which three Te atoms replace three Se atoms (sub 3). **c** A phase model with one Te atom occupying interstitial position (int). Interface models of

ZnSe_{0.7}Te_{0.3} (d) with TBs and (e) without TBs. f Comparison diagram of defect formation energy of three models of sub 1, sub 3 and int. g Comparison diagram of average energy per atom in the final state of interface models.

Supplementary Fig. 6 Phonon spectra for the structural models of (a) sub 1, (b) sub 3 and (c) int. d The average energy per atom in the final state of different configuration structures.

6: The K points used in the calculation is missed.

Our response: We thank the reviewer for pointing out this missing information.

According to the reviewer's suggestion, we have added the K points used in the calculation in the revised manuscript. The detailed revision is as follows:

Page 22-23: "The Brillouin zone was sampled by a Monkhorst-Pack k-point mesh of $2 \times 2 \times 2$ grid for bulk phase model and $2 \times 3 \times 1$ grid for interface model".

7: Reference needed for functionals used in the DFT calculation.

Our response: We thank the reviewer for pointing out this missing information.

According to the reviewer's suggestions, we have added the DFT references in the revised manuscript. The detailed revision is as follows:

Page 22-23: "All calculations were performed on the Vienna ab initio simulation package (VASP) with a cutoff energy of 450 eV⁴³. The exchange correlation interaction of electrons was described by the generalized gradient approximation (GGA) of Perdew–Burke–Ernzerhof (PBE) functional⁴⁴, and the interaction between electrons and ions was described by the projected augmented wave (PAW) method⁴⁵. The Brillouin zone was sampled by a Monkhorst-Pack k-point mesh of $2 \times 2 \times 2$ grid for bulk phase models and $2 \times 3 \times 1$ grid for interface models."

Page 28:

43. Zhao, S., Tang, Y., Yu, X. & Li, J. Superior reactivity of heterogeneous single-cluster catalysts for semi-hydrogenation of acetylene. *Sci. China Mater.* **66**, 3912-3921 (2023).
44. Hao, Y. et al. Plasma-treated ultrathin ternary FePSe₃ nanosheets as a bifunctional electrocatalyst for efficient zinc-air batteries. *ACS Appl. Mater. Interfaces* **12**, 29393–29403 (2020).
45. Xia, H. et al. Evolution of stabilized 1T-MoS₂ by atomic-interface engineering of 2H-MoS₂/Fe-N_x towards enhanced sodium ion storage. *Angew. Chem. Int. Ed.* **62**, e202218282 (2023).

8: How do authors confirmed that the structure used here is not a saddle point but is a true optimized one? Do the system mechanically, thermodynamically and dynamically stable in the DFT simulation? I have not found anything in support of it in the manuscript. Without proving it through DFT simulation what is the meaning of DFT results of a stable structure? I suggest to include these DFT studies that may give you a firm basis of the stability of the structure used here.

Our response: We appreciate the reviewer's suggestions. Accordingly, the phonon spectra of the structural models have been calculated to verify that the structural models used are truly optimized, and to verify the mechanical and dynamic stability of the systems. In addition, the average energy per atom in the final state of the models

has also been calculated to verify the thermodynamic stability. These revisions have been added in the revised manuscript. The detailed revision is as follows:

Page 7-8: “In order to further verify theoretically whether Te replaces Se atoms or occupies interstitial sites in ZnSe, we constructed the structural models of different configurations in Fig. 3a–3c and then calculated the corresponding phonon spectra, average energy per atom in the final state, and defect formation energy. From Supplementary Fig. 6a and 6b, it can be seen that the phonon spectra of sub 1 model with one Te atom replacing one Se atom and sub 3 model with three Te atoms replacing three Se atoms have no virtual phonon mode, indicating that the models are mechanically and dynamically stable. However, the phonon spectrum for int model in which Te occupies interstitial position has a tiny virtual phonon mode, proving that its structure is somewhat unstable, as shown in Supplementary Fig. 6c. This may be because larger Te occupies interstitial positions causing gigantic structural distortion, profoundly affecting the normal arrangement of surrounding atoms. The average energy of each atom in the final states of these different configurations is negative, which indicates that they are thermodynamically stable (Supplementary Fig. 6d).”

Page 23: “The defect formation energy of structural models was also calculated with the chemical potential of the components and the calculation formula was as follows:

$$\Delta H_D = E_D - E_h + \sum n_i \mu_i$$

where ΔH_D is the defect formation energy, E_D is the total energy of the supercell with the defect (D), E_h is the total energy of the ZnSe supercell, μ_i is the chemical potentials of element Te and Se. n_i is the numbers of Te and Se atoms that were removed or added to form the system.

$$E_{\text{per atom}} = E_{\text{total}}/N_{\text{atom}}$$

$E_{\text{per atom}}$ is the average energy per atom in the final state, E_{total} is the total energy of the structural model, and N_{atom} is the total number of atoms in the structural model.”

Fig. 3 | Calculations of defect formation energy and average energy per atom in the final state. a A phase model in which one Te atom replaces one Se atom (sub 1). **b** A phase model in which three Te atoms replace three Se atoms (sub 3). **c** A phase model with one Te atom occupying interstitial position (int). Interface models of $\text{ZnSe}_{0.7}\text{Te}_{0.3}$ **(d)** with TBs and **(e)** without TBs. **f** Comparison diagram of defect formation energy of three models of sub 1, sub 3 and int. **g** Comparison diagram of average energy per atom in the final state of interface models.

Supplementary Fig. 6 Phonon spectra for the structural models of (a) sub 1, (b) sub 3 and (c) int. d The average energy per atom in the final state of different configuration structures.

9: It is expected that at some points theoretical findings should be reflected in experimental results. So a connection between them is established.

Our response: Thanks for your kind consideration. Perhaps we did not make a clear explanation of the connection between theoretical calculation and experimental results, so the reviewers did not understand the significance of our calculation. Firstly, we construct structural models with different configurations (sub 1, sub 3, int), calculate the defect formation energy, and prove that Te atoms replace Se atoms in ZnSe. Then, we construct the twin and non-twin interface models of ZnSe_{0.7}Te_{0.3}, and calculate the average energy per atom in the final state, which proves that the thermodynamic stability of the twin structure is better than that of the non-twin structure. The last part is to calculate the sodium diffusion energy barrier of the

ZnSe_{0.7}Te_{0.3} without TBs, ZnSe_{0.7}Te_{0.3} with TBs and ZnSe_{0.7}Te_{0.3} with two TBs. The sodium diffusion energy barrier of the ZnSe_{0.7}Te_{0.3} with TBs is lower than that of the ZnSe_{0.7}Te_{0.3} without TBs, which indicates that the existence of the twin structure promotes the sodium ion diffusion kinetics. The sodium diffusion energy barrier of the ZnSe_{0.7}Te_{0.3} with two TBs is lower than that of the ZnSe_{0.7}Te_{0.3} with TBs, which indicates that more TBs are beneficial to improve the diffusion kinetics of sodium ions. These revisions have been added in the revised manuscript. The detailed revision is as follows:

Page 7-9: “According to the above spherical aberration corrected electron microscopy results, no obvious vacancy defect was found in ZnSe_{0.7}Te_{0.3}@C, and the arrangement of atoms was highly ordered and complete. Therefore, Te can either replace Se atoms or occupy different interstitial sites in ZnSe, considering Te and Se are in the same group²³. In order to further verify theoretically whether Te replaces Se atoms or occupies interstitial sites in ZnSe, we constructed the structural models of different configurations in Fig. 3a–3c and then calculated the corresponding phonon spectra, average energy per atom in the final state, and defect formation energy. From Supplementary Fig. 6a and 6b, it can be seen that the phonon spectra of sub 1 model with one Te atom replacing one Se atom and sub 3 model with three Te atoms replacing three Se atoms have no virtual phonon mode, indicating that the models are mechanically and dynamically stable. However, the phonon spectrum for int model in which Te occupies interstitial position has a tiny virtual phonon mode, proving that its structure is somewhat unstable, as shown in Supplementary Fig. 6c. This may be because larger Te occupies interstitial positions causing gigantic structural distortion, profoundly affecting the normal arrangement of surrounding atoms. The average energy of each atom in the final states of these different configurations is negative, which indicates that they are thermodynamically stable (Supplementary Fig. 6d). The defect formation energy of sub 1 model is 0.76 eV, which is much lower than that of sub 3 and int models (2.23 and 3.31 eV), as shown in Fig. 3f. This indicates that Te tends to replace Se atoms rather than occupy interstitial positions, which is consistent with the above atom mapping results. Figure 3d and 3e show the atomic interface structure models of ZnSe_{0.7}Te_{0.3} with TBs and ZnSe_{0.7}Te_{0.3} without TBs based on the HAADF–STEM imaging results. The total number of atoms of ZnSe_{0.7}Te_{0.3} models with TBs and without TBs is the same, and the ratio of Se to Te atoms is about 7:3.

The average energy per atom in the final state of the $\text{ZnSe}_{0.7}\text{Te}_{0.3}$ with TBs is ~ -2.99 eV, which is lower than that of $\text{ZnSe}_{0.7}\text{Te}_{0.3}$ without TBs (-2.8 eV), indicating that $\text{ZnSe}_{0.7}\text{Te}_{0.3}$ with twin structure is more stable thermodynamically, as shown in Fig. 3g. In addition, the defect formation energy of $\text{ZnSe}_{0.7}\text{Te}_{0.3}$ with TBs is relatively small (0.86 eV), indicating that the twin structure is easy to form. Based on the above analyses, it's known that the introduction of Te substitute atoms will increase the energy of ZnSe system. But, the formation of twin structure can, to some extent, stabilize Te doped $\text{ZnSe}_{0.7}\text{Te}_{0.3}$ system. So, the Te substitutional atoms thermodynamically promote the formation of twin structures in ZnSe.”

Thanks for your comments again. If it could be considered for publication in **Nat. Commun.**, it would be greatly appreciated!

Reviewer #2:

General comments: The authors investigate a doping and nano-structuring strategy for the sodium battery anode material ZnSe. They doping the material with some tellurium in place of selenium in order to induce the formation of twin boundaries, which they argue improve the material's properties and electrochemical performance for Na-ion batteries. The authors did a tremendous amount of materials characterization on the as-prepared materials, but the investigation of the mechanisms causing superior performance are not strongly supported with the experimental evidence provided. Additionally, the final composition that is presented is not properly justified. Additionally, the work is not addressed towards the broad scientific readership of this journal. Overall, the manuscript is not suited for publication in Nature Communications and must be rejected at this time. The following comments and suggestions are provided in the spirit of improving the quality this work.

Our response: Thank you very much for your comments and kind suggestions for further improving our manuscript. During the revision, a series of experiments and discussions have been performed to provide more evidence to support our results and improve the quality of this manuscript. In addition, we have revised the article for ease of reading for a broad scientific readership. And we have proved and modified the last paragraph of the article. Please also see the detailed responses to the following comments.

Abstract:

Q1: The abstract is very vague and it is unclear which electrochemical system these materials are targeted being used for.

Q2: The impact and significance of this work and how it will effect the field or how it is relevant to a broader scientific community is also missing.

Our response for Q1-2: Many thanks to the helpful comments. We have re-phrased the whole abstract to confirm that the materials we made are for sodium-ion batteries and to illustrate the impact and significance of this work. The detailed revision is as follows:

Page 2: “Compared with lithium-ion batteries (LIBs), sodium-ion batteries (SIBs) are an alternative technology for future energy storage due to their abundant resources and economic benefits. Constructing various defects is considered to be a common viable means of improving the performance of sodium storage. However, it is of significance to thoroughly scrutinize the formation mechanism of defects and their effects and transition during the charge–discharge process. Here, twin structures are introduced into $\text{ZnSe}_{0.7}\text{Te}_{0.3}$ nanocrystals by doping of Te heteroatoms. The Te dopants are visualized to locate in the lattices of ZnSe by spherical aberration electron microscopy. The formation of twin structures is thermodynamically promoted by Te heteroatoms partially replacing Se based on the theoretical calculation results. Moreover, calculation results show that with the increase of twin boundaries (TBs), the sodium diffusion energy barrier is greatly reduced, which helps the kinetics of sodium ion diffusion. In the connection, the composition and amount of TBs are optimized via tuning the doping level. The combined effect of point defects and twin structures greatly improves the sodium storage performance of $\text{ZnSe}_{0.7}\text{Te}_{0.3}\text{@C}$. Our work reveals the mechanism of the point defect on the twin plane defect and systematically investigates their effect on the electrochemical performance, which is greatly helpful for the elegant design of advanced SIBs with longlife and rapid charge-discharge capability.”

Introduction

Q3: The introduction still does not do a good job of establishing which battery or electrochemical system these materials are being targeted for.

Q4: The introduction is not written for a general scientific audience.

Our response for Q3-4: We appreciate the reviewer’s suggestions.

According to the reviewer’s suggestions including the below ones, we have re-written the whole introduction part aimed for a general scientific audience. The detailed revision is as follows:

Page 3-5: “In recent years, sodium-ion batteries (SIBs) have made great progress, and are expected to be one of the best substitutes for widely commercialized lithium-ion batteries (LIBs), due to the abundant reserves of sodium resources and similar electrochemistry and operation mechanisms^{1,2}. As a typical anode material for

LIBs, graphite has been widely used. In view of the limited battery capacity and structural instability of sodium-graphite intercalation compounds, it's urgent to search high-capacity and long-life SIBs anode materials to meet the needs of large-scale energy storage applications^{3,4}. Thanks to their structural diversity and high theoretical capacity (about 400-600 mAh g⁻¹), transition metal chalcogenides (TMC) have great application potential in the anode of SIBs⁵. ZnSe, an important member of TMC, is an alloy-conversion combined anode material prospective for SIBs. It has decent electrical conductivity better than its oxide counterpart, and weak metal-selenium (Se) bond that facilitates the electrochemical conversion reactions⁶. But, it has some problems such as poorer electronic conductivity than carbon, sluggish kinetics and large volume change in the long-term charge-discharge process, which limits its rapid charge-discharge ability and structural stability. To solve these problems, various strategies have been developed, such as combining ZnSe with carbon-based materials, and constructing robust three-dimensional nanostructures. Moreover, defect engineering is demonstrated to be an effective method to reasonably design electrode materials for rechargeable batteries to achieve the improved electrochemical performance⁷.

Defects in crystals can be classified into point defects, line defects, and planar defects from the dimension of defects. They have significant effects on the chemical properties, thermal stability and mechanical properties of materials^{8,9}. Point defects such as vacancies, substitute and interstitial atoms can increase adsorption sites, accelerate the ion diffusion, and improve the electronic conductivity of the LIB and SIB electrode materials¹⁰, such as Co and F codoped SnO₂¹¹, MoS₂/C with S vacancies¹², Cu-doping cobalt embedded nitrogen-doped porous carbon (CoCu@NC)¹³. Dislocations, one kind of line defects, can prevent cracking, loss of active materials and adverse interface reactions with electrolytes by reducing strain during the phase transition of spinel material LiNi_{0.5}Mn_{1.5}O₄¹⁴. In addition, the diffusion rate of Na⁺ at grain boundaries (planar defects) is much faster than in the bulk phase of Ta⁵⁺-substituted Na₃V₂(PO₄)₃¹⁵. As a member of special planar defects, twin boundaries (TBs) also often appear in crystal materials with twin structure. The existence of twin structure is conducive to the diffusion of lithium ions in electrode materials, which helps to improve the electrochemical dynamics of batteries^{8,16}. For instance, Nie et al demonstrated that TBs promote the diffusion of lithium ions in single-crystal SnO₂ nanowires¹⁷. Wang et al. studied the formation of TBs in lithium

manganate oxide, and also demonstrated TBs can enable fast lithium-ion diffusion and charging performance¹⁸. It follows that the defect investigations have been done thoroughly especially in LIBs; however, more efforts should be made on the study of anode materials for SIBs. The electrical conductivity of Te is about $2 \times 10^2 \text{ S cm}^{-1}$, which is much higher than that of Se ($1 \times 10^{-4} \text{ S cm}^{-1}$)¹⁹. In addition, Se and Te atoms are in the same main group, and the latter has a slightly larger radius than the former²⁰. This indicates that Te heteroatom doping of ZnSe may improve its electrical conductivity and potentially introduce some additional defects. The introduction of two kinds of defects into ZnSe can collectively increase adsorption sites and promote reaction kinetics, contributing to better electrochemical sodium storage. At the same time, the systematic investigations of the formation, effects, and transition of defects during the electrochemistry also require further efforts.

Herein, we prepared $\text{ZnSe}_{0.7}\text{Te}_{0.3}$ nanocrystals with twin structure as anode material for SIBs using zeolitic imidazolate frameworks (ZIF-8) as template and Te heteroatom doping hybridized with ultra-thin hollow carbon structure ($\text{ZnSe}_{0.7}\text{Te}_{0.3}@\text{C}$). The doping of heterogeneous Te increased the energy of the system and lattice distortion. Alternatively, the crystal matrix introduced TB defects to alleviate this tendency and maintain the system stable. Moreover, via the composition adjustment during the synthesis, $\text{ZnSe}_{0.7}\text{Te}_{0.3}$ was determined as the optimized Te doping level with optimal TB amount. By combination of a series of structural characterizations, the electrochemical reactions of $\text{ZnSe}_{0.7}\text{Te}_{0.3}$ with sodium ions were confirmed, which also demonstrated the transition of defects in $\text{ZnSe}_{0.7}\text{Te}_{0.3}$ during the charging/discharging. $\text{ZnSe}_{0.7}\text{Te}_{0.3}@\text{C}$ electrode shows the significantly superior sodium-storage properties to pristine $\text{ZnSe}@\text{C}$ electrode, including a higher capacity (5 A g^{-1} ; 307 mAh g^{-1} vs 118.8 mAh g^{-1} after 1000 cycles), better rate performance (20 A g^{-1} ; 256.2 vs 121.5 mAh g^{-1}). The excellent storage performance results from the promotive effect of two defect dimensions, TB (planar defect) and substitution dopant (point defect) in $\text{ZnSe}_{0.7}\text{Te}_{0.3}$. When designing the anode materials of SIBs, the defects of two dimensions are introduced at the same time, which will overcome their shortcomings from different aspects, and finally realize the comprehensive and significant improvement of sodium storage performance.”

Q5: Are the various defect mechanism covered in the introduction as improving

the diffusion rate of Na⁺ ions applicable to all types of Na⁺ active materials? Or only specific structures? Or only specific materials with certain types of reaction mechanisms (i.e. alloying, intercalation, etc)?

Our response: Thank you for your questions.

Defects in crystals can be classified into point defects, line defects, and planar defects from the dimension of defects. They are mentioned in the manuscript. Point defects mainly include vacancy, interstitial atom, substitute atom, etc., which are studied most in the field of electrode materials. Point defects can increase electrochemical activity and adsorption sites, and improve the material's electronic conductivity and ion diffusion rate. Point defects are applicable to all types of Na⁺ active materials. For example, Yan Zhang et al. introduced oxygen vacancy in TiO₂(B), which improved the material conductivity and accelerated the sodium ion diffusion kinetics (*Adv. Funct. Mater.* **2017**, 27, 1700856). Xia Wen et al. doped MoS₂ with single atom cobalt to change the electronic structure of MoS₂, improve the conductivity, and promote the migration of sodium ions (*Adv. Mater.* **2023**, 35, 2211690). Shi Tao et al. co-doped with nitrogen and phosphorus on mesoporous hard carbon to improve the diffusion, adsorption and electron transport properties of Na ions (*Carbon* **2021**, 178, 233-242). As an important type of defect, the existence of planar defects will inevitably influence ionic diffusion in electrode materials for batteries. The twin plane defects increase the diffusion rate of Na⁺/Li⁺, which is suitable for various Na⁺/Li⁺ active materials. In the past few years, twin plane defects have been mainly studied in lithium-ion batteries. For example, for oxides, Anmin Nie et al. demonstrated that TBs promote the diffusion of lithium ions in single-crystal SnO₂ nanowires (*Nano Lett.* **2015**, 15, 610-615). In addition, twin boundary is mostly used in lithium-ion battery cathode materials, such as LiCoO₂, lithium rich manganese oxide and so on. For example, Yali Yang et al. preliminarily synthesized lithium-rich Mn-based oxides with quasi-three-dimensional lithium-ion diffusion channels by introducing twin structures with high lithium-ion diffusion coefficients into crystals (*Adv. Mater.* **2023**, 35, 2307138). Recently, in view of the promotion effect of twin plane defects on the diffusion of lithium ions, researchers have introduced twin plane defects into sodium-ion batteries for research

(*J. Mater. Chem. A*, **2022**, 10, 23799-23810; *J. Mater. Chem. A*, **2020**, 8, 8049-8057; *Chem. Commun.*, **2023**, 59, 10785-10788). For example, Zuguang Yang et al. prepared a platelet-like CuS material with twin crystals for sodium ion battery negative electrode, and obtained excellent rate performance (*J. Mater. Chem. A*, **2020**, 8, 8049-8057). In order to make this part more understandable, we have rephrased it in the revised manuscript. The detailed revision is as follows:

Page 3-4: “Defects in crystals can be classified into point defects, line defects, and planar defects from the dimension of defects. They have significant effects on the chemical properties, thermal stability and mechanical properties of materials^{8,9}. Point defects such as vacancies, substitute and interstitial atoms can increase adsorption sites, accelerate the ion diffusion, and improve the electronic conductivity of the LIB and SIB electrode materials¹⁰, such as Co and F codoped SnO₂¹¹, MoS₂/C with S vacancies¹², Cu-doping cobalt embedded nitrogen-doped porous carbon (CoCu@NC)¹³. Dislocations, one kind of line defects, can prevent cracking, loss of active materials and adverse interface reactions with electrolytes by reducing strain during the phase transition of spinel material LiNi_{0.5}Mn_{1.5}O₄¹⁴. In addition, the diffusion rate of Na⁺ at grain boundaries (planar defects) is much faster than in the bulk phase of Ta⁵⁺-substituted Na₃V₂(PO₄)₃¹⁵. As a member of special planar defects, twin boundaries (TBs) also often appear in crystal materials with twin structure. The existence of twin structure is conducive to the diffusion of lithium ions in electrode materials, which helps to improve the electrochemical dynamics of batteries^{8,16}. For instance, Nie et al demonstrated that TBs promote the diffusion of lithium ions in single-crystal SnO₂ nanowires¹⁷. Wang et al. studied the formation¹⁷ of TBs in lithium manganate oxide, and also demonstrated TBs can enable fast lithium-ion diffusion and charging performance¹⁸. It follows that the defect investigations have been done thoroughly especially in LIBs;”

Q6: You mention that TM chalcogenides have achieved a lot of attention, but you do not state to what end they have been researched for? Cathodes? Anodes? Providing clarification will give readers in the battery audience better

context for comparison.

Our response: Thank you for your questions. According to the reviewer's suggestions, we have carefully modified the TM chalcogenides part in the introduction in the revised manuscript. The detailed revision is as follows:

Line 5-18, Page 3: "In view of the limited battery capacity and structural instability of sodium-graphite intercalation compounds, it's urgent to search high-capacity and long-life SIBs anode materials to meet the needs of large-scale energy storage applications^{3,4}. Thanks to their structural diversity and high theoretical capacity (about 400-600 mAh g⁻¹), transition metal chalcogenides (TMC) have great application potential in the anode of SIBs⁵. ZnSe, an important member of TMC, is an alloy-conversion combined anode material prospective for SIBs. It has decent electrical conductivity better than its oxide counterpart, and weak metal-selenium (Se) bond that facilitates the electrochemical conversion reactions⁶. But, it has some problems such as poorer electronic conductivity than carbon, sluggish kinetics and large volume change in the long-term charge-discharge process, which limits its rapid charge-discharge ability and structural stability. To solve these problems, various strategies have been developed, such as combining ZnSe with carbon-based materials, and constructing robust three-dimensional nanostructures."

Q7: The statements made about Te doping in the introduction need a reference or should be rephrased as there is no support for the claims made.

Our response: Thank you for your suggestions. We have rephrased the statement about Te doping in the introduction and added the references in the revised manuscript. The detailed revision is as follows:

Line 17-21, Page 4: "The electrical conductivity of Te is about 2×10^2 S cm⁻¹, which is much higher than that of Se (1×10^{-4} S cm⁻¹)¹⁹. In addition, Se and Te atoms are in the same main group, and the latter has a slightly larger radius than the former²⁰. This indicates that Te heteroatom doping of ZnSe may improve its electrical conductivity and potentially introduce some additional defects."

Page 25:

19. Zhang, J., Yin, Y.-X. & Guo, Y.-G. High-capacity Te anode confined in microporous carbon for long-life Na-ion batteries. *ACS Appl. Mater. Interfaces* **7**, 27838-27844 (2015).
20. Zhu, Y. et al. Te-doped Bi₂Se₃@NC nanocomposites for high-performance Li-ion battery anodes. *Sustainability* **15**, 16210 (2023).

Q8: The results of the new Te-doped material are summarized at the end of the introduction, but the results of a baseline ZnSe material should also be provided for comparison to show how significant the improve was.

Our response: We thank the reviewer for pointing this out. According to the reviewer's suggestions, we have provided the results of the baseline ZnSe material at the end of the introduction in the revised manuscript, which helps to show how significant the improvement is. The detailed revision is as follows:

Line 11-14, Page 5: "ZnSe_{0.7}Te_{0.3}@C electrode shows the significantly superior sodium-storage properties to pristine ZnSe@C electrode, including a higher capacity (5 A g⁻¹; 307 mAh g⁻¹ vs 118.8 mAh g⁻¹ after 1000 cycles), better rate performance (20 A g⁻¹; 256.2 vs 121.5 mAh g⁻¹)."

Q9: Were other twin-boundary inducing dopants besides Te tried for this material? How did the authors come to settle on Te for doping and on the specific composition that they chose. How are the authors sure that all of the improvents came from the twin boundary formation rather than alternative effects induced by the chemical or Na-storage properties of Te.

Our response: We appreciate the reviewer's professional questions.

ZnSe is a potential negative electrode material for sodium ion batteries. Since Se and Te are elements belonging to the same main group, their chemical properties are similar. Hence, it's facile to dope ZnSe with Te. In addition, Te has stronger metallic properties and better conductivity than Se, which is conducive to playing its inherent potential as a negative electrode material for sodium-ion batteries. The atomic radius of Te is slightly larger than that of Se, and the partial introduction of Te is easy to

cause internal defects in ZnSe, thus contributing to sodium storage performance. Therefore, we choose Te as the dopant. In terms of the synthesis, the precursor containing Zn was fully ground with Se and Te powder which then reacted at high temperature under Ar/H₂ mixed gas to become H₂Se and H₂Te. Finally, the Zn-containing precursor reacted with H₂Se and H₂Te at the same time to produce ZnSe_{0.7}Te_{0.3}@C. We also carried out a comparative experiment in which the Zn-containing precursor was mixed with Se powder and reacted under Ar/H₂ mixture at high temperature to produce ZnSe@C. Then, ZnSe@C was fully ground with Te powder and reacted under Ar/H₂ mixture at high temperature to obtain the sample, which was tested by TEM characterization. The results showed that no TBs were produced, as shown in Fig. R1. No twin boundary striations can be found in the figure. So, the synthesis of ZnSe_{0.7}Te_{0.3} with TBs required special experimental conditions.

Figure R1. TEM image of the sample.

In order to optimize the composition, we prepared three samples ZnSe_{0.8}Te_{0.2}@C,

ZnSe_{0.7}Te_{0.3}@C, and ZnSe_{0.5}Te_{0.5}@C by adjusting the usage amount of Se and Te powder during the synthesis. Then, we tested the sodium ion storage performance of samples with different components. By comparing their performances, we determined the specific composition. After Te doping, the electrochemical performance of ZnSe host can be enhanced compared with pristine ZnSe. The improvements come from the improved electrical conductivity by Te atom doping and the facilitated sodium ion diffusion dynamics related with twin structures also introduced by Te atom doping. Hence, the improvements come from the collective effect of Te atom doping, not just twin boundary formation. The defects of two dimensions (point defects and twin plane defects) in electrode materials play an important role in the improvement of battery performance. In a word, the cycling performance and rate capability of ZnSe_{0.7}Te_{0.3}@C are much better than that of ZnSe_{0.8}Te_{0.2}@C and ZnSe_{0.5}Te_{0.5}@C, indicating that ZnSe_{0.7}Te_{0.3}@C has the best number of twin boundaries and the best amount of Te atom doping. We have revised it in the revised manuscript. The detailed revision is as follows:

Page 13-15: “To consolidate the above theoretical analyses, different samples were controlled in terms of the amount of twin boundaries via tuning the doping level of tellurium. By changing the usage amount of Se and Te during the synthesis, ZnSe_{0.8}Te_{0.2}@C, ZnSe_{0.7}Te_{0.3}@C and ZnSe_{0.5}Te_{0.5}@C were prepared. Atomic percentages of C, N, O, Zn, Se and Te in samples of different compositions were shown by the energy spectrometer attached to the SEM, as shown in Supplementary Table 6. According to the component content in the energy spectrum, the atomic percentage of Se and Te in the sample can be determined, so as to obtain ZnSe_{0.8}Te_{0.2}@C, ZnSe_{0.7}Te_{0.3}@C and ZnSe_{0.5}Te_{0.5}@C. In order to determine the amount of twin boundaries in samples of different components, the samples were characterized by the technique of TEM. Supplementary Fig. 14a-14c show the low magnification TEM images of ZnSe_{0.8}Te_{0.2}@C, ZnSe_{0.7}Te_{0.3}@C and ZnSe_{0.5}Te_{0.5}@C, in which some of the grains have some fine stripes alternating between light and dark. When these streaks are enlarged, they have a typical twin structure, as shown in Supplementary Fig. 14d-14f. Therefore, the grains with twin boundary fringes of different compositions are analyzed statistically and quantitatively. The percentage of grains with twin boundary fringes in ZnSe_{0.8}Te_{0.2}@C, ZnSe_{0.7}Te_{0.3}@C and

ZnSe_{0.5}Te_{0.5}@C is 4.5%, 11.4% and 6.2%, respectively (Supplementary Fig. 14g). Compared with ZnSe_{0.8}Te_{0.2}@C and ZnSe_{0.5}Te_{0.5}@C, the grain twin boundary ratio of ZnSe_{0.7}Te_{0.3}@C is the largest, which may improve the performance the most.

Then, the rate performance of ZnSe@C, ZnSe_{0.8}Te_{0.2}@C, ZnSe_{0.7}Te_{0.3}@C and ZnSe_{0.5}Te_{0.5}@C is tested and shown in Fig. 5c. ZnSe_{0.7}Te_{0.3}@C is capable of releasing reversible specific capacities of 351.1, 333.3, 321.7, 310.2, 294.2, 277.2, 256.2 mAh g⁻¹ at current densities of 0.2, 0.5, 1, 2, 5, 10 and 20 A g⁻¹, respectively. For comparison, ZnSe@C is able to release reversible specific capacities of 245.0, 204.9, 184.5, 169.3, 149.4, 133.9, 121.5 mAh g⁻¹ at the same current density. ZnSe_{0.8}Te_{0.2}@C releases specific capacities of 293, 277.9, 264.7, 246.2, 224.8, 206.7, 185.3 mAh g⁻¹, respectively and ZnSe_{0.5}Te_{0.5}@C releases specific capacities of 298.2, 270.4, 246.9, 219.9, 191.7, 170, 152.1 mAh g⁻¹, respectively at the same current density. The rate performance of ZnSe_{0.7}Te_{0.3}@C, ZnSe_{0.8}Te_{0.2}@C and ZnSe_{0.5}Te_{0.5}@C is better than that of ZnSe@C, which is due to the collective effect of Te atom doping improving conductivity and twin structure improving sodium ion diffusion dynamics. The rate performance of ZnSe_{0.7}Te_{0.3}@C is better than that of ZnSe_{0.8}Te_{0.2}@C and ZnSe_{0.5}Te_{0.5}@C, indicating that ZnSe_{0.7}Te_{0.3}@C has the best number of TBs and the best amount of Te atom doping. The rate performance of ZnSe_{0.5}Te_{0.5}@C is slightly poorer than that of ZnSe_{0.8}Te_{0.2}@C, which is due to the excessive amount of Te atom doping. The charge and discharge curves of ZnSe_{0.7}Te_{0.3}@C nanocomposites at different current densities are shown in Fig. 5d. When the current density increases from 0.2 to 20 A g⁻¹, the voltage gap changes slightly between the charge and discharge voltage platforms, indicating the smaller polarization of reactions. Figure 5f compares the rate properties of zinc-based selenides and tellurides with those previously reported^{35-36, 38-42}, exhibiting much more excellent rate capability especially at higher rates. In order to study the contribution of amorphous carbon to capacity in the sample, the cycle and rate performance were tested, as shown in Supplementary Fig. 15. At a current density of 1 A g⁻¹, amorphous carbon has only a specific capacity of 32.3 mAh g⁻¹ after 1000 cycles (Supplementary Fig. 15a). Moreover, amorphous carbon releases reversible capacities of 58.2, 35.3, 24.8, 17.7, 12.2, 8.9, 11.5 mAh g⁻¹ at current densities of 0.2, 0.5, 1, 2, 5, 10 and 20 A g⁻¹ (Supplementary Fig. 15b). Obviously, it contributes little capacity in these hybrid nanocomposites.

Furthermore, Fig. 5e shows the correspondingly cycle performance at the current

density of 1 A g^{-1} . The first discharge capacities of $\text{ZnSe}_{0.7}\text{Te}_{0.3}@C$, $\text{ZnSe}_{0.8}\text{Te}_{0.2}@C$, $\text{ZnSe}_{0.5}\text{Te}_{0.5}@C$, and $\text{ZnSe}@C$ are 388.4, 382.0, 445.3, 303.4 mAh g^{-1} , respectively. After 800 cycles, the discharge capacity of $\text{ZnSe}_{0.7}\text{Te}_{0.3}@C$, $\text{ZnSe}_{0.8}\text{Te}_{0.2}@C$ and $\text{ZnSe}_{0.5}\text{Te}_{0.5}@C$ can retain 317.4, 272.5 and 240.3 mAh g^{-1} respectively, significantly higher than that of $\text{ZnSe}@C$ (191.0 mAh g^{-1}). This is because Te atom doping introduces twin plane defects, thereby increasing the active sites and improving the kinetics. Besides, Te atom doping also reduces the size of the nanoparticles, increases the specific surface area of the active material in contact with the electrolyte, and makes the nanoparticles fully react with sodium ions. Compared with $\text{ZnSe}_{0.8}\text{Te}_{0.2}@C$ and $\text{ZnSe}_{0.5}\text{Te}_{0.5}@C$, $\text{ZnSe}_{0.7}\text{Te}_{0.3}@C$ has the highest specific capacity, indicating that $\text{ZnSe}_{0.7}\text{Te}_{0.3}@C$ has the optimal Te doping level and number of TBs. The stability and capacity of batteries at high currents are still significant, so a high-current cycle performance test of $\text{ZnSe}_{0.7}\text{Te}_{0.3}@C$ and $\text{ZnSe}@C$ was carried out, as shown in Fig. 5g. It can be seen that both $\text{ZnSe}_{0.7}\text{Te}_{0.3}@C$ and $\text{ZnSe}@C$ have good cyclic stability. After 1000 cycles, the discharge capacity of $\text{ZnSe}_{0.7}\text{Te}_{0.3}@C$ remains 307 mAh g^{-1} , while that of $\text{ZnSe}@C$ is only 118.8 mAh g^{-1} . The excellent electrochemical performance of $\text{ZnSe}_{0.7}\text{Te}_{0.3}@C$ is attributed to the twin structures by Te atom doping, the optimal number of TBs and the improved conductivity. Hence, $\text{ZnSe}_{0.7}\text{Te}_{0.3}@C$ was determined to continue other electrochemistry testing.”

In the “Methods” part, the synthesis of these samples has been added:

Page 20: “0.2 g of ZIF-8@MF, 0.2 g of Te powder and 0.12 g of Se powder were ground evenly and then heated in Ar/H₂ mixed atmosphere at 800 °C for 2 h at a heating rate of 4 °C/min to obtain $\text{ZnSe}_{0.8}\text{Te}_{0.2}@C$. 0.2 g of ZIF-8@MF, 0.2 g of Te powder and 0.1 g of Se powder were ground evenly and then heated in Ar/H₂ mixed atmosphere at 800 °C for 2 h at a heating rate of 4 °C/min to obtain $\text{ZnSe}_{0.7}\text{Te}_{0.3}@C$. 0.2 g of ZIF-8@MF, 0.2 g of Te powder and 0.07 g of Se powder were ground evenly and then heated in Ar/H₂ mixed atmosphere at 800 °C for 2 h at a heating rate of 4 °C/min to obtain $\text{ZnSe}_{0.5}\text{Te}_{0.5}@C$.”

Fig. 5 | Electrochemical characterizations of the ZnSe@C, ZnSe_{0.5}Te_{0.5}@C, ZnSe_{0.8}Te_{0.2}@C and ZnSe_{0.7}Te_{0.3}@C nanocomposites. a CV curves and **b** galvanostatic charge and discharge curves at a current density of 0.2 A g⁻¹ of ZnSe_{0.7}Te_{0.3}@C. **c** Rate performance and **e** long-term cycling at the current density of 1 A g⁻¹ of ZnSe@C, ZnSe_{0.5}Te_{0.5}@C, ZnSe_{0.8}Te_{0.2}@C and ZnSe_{0.7}Te_{0.3}@C (1 A g⁻¹ = 1.6 C). **d** Charge and discharge curves of ZnSe_{0.7}Te_{0.3}@C at different current densities. **f** Comparison of the rate performance with those previously reported zinc-based selenides and tellurides^{35-36, 38-42}. **g** Long cycle performance at the current density of 5 A g⁻¹ of ZnSe@C and ZnSe_{0.7}Te_{0.3}@C.

Supplementary Fig. 14 TEM and HRTEM images of **a, d** ZnSe_{0.8}Te_{0.2}@C, **b, e** ZnSe_{0.7}Te_{0.3}@C and **c, f** ZnSe_{0.5}Te_{0.5}@C. **g** Percentage of the number of TBs in ZnSe_{0.8}Te_{0.2}@C, ZnSe_{0.7}Te_{0.3}@C and ZnSe_{0.5}Te_{0.5}@C.

Q10: The introduction does not do a good job of framing the significance of Na-ion batteries and how this work will improve our understanding towards designing better Na-ion anode materials. This would also provide context to the broader scientific community.

Our response: Thank you for your constructive suggestions. Accordingly, we have carefully revised the introduction to highlight the importance of the sodium-ion batteries and rephrased how this work will improve our understanding towards designing better Na-ion anode materials. The detailed revision is as follows:

Line 1-8, Page 3: “In recent years, sodium-ion batteries (SIBs) have made great progress, and are expected to be one of the best substitutes for widely commercialized lithium-ion batteries (LIBs), due to the abundant reserves of sodium resources and similar electrochemistry and operation mechanisms^{1,2}. As a typical anode material for LIBs, graphite has been widely used. In view of the limited battery capacity and structural instability of sodium-graphite intercalation compounds, it’s urgent to search high-capacity and long-life SIBs anode materials to meet the needs of large-scale energy storage applications^{3,4}.”

Line 16-19, Page 5: “When designing the anode materials of SIBs, the defects of two dimensions are introduced at the same time, which will overcome their shortcomings from different aspects, and finally realize the comprehensive and significant improvement of sodium storage performance”.

Materials Synthesis and Characterization

Q11: The conclusions drawn from the HAADF-STEM showing the low contrast of the carbon structure is not immediately clear from Fig. 1d, further explanation on how to interpret the image to come to this conclusion should be provided.

Our response: Many thanks for the comments. As you know, in HAADF-STEM images, the contrast intensity is approximately in proportion to the square of the atomic numbers. Thus, the contrast of C is much lower than the metal elements here. In the revised image in Figure R2, we slightly adjust the contrast of the HAADF-STEM image. The C now is immediately clear, but still lower than that of the nanoparticles. Furthermore, from the C elemental mapping, we can distinguish the distribution of the C.

Figure R2 HAADF-STEM image of $\text{ZnSe}_{0.7}\text{Te}_{0.3}@C$.

Q12: Would adding in additional Te beyond the $\text{Se}_{0.7}\text{Te}_{0.3}$ composition induce more twin boundaries and further improve performance? This point ties back to how was this specific composition decided, were a series of materials tried? Were there some calculations run that suggested that this composition would induce the optimal amount of twin boundaries?

Our response: We appreciate the reviewer's professional suggestion.

In order to optimize the composition, we prepared three samples $\text{ZnSe}_{0.8}\text{Te}_{0.2}@C$, $\text{ZnSe}_{0.7}\text{Te}_{0.3}@C$, and $\text{ZnSe}_{0.5}\text{Te}_{0.5}@C$ by adjusting the usage amount of Se and Te powder during the synthesis. Then, we tested the sodium ion storage performance of samples with different components. To determine the number of TBs in samples of different components, the samples were characterized by low and high magnification TEM, as shown in Supplementary Fig. 14a-14f. We quantitatively analyzed the number of twin boundaries in samples of different compositions, as shown in

Supplementary Fig. 14g. In order to theoretically verify the effect of the number of TBs on the performance, the model of $\text{ZnSe}_{0.7}\text{Te}_{0.3}$ with two TBs (Supplementary Fig. 12) is constructed and the sodium ion diffusion energy barrier (Fig. 4i) is calculated. The detailed revision is as follows:

Line 9-16, Page 12: “In order to theoretically investigate the effect of the number of TBs on the performance, the model of $\text{ZnSe}_{0.7}\text{Te}_{0.3}$ with two TBs is constructed (Supplementary Fig. 12). And the sodium ion diffusion energy barrier is calculated, as shown in Fig. 4i. Obviously, the Na^+ diffusion energy barrier across the TBs is 0.39 eV for $\text{ZnSe}_{0.7}\text{Te}_{0.3}$ with two TBs, much lower than that of the $\text{ZnSe}_{0.7}\text{Te}_{0.3}$ with one TB (0.70 eV). The calculation results show that with the increase of the number of TBs, the sodium diffusion energy barrier is greatly reduced, which helps the kinetics of sodium ion reactions.”

Page 13-15: “To consolidate the above theoretical analyses, different samples were controlled in terms of the amount of twin boundaries via tuning the doping level of tellurium. By changing the usage amount of Se and Te during the synthesis, $\text{ZnSe}_{0.8}\text{Te}_{0.2}@C$, $\text{ZnSe}_{0.7}\text{Te}_{0.3}@C$ and $\text{ZnSe}_{0.5}\text{Te}_{0.5}@C$ were prepared. Atomic percentages of C, N, O, Zn, Se and Te in samples of different compositions were shown by the energy spectrometer attached to the SEM, as shown in Supplementary Table 6. According to the component content in the energy spectrum, the atomic percentage of Se and Te in the sample can be determined, so as to obtain $\text{ZnSe}_{0.8}\text{Te}_{0.2}@C$, $\text{ZnSe}_{0.7}\text{Te}_{0.3}@C$ and $\text{ZnSe}_{0.5}\text{Te}_{0.5}@C$. In order to determine the amount of twin boundaries in samples of different components, the samples were characterized by the technique of TEM. Supplementary Fig. 14a-14c show the low magnification TEM images of $\text{ZnSe}_{0.8}\text{Te}_{0.2}@C$, $\text{ZnSe}_{0.7}\text{Te}_{0.3}@C$ and $\text{ZnSe}_{0.5}\text{Te}_{0.5}@C$, in which some of the grains have some fine stripes alternating between light and dark. When these streaks are enlarged, they have a typical twin structure, as shown in Supplementary Fig. 14d-14f. Therefore, the grains with twin boundary fringes of different compositions are analyzed statistically and quantitatively. The percentage of grains with twin boundary fringes in $\text{ZnSe}_{0.8}\text{Te}_{0.2}@C$, $\text{ZnSe}_{0.7}\text{Te}_{0.3}@C$ and $\text{ZnSe}_{0.5}\text{Te}_{0.5}@C$ is 4.5%, 11.4% and 6.2%, respectively (Supplementary Fig. 14g). Compared with $\text{ZnSe}_{0.8}\text{Te}_{0.2}@C$ and $\text{ZnSe}_{0.5}\text{Te}_{0.5}@C$, the grain twin boundary ratio of $\text{ZnSe}_{0.7}\text{Te}_{0.3}@C$ is the largest, which may improve the performance the most.

Then, the rate performance of ZnSe@C, ZnSe_{0.8}Te_{0.2}@C, ZnSe_{0.7}Te_{0.3}@C and ZnSe_{0.5}Te_{0.5}@C is tested and shown in Fig. 5c. ZnSe_{0.7}Te_{0.3}@C is capable of releasing reversible specific capacities of 351.1, 333.3, 321.7, 310.2, 294.2, 277.2, 256.2 mAh g⁻¹ at current densities of 0.2, 0.5, 1, 2, 5, 10 and 20 A g⁻¹, respectively. For comparison, ZnSe@C is able to release reversible specific capacities of 245.0, 204.9, 184.5, 169.3, 149.4, 133.9, 121.5 mAh g⁻¹ at the same current density. ZnSe_{0.8}Te_{0.2}@C releases specific capacities of 293, 277.9, 264.7, 246.2, 224.8, 206.7, 185.3 mAh g⁻¹, respectively and ZnSe_{0.5}Te_{0.5}@C releases specific capacities of 298.2, 270.4, 246.9, 219.9, 191.7, 170, 152.1 mAh g⁻¹, respectively at the same current density. The rate performance of ZnSe_{0.7}Te_{0.3}@C, ZnSe_{0.8}Te_{0.2}@C and ZnSe_{0.5}Te_{0.5}@C is better than that of ZnSe@C, which is due to the collective effect of Te atom doping improving conductivity and twin structure improving sodium ion diffusion dynamics. The rate performance of ZnSe_{0.7}Te_{0.3}@C is better than that of ZnSe_{0.8}Te_{0.2}@C and ZnSe_{0.5}Te_{0.5}@C, indicating that ZnSe_{0.7}Te_{0.3}@C has the best number of TBs and the best amount of Te atom doping. The rate performance of ZnSe_{0.5}Te_{0.5}@C is slightly poorer than that of ZnSe_{0.8}Te_{0.2}@C, which is due to the excessive amount of Te atom doping. The charge and discharge curves of ZnSe_{0.7}Te_{0.3}@C nanocomposites at different current densities are shown in Fig. 5d. When the current density increases from 0.2 to 20 A g⁻¹, the voltage gap changes slightly between the charge and discharge voltage platforms, indicating the smaller polarization of reactions. Figure 5f compares the rate properties of zinc-based selenides and tellurides with those previously reported^{35-36, 38-42}, exhibiting much more excellent rate capability especially at higher rates. In order to study the contribution of amorphous carbon to capacity in the sample, the cycle and rate performance were tested, as shown in Supplementary Fig. 15. At a current density of 1 A g⁻¹, amorphous carbon has only a specific capacity of 32.3 mAh g⁻¹ after 1000 cycles (Supplementary Fig. 15a). Moreover, amorphous carbon releases reversible capacities of 58.2, 35.3, 24.8, 17.7, 12.2, 8.9, 11.5 mAh g⁻¹ at current densities of 0.2, 0.5, 1, 2, 5, 10 and 20 A g⁻¹ (Supplementary Fig. 15b). Obviously, it contributes little capacity in these hybrid nanocomposites.

Furthermore, Fig. 5e shows the correspondingly cycle performance at the current density of 1 A g⁻¹. The first discharge capacities of ZnSe_{0.7}Te_{0.3}@C, ZnSe_{0.8}Te_{0.2}@C, ZnSe_{0.5}Te_{0.5}@C, and ZnSe@C are 388.4, 382.0, 445.3, 303.4 mAh g⁻¹, respectively. After 800 cycles, the discharge capacity of ZnSe_{0.7}Te_{0.3}@C, ZnSe_{0.8}Te_{0.2}@C and

ZnSe_{0.5}Te_{0.5}@C can retain 317.4, 272.5 and 240.3 mAh g⁻¹ respectively, significantly higher than that of ZnSe@C (191.0 mAh g⁻¹). This is because Te atom doping introduces twin plane defects, thereby increasing the active sites and improving the kinetics. Besides, Te atom doping also reduces the size of the nanoparticles, increases the specific surface area of the active material in contact with the electrolyte, and makes the nanoparticles fully react with sodium ions. Compared with ZnSe_{0.8}Te_{0.2}@C and ZnSe_{0.5}Te_{0.5}@C, ZnSe_{0.7}Te_{0.3}@C has the highest specific capacity, indicating that ZnSe_{0.7}Te_{0.3}@C has the optimal Te doping level and number of TBs. The stability and capacity of batteries at high currents are still significant, so a high-current cycle performance test of ZnSe_{0.7}Te_{0.3}@C and ZnSe@C was carried out, as shown in Fig. 5g. It can be seen that both ZnSe_{0.7}Te_{0.3}@C and ZnSe@C have good cyclic stability. After 1000 cycles, the discharge capacity of ZnSe_{0.7}Te_{0.3}@C remains 307 mAh g⁻¹, while that of ZnSe@C is only 118.8 mAh g⁻¹. The excellent electrochemical performance of ZnSe_{0.7}Te_{0.3}@C is attributed to the twin structures by Te atom doping, the optimal number of TBs and the improved conductivity. Hence, ZnSe_{0.7}Te_{0.3}@C was determined to continue other electrochemistry testing.”

In the “Methods” part, the synthesis of these samples has been added:

Page 20: “0.2 g of ZIF-8@MF, 0.2 g of Te powder and 0.12 g of Se powder were ground evenly and then heated in Ar/H₂ mixed atmosphere at 800 °C for 2 h at a heating rate of 4 °C/min to obtain ZnSe_{0.8}Te_{0.2}@C. 0.2 g of ZIF-8@MF, 0.2 g of Te powder and 0.1 g of Se powder were ground evenly and then heated in Ar/H₂ mixed atmosphere at 800 °C for 2 h at a heating rate of 4 °C/min to obtain ZnSe_{0.7}Te_{0.3}@C. 0.2 g of ZIF-8@MF, 0.2 g of Te powder and 0.07 g of Se powder were ground evenly and then heated in Ar/H₂ mixed atmosphere at 800 °C for 2 h at a heating rate of 4 °C/min to obtain ZnSe_{0.5}Te_{0.5}@C”.

Fig. 4i The energy barrier of sodium ion for $\text{ZnSe}_{0.7}\text{Te}_{0.3}$ with two TBs across the TB.

Supplementary Fig. 12 Model of $\text{ZnSe}_{0.7}\text{Te}_{0.3}$ with two TBs.

Fig. 5 | Electrochemical characterizations of the ZnSe@C , $\text{ZnSe}_{0.5}\text{Te}_{0.5}@C$, $\text{ZnSe}_{0.8}\text{Te}_{0.2}@C$ and $\text{ZnSe}_{0.7}\text{Te}_{0.3}@C$ nanocomposites. a CV curves and **b** galvanostatic charge and discharge curves at a current density of 0.2 A g^{-1} of $\text{ZnSe}_{0.7}\text{Te}_{0.3}@C$. **c** Rate performance and **e** long-term cycling at the current density of

1 A g⁻¹ of ZnSe@C, ZnSe_{0.5}Te_{0.5}@C, ZnSe_{0.8}Te_{0.2}@C and ZnSe_{0.7}Te_{0.3}@C (1 A g⁻¹ = 1.6 C). **d** Charge and discharge curves of ZnSe_{0.7}Te_{0.3}@C at different current densities. **f** Comparison of the rate performance with those previously reported zinc-based selenides and tellurides^{35-36, 38-42}. **g** Long cycle performance at the current density of 5 A g⁻¹ of ZnSe@C and ZnSe_{0.7}Te_{0.3}@C.

Supplementary Fig. 14 TEM and HRTEM images of **a, d** ZnSe_{0.8}Te_{0.2}@C, **b, e** ZnSe_{0.7}Te_{0.3}@C and **c, f** ZnSe_{0.5}Te_{0.5}@C. **g** Percentage of the number of TBs in ZnSe_{0.8}Te_{0.2}@C, ZnSe_{0.7}Te_{0.3}@C and ZnSe_{0.5}Te_{0.5}@C.

Q13: Were base ZnSe and ZnSe_{0.7}Te_{0.3} tried without the carbon nanostructuring to remove the possibility of some synergistic effects?

Our response: Thanks for your good question.

Firstly, ZIF-8 is used as the template in the preparation of ZnSe_{0.7}Te_{0.3}@C material. After high temperature calcination, an interconnected carbon network structure is formed, which makes the nanoparticles evenly dispersed in the carbon matrix. We conducted electrochemical performance tests on the pure carbon structure, and the test results showed that the carbon structure contributed a very low capacity, as shown in Supplementary Fig. 15. The effect of Te heteroatom doping and twin structure on electrochemical properties is not affected by carbon structure. The carbon material can work well as the matrix for dispersing active substances and avoiding the aggregation. The presence of carbon matrix can give full play to the electrochemical properties of electrode materials. Actually, there are many similar researchers to make use of carbon matrix to disperse active materials and then do the targeted investigations. For example, Se-CoS₂/CoS_{1.035}@C composites enriched with anionic vacancies are successfully fabricated via one-step doping of Se onto CoS₂/CoS_{1.035}@C heterojunction by Bo Wen et al. The rod-like structure of amorphous carbon is used as the carrier (*J. Colloid Interface Sci.*, **2024**, 675, 980-988). Yongjin Fang et al. achieved Cu₂S@carbon@MoS₂ structure by growing ultra-thin MoS₂ nanosheets on nitrogen-doped carbon-coated Cu₂S nanocapsules through a simple multi-step template strategy. The carbon nanoboxes play a role in increasing the material's electrical conductivity and loading MoS₂ nanosheets (*Angew. Chem. Int. Ed.* **2020**, 59, 7178-7183).

Supplementary Fig. 15 a Cycle performance and **b** rate performance of amorphous carbon.

Q14: Were different types of Te orderings tested, such as Te clusters, in the DFT calculations to see which would be the lowest energy? From the manuscript it looks like only two Te orderings were tried, but were uniform.

Our response: We appreciate the reviewer's for the significant comments. According to the reviewer's suggestions, we have calculated three structural models, including one Te replacing one Se atom, three Te replacing three Se atoms, and Te atoms occupying interstitial sites in ZnSe. And we then calculated the corresponding phonon spectra, average energy per atom in the final state, and defect formation energy. These revisions have added in the revised manuscript. The detailed revision is as follows:

Page 7-8: "In order to further verify theoretically whether Te replaces Se atoms or occupies interstitial sites in ZnSe, we constructed the structural models of different configurations in Fig. 3a–3c and then calculated the corresponding phonon spectra, average energy per atom in the final state, and defect formation energy. From Supplementary Fig. 6a and 6b, it can be seen that the phonon spectra of sub 1 model with one Te atom replacing one Se atom and sub 3 model with three Te atoms replacing three Se atoms have no virtual phonon mode, indicating that the models are mechanically and dynamically stable. However, the phonon spectrum for int model in which Te occupies interstitial position has a tiny virtual phonon mode, proving that its structure is somewhat unstable, as shown in Supplementary Fig. 6c. This may be because larger Te occupies interstitial positions causing gigantic structural distortion, profoundly affecting the normal arrangement of surrounding atoms. The average energy of each atom in the final states of these different configurations is negative, which indicates that they are thermodynamically stable (Supplementary Fig. 6d). The defect formation energy of sub 1 model is 0.76 eV, which is much lower than that of sub 3 and int models (2.23 and 3.31 eV), as shown in Fig. 3f."

Page 23: "The defect formation energy of structural models was also calculated with the chemical potential of the components and the calculation formula was as follows:

$$\Delta H_D = E_D - E_h + \sum n_i \mu_i$$

where ΔH_D is the defect formation energy, E_D is the total energy of the supercell with

the defect (D), E_h is the total energy of the ZnSe supercell, μ_i is the chemical potentials of element Te and Se. n_i is the numbers of Te and Se atoms that were removed or added to form the system.

$$E_{\text{per atom}} = E_{\text{total}}/N_{\text{atom}}$$

$E_{\text{per atom}}$ is the average energy per atom in the final state, E_{total} is the total energy of the structural model, and N_{atom} is the total number of atoms in the structural model.”

Fig. 3 | Calculations of defect formation energy and average energy per atom in the final state. a A phase model in which one Te atom replaces one Se atom (sub 1). **b** A phase model in which three Te atoms replace three Se atoms (sub 3). **c** A phase model with one Te atom occupying interstitial position (int). Interface models of ZnSe_{0.7}Te_{0.3} **(d)** with TBs and **(e)** without TBs. **f** Comparison diagram of defect

formation energy of three models of sub 1, sub 3 and int. **g** Comparison diagram of average energy per atom in the final state of interface models.

Supplementary Fig. 6 Phonon spectra for the structural models of (a) sub 1, (b) sub 3 and (c) int. **d** The average energy per atom in the final state of different configuration structures.

Q15: In Figure 3, it does not look like there is any Se vacancy to account for Te in the interstitial sites, this would not properly charge balance and possibly cause the formation energy produced by the calculation to be higher than it should be as there would be additional atoms in the unit cell that should not be there.

Our response: Thanks for your consideration. When Te occupies the interstitial position of ZnSe, Te is initially added in the form of atom. So the total charge is neutral. In most of the calculations research, the doped species are handled with atoms, not ions. For example, Shiwen Wang et al. prepared an interstitial boron-doped

tunnel-type VO₂(B) by a simple hydrothermal method. B is located in the interstitial position of the tunnel-type VO₂, and the surrounding oxygen atoms are not missing (*Carbon Energy*, **2023**, 5, e330), as shown in Fig. R3a. Xinyu Ping et al. inserted Si atoms into the interstitial position of RuO₂ lattice to construct stable RuO₂ for acidic oxygen evolution reaction. While Si atoms occupy interstitial positions, there is no absence of Ru atoms around them (*Nat. Commun.*, **2024**, 15, 2501), as shown in Fig. R3b.

Figure R3. **a** Referenced B doped VO₂ materials in reported literature (*Carbon Energy*, **2023**, 5, e330). **b** Referenced Si doped RuO₂ materials in reported literature (*Nat. Commun.*, **2024**, 15, 2501).

Q16: Why is the ZnSe_{0.7}Te_{0.3} material with twin boundaries showing formation energy of -2.8eV when it is stated as 1.09 eV earlier in the same paragraph. It is not clear what is different about these two calculations that would merit the

same material showing two different formation energies.

Our response: Thanks for your good question. It may be that we have not written enough details in this part, which is easy to confuse readers. Firstly, we construct the phase structure model of three configurations (sub 1, sub 3, int) without twin structure based on ZnSe crystal, and calculate the defect formation energy. The defect formation energy of one Se atom position replaced by one Te atom is 0.76 eV, which is much lower than that of three Se atom positions replaced by three Te atoms and one Te atom occupying interstitial positions (2.23 eV, 3.31 eV). Then, based on the HAADF-STEM imaging results, the atomic interface structure models of ZnSe_{0.7}Te_{0.3} with and without TBs are established. The average energy per atom in the final state of the ZnSe_{0.7}Te_{0.3} with TBs is ~ -2.99 eV, which is lower than that of ZnSe_{0.7}Te_{0.3} without TBs (-2.8 eV), indicating that ZnSe_{0.7}Te_{0.3} with twin structure is more stable thermodynamically. These revisions have added in the revised manuscript. The detailed revision is as follows:

Line 12-24, Page 8: “The defect formation energy of sub 1 model is 0.76 eV, which is much lower than that of sub 3 and int models (2.23 and 3.31 eV), as shown in Fig. 3f. This indicates that Te tends to replace Se atoms rather than occupy interstitial positions, which is consistent with the above atom mapping results. Figure 3d and 3e show the atomic interface structure models of ZnSe_{0.7}Te_{0.3} with TBs and ZnSe_{0.7}Te_{0.3} without TBs based on the HAADF-STEM imaging results. The total number of atoms of ZnSe_{0.7}Te_{0.3} models with TBs and without TBs is the same, and the ratio of Se to Te atoms is about 7:3. The average energy per atom in the final state of the ZnSe_{0.7}Te_{0.3} with TBs is ~ -2.99 eV, which is lower than that of ZnSe_{0.7}Te_{0.3} without TBs (-2.8 eV), indicating that ZnSe_{0.7}Te_{0.3} with twin structure is more stable thermodynamically, as shown in Fig. 3g. In addition, the defect formation energy of ZnSe_{0.7}Te_{0.3} with TBs is relatively small (0.86 eV), indicating that the twin structure is easy to form.”

Q17: For the X-ray diffraction data, the analysis is too general and qualitative. LeBail or Rietveld refinement can be performed to show the change in the unit

cell from the undoped and doped material and scherrer analysis can be performed to indicate difference in particle sizes. Just stating that the peaks moved to lower angles is insufficient because that can also be caused from a difference in sample height in the instrument itself.

Our response: We appreciate the reviewer's suggestions. According to your suggestion, Rietveld refinement and Scherrer analysis were performed for undoped and doped materials. They have been added in the revised manuscript. The detailed revision is as follows:

Line 4-21, Page 9: "The crystal phase of the samples was characterized by X-ray diffraction (XRD) measurements, as shown in Supplementary Fig. 7. Both ZnSe@C and ZnSe_{0.7}Te_{0.3}@C show a set of diffraction peaks notably consistent with the face-centered cubic ZnSe with space group F-43m. No impurity peaks were detected in either sample, and sharp Bragg peaks indicated good crystallinity in both ZnSe@C and ZnSe_{0.7}Te_{0.3}@C. Accurate structural information of ZnSe@C and ZnSe_{0.7}Te_{0.3}@C is obtained through Rietveld refinement, and the results are listed in Supplementary Table 1-3. It can be seen from the refinement results that the cell parameters and cell volume of ZnSe_{0.7}Te_{0.3}@C are larger than that of ZnSe@C, indicating that Te atoms are successfully doped into ZnSe (Supplementary Table 3). The occupancy of Se and Te atoms in ZnSe_{0.7}Te_{0.3}@C was 0.029 and 0.013, respectively, which was consistent with the atomic percentage of Se and Te in XPS results (Supplementary Table 2). In addition, compared with ZnSe@C, the diffraction peaks of ZnSe_{0.7}Te_{0.3}@C have different degrees of deviation towards small angles, which was caused by Te heteroatoms successfully doped into ZnSe crystal lattices. The average crystallite size of ZnSe@C and ZnSe_{0.7}Te_{0.3}@C is calculated using the Scherrer's method to be 43.4 nm and 17.8 nm, respectively, which proves that the nanoparticle size of ZnSe_{0.7}Te_{0.3}@C is smaller (Supplementary Table 4).".

Supplementary Fig. 7 Rietveld refinement for powder XRD patterns of **a** ZnSe@C and **b** ZnSe_{0.7}Te_{0.3}@C materials.

Supplementary Table 1. Atomic parameters of Rietveld refinement for powder XRD pattern of ZnSe@C.

Atom	x	y	z	Occupancy
Zn1	0.00000	0.00000	0.00000	0.042
Se1	0.25000	0.25000	0.25000	0.042

Supplementary Table 2. Atomic parameters of Rietveld refinement for powder XRD pattern of ZnSe_{0.7}Te_{0.3}@C.

Atom	x	y	z	Occupancy
Zn1	0.00000	0.00000	0.00000	0.042
Se1	0.25000	0.25000	0.25000	0.029

Te1	0.25000	0.25000	0.25000	0.013
-----	---------	---------	---------	-------

Supplementary Table 3. Cell parameters of Rietveld refinement for powder XRD patterns of ZnSe@C and ZnSe_{0.7}Te_{0.3}@C.

Sample	a/b/c (Å)	α/β/γ (°)	V(Å ³)
ZnSe@C	5.66911	90.00000	182.1984
ZnSe _{0.7} Te _{0.3} @C	5.69440	90.00000	184.6477

Supplementary Table 4. The crystallite size of ZnSe@C and ZnSe_{0.7}Te_{0.3}@C from XRD characterization according to Scherrer's relation.

ZnSe@C		ZnSe _{0.7} Te _{0.3} @C	
2θ	particle size (nm)	2θ	particle size (nm)
27.21	43.50	27.15	25.70
45.24	42.30	45.04	18.50
53.60	44.30	53.36	16.30
65.93	44.60	65.56	14.90
72.74	42.50	72.28	13.80

Note: Scherrer's formula is used to calculate the size of particles⁵.

$$D = \frac{K\lambda}{\beta \cos\theta}$$

where D represents the size of the grain, K is constant, λ is the X-ray wavelength, β is the full width of the diffraction peak at half maxima, θ represents the diffraction angle.

Q18: All references should be added after the period, to avoid confusion with exponents.

Our response: Thanks for your careful examination. References that are easily confused with the index have been revised.

Line 12-14, Page 10: “The Zn 2p XPS spectrum (Supplementary Fig. 8f) contains two main peaks at 1044.9 and 1021.9 eV, respectively coming from Zn 2p_{1/2} and Zn 2p_{2/3} orbitals, a token of bivalent Zn²⁺”.

Q19: The XPS results indicate the formation of TeO₂, but if there is a significant amount of TeO₂, then this could cause a significant deviation in the sample composition. How was the sample composition verified?

Our response: Thanks for your good questions. The XPS results did show the presence of Te-O bond. The XPS detection depth is very shallow, about a few nanometers, and the sample is porous with a large specific surface area. Therefore, in the inevitable contact process with the air, an oxide film will be formed on the surface, which has a certain impact on the XPS results. However, in fact, during the preparation of the sample, the heat treatment is carried out in Ar/H₂ mixed atmosphere, which can prevent oxidation. XRD detects the entire phase composition. According to the results of XRD, there is no TeO₂ generation. In addition, no oxygen atom was found to enter the twin structure in the atomic mapping results of spherical aberration electron microscopy, which is consistent with XRD results. By SEM energy spectrum analysis, the contents of C, N, O, Zn, Se and Te in ZnSe_{0.7}Te_{0.3}@C were 46.94%, 2.08%, 1.54%, 22.05%, 15.59% and 11.80%, respectively. So the content of oxygen in the sample is small (1.54%) and is attributed to the presence of the C=O bond. Furthermore, by referring to ZnTe based materials reported in literature (such as, Adv. Energy Mater. 2022, 12, 2203118; Chem. Eur. J. 2023, 29, e202203339; Small 2023, 19, 2304916; Adv. Funct. Mater. 2021, 31, 2006425), XPS results also show strong Te-O bond (Fig. R4).

Figure R4. Referenced XPS results of ZnTe based materials in reported literature (Adv. Energy Mater. 2022, 12, 2203118; Chem. Eur. J. 2023, 29, e202203339; Small 2023, 19, 2304916; Adv. Funct. Mater. 2021, 31, 2006425).

Q20: The TGA results show that there is a lot of carbon in the samples - is the carbon also contributing to the specific capacity of these materials?

Our response: We appreciate the reviewer’s professional suggestion. In fact, there is a lot of amorphous carbon in the samples. According to your suggestion, the ZnSe@C was soaked and ultrasonic treated with aqua regia, and then centrifugally washed with ultra-pure water and dried to obtain amorphous carbon. Amorphous carbon was used as the active material to prepare the electrode and then assembled into batteries for cycle and rate performance testing. The test results are shown in Supplementary Fig. 15, and the contribution capacity of amorphous carbon is very low. They have been added in the revised manuscript. The detailed revision is as follows:

Page 14-15: “In order to study the contribution of amorphous carbon to capacity in the sample, the cycle and rate performance were tested, as shown in Supplementary Fig.

15. At a current density of 1 A g^{-1} , amorphous carbon has a specific capacity of 32.3 mAh g^{-1} after 1000 cycles (Supplementary Fig. 15a). Moreover, amorphous carbon releases reversible capacities of 58.2, 35.3, 24.8, 17.7, 12.2, 8.9, 11.5 mAh g^{-1} at current densities of 0.2, 0.5, 1, 2, 5, 10 and 20 A g^{-1} (Supplementary Fig. 15b). Obviously, it contributes little capacity in these hybrid nanocomposites.”

In the “Methods” part, the synthesis of the sample has been added:

Page 20-21: “The black suspension was obtained by fully soaking $\text{ZnSe}_{0.7}\text{Te}_{0.3}@C$ in aqua regia and ultrasonic treatment. Afterwards, the solution was extracted and filtered, washed with ultra-pure water and dried at $60 \text{ }^\circ\text{C}$ to prepare amorphous carbon.”

Supplementary Fig. 15 a Cycle performance and **b** rate capability of amorphous carbon.

Q21: The Te-doped GITT curve looks much shorter than the undoped samples, were the material loadings different? If not it would seem the specific capacity of the doped material is much lower than the undoped material, which contradicts what is stated later in the section on the electrochemical results.

Our response: Thanks for your professional questions. In fact, the material loadings of $\text{ZnSe}@C$ and $\text{ZnSe}_{0.7}\text{Te}_{0.3}@C$ samples are indeed quite different. In order to make the results more accurate, the $\text{ZnSe}@C$ and $\text{ZnSe}_{0.7}\text{Te}_{0.3}@C$ samples were re-tested with electrodes with little difference in active material loading. GITT curves of the charge/discharge process of $\text{ZnSe}@C$ and $\text{ZnSe}_{0.7}\text{Te}_{0.3}@C$ were showed in Supplementary Fig. 11. The diffusion rate of sodium ions during charging and discharging calculated by GITT were showed in Fig. 4g and 4h.

Supplementary Fig. 11 GITT curves of **a** ZnSe@C and **b** ZnSe_{0.7}Te_{0.3}@C.

Fig. 4 The diffusion rate of sodium ions during **g** charging and **h** discharging calculated by GITT.

Q22: The diffusion coefficients primarily look lower for the doped material in the 0.5-2.0V, has the conversion reaction already occurred by this point? How much sodium is in the material within this potential range?

Our response: Thanks for your professional questions. The diffusion coefficient of sodium ions of ZnSe_{0.7}Te_{0.3}@C is low in the voltage range of 0.65-0.9 V and the conversion reaction indeed occurred by this point during the discharge process. During the discharge process, the content of sodium in the material is 37.5% in the range of 0.65-0.9 V potential. As can be seen from Fig. 4g, during the charging process, ZnSe_{0.7}Te_{0.3}@C has a lower diffusion coefficient of sodium ion in the voltage range of 0.9-1.2 and 1.9-2.5V. The conversion reaction occurred in the voltage range of 0.9-1.2V during the charge process, and the content of sodium in the material is 41.2%. The content of sodium in the material is 19.6% in the range of 1.9-2.5V

potential.

Q23: For the band gap calculations, the DOS diagrams in the supporting information show states that are < 1eV for both materials which does not match what is shown in Figure 4i - can the authors comment on this discrepancy.

Our response: We thank the reviewer for pointing it out. We have previously calculated the band gap of ZnSe and ZnSe_{0.7}Te_{0.3} using band structure and DOS diagram as shown in Fig. R5. Actually, the band structure diagrams and DOS diagram has the same statement as Fig. 4i in the previous manuscript. We adjusted the valence band top in the previous dos diagram to 0, as shown in Fig. R5c and R5d. Because there is a little broadening when calculating the DOS diagram, the band gap in the DOS diagram is smaller than that in the band structure diagram. In order not to confuse the readers, we have removed the DOS diagrams from the revised supporting information and the corresponding discussion in the revised manuscript.

Figure R5. Band structure of **a** ZnSe_{0.7}Te_{0.3} and **b** ZnSe. DOS diagrams of **c** ZnSe

and **d** ZnSe_{0.7}Te_{0.3}.

Q24: The statement that Te improves the electrical conductivity of the material should be removed as just because a material has a smaller bandgap that does not mean it has superior electronic conductivity, it just means the energy required for the material to conduct electrons is lower.

Our response: We appreciate the reviewer's suggestions. According to the reviewer's suggestions, we have revised the similar statement in the revised manuscript. The revision is as follows:

In the revised manuscript, we have removed the related discussion with "The band gap of ZnSe_{0.7}Te_{0.3} is 0.956 eV, which is lower than that of ZnSe (1.109 eV) in Fig. 4i, indicating that Te atom doping improves the electronic conductivity."

Q25: The authors do not provide an equivalent circuit for how they interpreted their impedance spectra. An equivalent circuit should be provided so the readers have context on how the authors are interpreting their data.

Our response: We appreciate the reviewer's suggestions. We have added the equivalent circuit for impedance spectra in the revised manuscript and supporting information. The detailed revision is as follows:

Page 12-13: "To further investigate the kinetics of ZnSe@C and ZnSe_{0.7}Te_{0.3}@C, electrochemical impedance spectroscopy (EIS) tests were performed. The Nyquist plots are fitting by an equivalent circuit model shown in Supplementary Fig. 13a and the obtained values of resistance are listed in Supplementary Table 5. As shown in Supplementary Fig. 13a, the R_{ct} of ZnSe_{0.7}Te_{0.3}@C is about only one third of that of ZnSe@C, implying ZnSe_{0.7}Te_{0.3}@C has a higher charge transfer rate. The $Z' - \omega^{-1/2}$ plot derives from the EIS spectra in Supplementary Fig. 13b and the slope called Warburg coefficient σ is related to the diffusion of sodium ions in the electrode materials. Obviously, ZnSe_{0.7}Te_{0.3}@C has a slope of 26.3 much lower than that of ZnSe@C (80.4), further demonstrating the faster sodium ion diffusion of ZnSe_{0.7}Te_{0.3}@C."

Supplementary Information, Page 14-15: "In the fitted equivalent circuit, R_e

represents the ohmic resistance in the cell system, which is related to the semicircular intercept of the Z' axis. CPE1 represents a double layer between the electrode and the electrolyte. R_{ct} is the charge transfer resistance, which represents the resistance of charge transfer at the interface between the electrode and the electrolyte. ”

Supplementary Fig. 13 a EIS spectra and equivalent circuit model of ZnSe@C and ZnSe_{0.7}Te_{0.3}@C (line: fitted data; dot: pristine data). b The corresponding Z' - $\omega^{-1/2}$ plots.

Supplementary Table 5. Equivalent circuit fitting value of ZnSe@C and ZnSe_{0.7}Te_{0.3}@C samples.

Sample	ZnSe@C	ZnSe _{0.7} Te _{0.3} @C
R_e	4.61	3.51
R_{ct}	68.26	21.83

Q26: More information on how the CV peaks are being assigned should be provided. The reference listed (21) does not have any mention of Te, so how are they assigned the Te related reactions?

Our response: Thanks for your kind consideration. According to your comments, we have added more information on the CV curves and corresponding references in the revised manuscript. Reference 21 focused on the research of ZnSe. We cited it here

to explain the formation of Na₂Se. The detailed revision is as follows:

Page 16: “During the first cathodic scan, an unobtrusively wide peak appears at 0.80 V, which results from the formation of the SEI film and the insertion of Na ions into the ZnSe_{0.7}Te_{0.3}^{32,33}. Two additional strong cathodic peaks at about 0.35 and 0.19 V may be due to the conversion of ZnSe_{0.7}Te_{0.3} to the metal Zn, Na₂Se, and Na₂Te and further alloying of Zn³⁴⁻³⁶. During the first anodic scan, oxidation peaks at about 0.97 and 1.16 V are associated with the dealloying reaction of NaZn₁₃ and the oxidation of metal Zn to ZnSe, respectively³⁴. The small anodic peak at about 2.18 V may be related to the oxidation of Na₂Te to Te^{37,24}. These analyses will be further demonstrated by the XRD and HRTEM measurements below. A pair of weak redox peaks near 0 V can be attributed to the insertion/extraction of Na⁺ from the hollow bowl-like carbon³⁸. In subsequent scans, the CV curves almost overlapped, indicating excellent electrochemistry reversibility and cyclic stability. ZnSe@C has similar CV profiles to ZnSe_{0.7}Te_{0.3}@C in Supplementary Fig. 16a. For comparison, the presence of two cathode peaks at 0.34 V and 0.10 V in ZnSe@C correspond to the conversion of ZnSe to Zn and Na₂Se, and further alloying of Zn.”

Q27: Details on the cell testing should be provided in the main body of the manuscript.

Our response: We appreciate the reviewer’s suggestions. Details on the cell testing are provided in the main body of the revised manuscript. The detailed revision is as follows:

Line 12-15, Page 11: “To evaluate the electrochemical performance of the samples, the synthesized electrodes were assembled into coin-type cells and tested in a 0.01-3 V potential window at 25 °C, using sodium foil as both the reference and counter electrodes.”.

Q28: What were the specific capacities on the first cycle? Please add this information to the main body of the text.

Our response: We appreciate the reviewer’s suggestions. We have added the

specific capacities on the first cycle in the revised manuscript. The detailed revision is as follows:

Line 3-4, Page 15: “The first discharge capacities of ZnSe_{0.7}Te_{0.3}@C, ZnSe_{0.8}Te_{0.2}@C, ZnSe_{0.5}Te_{0.5}@C, and ZnSe@C at the current density of 1 A g⁻¹ are 388.4, 382.0, 445.3, 303.4 mAh g⁻¹, respectively.”.

Q29: It is mentioned that adding Te to the material increases the specific surface area, but I did not see where the authors conducted surface area measurements.

Our response: Thanks for your kind consideration. The specific surface area and pore size distribution data can be derived from the measurements of the nitrogen absorption and desorption test. We have explained more information on the specific surface area data in the revised manuscript. The detailed revision is as follows:

Page 10-11: “Nitrogen adsorption–desorption measurements were carried out to study their porous profiles and specific surface area of ZnSe@C and ZnSe_{0.7}Te_{0.3}@C nanocomposites, as shown in Supplementary Fig. 10. The adsorption isotherms of them are typical type III isotherms, and the H₃ hysteresis appears when the relative pressure of P/P₀ is greater than 4.5, indicating the presence of mesoporous structures²⁸. According to nitrogen adsorption–desorption measurements results, the Brunauer-Emmett-Teller (BET) specific surface area of ZnSe_{0.7}Te_{0.3}@C nanocomposites is 270.6 m² g⁻¹, which is much greater than that of ZnSe@C (172.7 m² g⁻¹), pertaining to the reduced size of nanoparticles after introduction of Te atoms.”.

Q30: Is Figure 6a ex-situ XRD of the electrode? This point should be noted more clearly. Why was ex-situ XRD only conducted after 2 full cycles? Conducting it at various points along the 1st charge discharge curve would be more instructive as to the conversion reaction mechanism and at what voltages different parts of the reaction occur. As it is, the proposed reaction schemes have little data to back them up.

Our response: We appreciate the reviewer's questions and suggestions. Figure 6a in the previous manuscript is ex-situ XRD pattern performed after 2 complete cycles. According to the reviewer's suggestion, we conducted ex-situ XRD characterization of ZnSe_{0.7}Te_{0.3}@C electrode during the initial charge and discharge process, and the results are shown in the revised Fig. 6a. The corresponding discussion has been added in the revised manuscript. The detailed revision is as follows:

Page 16-17: "As for the electrochemistry of sodium ion reactions at ZnSe_{0.7}Te_{0.3}@C, it was further revealed by ex-situ XRD and ex-situ TEM characterizations. Figure 6a shows the ex-situ XRD pattern of ZnSe_{0.7}Te_{0.3}@C during the first discharge and charge process. Copper foil acting as current collector shows strong XRD peaks at 43.3°, 50.3° and 74.0°. The other diffraction peaks are located at 27.1°, 45.0°, 53.4°, 65.7° and 72.3°, demonstrating the pure phase of ZnSe_{0.7}Te_{0.3}@C present in the original electrode. As the discharge continues, these peak intensities of ZnSe_{0.7}Te_{0.3}@C gradually decrease, and then disappear completely. When discharged to 0.4 V, new weak peaks appear at 23.8° & 34.4°, assignable to the production of Na₂Te, and peaks at 31.8° & 36.2° are attributable to the production of NaZn₁₃ and Zn. When discharged to 0.01 V, a small peak appears at 37.4° due to the formation of Na₂Se. This demonstrates the conversion reaction of ZnSe_{0.7}Te_{0.3} to Na₂Se, Na₂Te and NaZn₁₃ occurs during the discharge process. During the continuous charging process, these characteristic peaks of Na₂Se, Na₂Te, NaZn₁₃ gradually weaken and disappear, indicating that Na₂Se, Na₂Te, NaZn₁₃ gradually transform into ZnSe and Te. "

Fig. 6a Ex situ XRD patterns of $\text{ZnSe}_{0.7}\text{Te}_{0.3}@C$ electrodes during the first charge and discharge process.

Q31: For the comparative experiment, were the Te nanoparticles also carbon coated? Also, the cycle life of the $\text{ZnSe}@C/\text{Te}$ material is closer to 350 in the figure (maybe 330-340, but the authors state in text that it is 300).

Our response: Thanks for your kind consideration. The $\text{ZnSe}@C/\text{Te}$ material in the comparison experiment was characterized by TEM, as shown in Supplementary Fig. 17. It can be seen from the figure that Te nanoparticles are also coated with carbon. We are sorry for the wrong statement about the 300 cycle life. We have revised the expression in the revised manuscript. The detailed revision is as follows:

Line 14-20, Page 18: “The $\text{ZnSe}@C/\text{Te}$ material in the comparison experiment was characterized by TEM, as shown in Supplementary Fig. 17. It can be seen from the figure that Te and ZnSe nanoparticles are also coated with carbon. Supplementary Fig. 18 shows the cycling performance of $\text{ZnSe}@C/\text{Te}$ at the current density of 1 A g^{-1} . It just can maintain the lifetime of 339 cycles and then break. A low reversible capacity of 223 mAh g^{-1} was offered, much worse than $\text{ZnSe}_{0.7}\text{Te}_{0.3}@C$ (310.9 mAh g^{-1}).”.

Supplementary Fig. 17 TEM image of ZnSe@C/Te.

Discussion

Q32: This section should be renamed to conclusion.

Our response: Many thanks to the valuable suggestion. According to the reviewer's suggestion, we have renamed this section as Conclusion.

Other

Q33: Can the current densities also be provided as C-rates? That would be helpful in comparison to other materials.

Our response: Many thanks to the helpful comments and valuable suggestions. We converted the current density and the C rate, and got $1 \text{ A g}^{-1} = 1.6 \text{ C}$. On page 34 of revised manuscript, we have added the conversion of current density and C rate in the caption of Fig. 5 in the revised manuscript.

Q34: Chemical suppliers and purities should be provided in the methods section.

Our response: Thanks for your kind consideration. We have provided chemical suppliers and purities in the methods section in the revised manuscript. The detailed revision is as follows:

Line 17-23, Page 19: “Methanol ($\geq 99.5\%$), hexadecyltrimethylammonium bromide

(C₁₉H₄₂BrN, $\geq 99.0\%$), zinc nitrate hexahydrate (Zn(NO₃)₂·6H₂O, $\geq 99.0\%$), formaldehyde aqueous solution (CH₂O, 37.0-40.0%) and ammonia solution (NH₄OH, 25.0-28.0%) were bought from Sinopharm Chemical Reagent Co., Ltd. 2-methylimidazole (C₄H₆N₂, 98%), selenium (Se, $\geq 99.99\%$) and tellurium (Te, $\geq 99.9\%$) were bought from the Aladdin. M-aminophenol (C₆H₇NO, 98%) was bought from 9ding chemical (Shanghai) limited. None of the reagents were further purified.”.

Q35: More parameters/details should be provided for the structural characterization methods as they were for XRD.

Our response: Many thanks to the reviewer’s comments. We have supplemented the parameters/details of the structural characterization method in the revised manuscript. The detailed revision is as follows:

Line 7-21, Page 21: “The morphology, composition and microstructure of the samples were characterized by field emission scanning electron microscope (FESEM, GeminiSEM 300, Carl Zeiss Microscopy Ltd.) coupled with energy-dispersive X-ray spectroscopy (EDS) and transmission electron microscope (TEM, Talos F200X, accelerating voltage of 200 kV). High angle annular dark field-scanning TEM (HAADF-STEM) imaging was done on Thermo Scientific Spectra 300 equipped with spherical aberration correction system and the microscope was operated at 300 kV. The components and valence states of the ZnSe_{0.7}Te_{0.3}@C were measured by X-ray photoelectron spectroscopy (XPS, Thermo ESCALAB 250 Xi) with a monochromatic Al K α X-ray source ($h\nu = 1486.6$ eV). And all binding energies were calibrated using C 1s signals at 284.8 eV. Thermogravimetric analysis (TGA) of the sample was carried out on Simultaneous Thermal Analyzer (TGA/DSC3+) in an air atmosphere at a heating rate of 20 °C/min. Nitrogen adsorption–desorption measurements were carried out on the Autosorb 6B instrument at 70 K.”.

Q36: The electrolyte chosen for cell testing is a bit unusual, some context for this choice would be helpful.

Our response: Many thanks to the helpful and valuable suggestions. We have added

some explanation in the revised manuscript. The detailed revision is as follows:

Line 7-15, Page 11: “NaPF₆ as a conventional sodium electrolyte salt has excellent ionic conductivity²⁹. Compared with carbonate solvent, dimethoxyethane (DME) can effectively change the interface and reduce the charge transfer resistance³⁰. A thin but stable sodium ion permeable solid electrolyte interface (SEI) layer is easily formed in the electrolyte NaPF₆-DME, facilitating its cycling and rate performance³¹. So 1 M NaPF₆ in DME was used as the electrolyte for cell testing.”.

Q37: There is no information on galvanostatic cycling parameters in the methods section.

Our response: Many thanks to the reviewer’s comments. We have added the information in the revised manuscript. The detailed revision is as follows:

Line 6-9, Page 22: “The Land battery test system (LAND-CT2001A, Wuhan, China) was used to conduct galvanostatic charge/discharge tests at various current densities in a constant temperature chamber of 25° within the voltage range of 0.01-3 V vs Na⁺/Na.”.

Q38: What was the voltage range and scan rate for the cyclic voltammetry?

Our response: Many thanks to the reviewer’s comments. We have added the information in the revised manuscript. The detailed revision is as follows:

Line 12-14, Page 22: “The cyclic voltammetry (CV) curves of the samples were measured on an electrochemical workstation (Shanghai Chenhua electrochemistry workstation, CHI760D) with a sweep speed of 0.2 mV s⁻¹ and a voltage range of 0-3 V.”.

Q39: What was the frequency range and perturbation voltage for EIS measurements?

Our response: Many thanks to the reviewer’s comments. We have added the information in the revised manuscript. The detailed revision is as follows:

Line 15-18, Page 22: “Electrochemical impedance spectroscopy (EIS) measurements

were done on an electrochemical workstation (Shanghai Chenhua electrochemistry workstation, CHI760D), and the amplitude of the AC voltage was set at 5 mV at 100 kHz to 0.01 Hz.”.

Q40: What type of electrochemical workstation was used?

Our response: Thanks for your question. The electrochemical workstation used is Shanghai Chenhua Electrochemistry workstation, the model is CHI760E. We have added the information in the revised manuscript. The detailed revision is as follows:

Line 12-18, Page 22: “The cyclic voltammetry (CV) curves of the samples were measured on an electrochemical workstation (Shanghai Chenhua electrochemistry workstation, CHI760D) with a sweep speed of 0.2 mV s⁻¹ and a voltage range of 0-3 V. Electrochemical impedance spectroscopy (EIS) measurements were done on an electrochemical workstation (Shanghai Chenhua electrochemistry workstation, CHI760D), and the amplitude of the AC voltage was set at 5 mV at 100 kHz to 0.01 Hz”.

Q41: More references should be added in the DFT calculations section (for the PBE functional, for example).

Our response: Thanks for your suggestion. We have added more references to the DFT calculations section in the revised manuscript. The detailed revision is as follows:

Line 21-24, Page 22: “The exchange correlation interaction of electrons was described by the generalized gradient approximation (GGA) of Perdew–Burke–Ernzerhof (PBE) functional⁴⁴, and the interaction between electrons and ions was described by the projected augmented wave (PAW) method⁴⁵.”.

Page 28:

44 Hao, Y. et al. Plasma-treated ultrathin ternary FePSe₃ nanosheets as a bifunctional electrocatalyst for efficient zinc–air batteries. *ACS Appl. Mater. Interfaces* **12**, 29393–29403 (2020).

45 Xia, H. et al. Evolution of stabilized 1T-MoS₂ by atomic-interface engineering of 2H-MoS₂/Fe-N_x towards enhanced sodium ion storage. *Angew. Chem. Int. Ed.* **62**, e202218282 (2023).

Q42: In Figure 3, a legend should be provided for what color each atom is.

Our response: Thanks for your kind consideration. We have provided legends in Fig. 3 to illustrate the color of each atom on page 32 of revised manuscript.

Q43: In Figure 5, cycling curves of the ZnSe base material would be helpful for comparison.

Our response: We appreciate the reviewer's suggestions. We have provide the long cycle performance at the current density of 5 A g^{-1} of ZnSe@C and ZnSe_{0.7}Te_{0.3}@C, as shown in the revised Fig. 5g in the revised manuscript. The detailed revision is as follows:

Page 15: "The stability and capacity of batteries at high currents are still significant, so a high-current cycle performance test of ZnSe_{0.7}Te_{0.3}@C and ZnSe@C was carried out, as shown in Fig. 5g. It can be seen that both ZnSe_{0.7}Te_{0.3}@C and ZnSe@C have good cyclic stability. After 1000 cycles, the discharge capacity of ZnSe_{0.7}Te_{0.3}@C remains 307 mAh g^{-1} , while that of ZnSe@C is only 118.8 mAh g^{-1} . The excellent electrochemical performance of ZnSe_{0.7}Te_{0.3}@C is attributed to the twin structures by Te atom doping, the optimal number of TBs and the improved conductivity. Hence, ZnSe_{0.7}Te_{0.3}@C was determined to continue other electrochemistry testing."

Fig. 5g Long cycle performance at the current density of 5 A g^{-1} of ZnSe@C and ZnSe_{0.7}Te_{0.3}@C.

Q44: Supporting information Figure 3b looks like it was cutoff or the figure was not formatted correctly.

Our response: Thanks for your kind consideration. We have replaced the TEM image

of ZnSe@C in Supplementary Fig. 3b.

Supplementary Fig. 3 a SEM and b TEM images of ZnSe@C.

Thanks for your comments again. If it could be considered for publication in **Nat. Commun.**, it would be greatly appreciated!

Manuscript: NCOMMS-24-22711A-Z

Title: Twin structures and substitutional dopants in ZnSe_{0.7}Te_{0.3}: the effect and transition during sodium ion electrochemistry

The authors greatly appreciate reviewers' insightful comments and careful review on our manuscript (NCOMMS-24-22711A-Z). This paper has been revised carefully according to the comments of the reviewers. The responses are listed point-by-point in the following contents, and revisions have been highlighted by red color in the revised manuscript. Following are our responses and detailed explanation towards these comments from the reviewers.

Responses to Reviewers:

To Reviewer #1:	2-14
To Reviewer #2:	15

Response to reviewers' comments:

Reviewer #1:

General comments: Authors have partly answered my queries and still need some clarification for this work. Following are the points need to address:

Our response: Thank you very much for your comments and suggestions on our revised manuscript. We have addressed all remaining issues raised by the reviewers.

1: In fig 6c why there is discontinuity in frequency at R and U points? Please provide zoom version (lets say -50 Hz to 200 Hz frequency range) of the phonon spectra (Fig 6 c) (specifically Gamma, T and U points). If there is any negative frequency exist, what is its origin and implication? At U point there is large negative frequency so what is the reliability of the calculation?

Our response: Thanks so much for the helpful comments. The coordinate of K point was selected incorrectly when we confirmed the phonon spectrum diagram (Supplementary Fig. 6c). We have already revised Fig. 6c in the Supplementary information. In addition, we performed a local amplification of the Supplementary Fig. 6c, as shown in Supplementary Fig. 6d and e. As can be seen from Supplementary Fig. 6c-e, the phonon spectrum for int model in which Te occupies interstitial position has virtual phonon mode, proving that its structure is unstable. The virtual frequency may be caused by the large Te atom occupying the interstitial position, causing huge structural distortion and profoundly affecting the normal arrangement of the surrounding atoms. Actually, according to spherical aberration corrected electron microscopy results, we have experimentally demonstrated that Te atoms replace Se atoms in ZnSe crystals instead of occupying interstitial positions in ZnSe crystals. The virtual frequency of phonon spectrum for int model is proved to be unstable by theoretical calculation, which is in agreement with our experimental results. The similar phenomenon of virtual frequency was reported in literature (such as *Phys. Rev. B*, **2015**, 92, 245131; *Phys. Rev. B*, **2020**, 101, 235405; *Phys. Rev. B*, **2022**, 105, 174105; *Appl. Phys. Lett.*, **2010**, 97, 162907). They mention that the phonon spectrum has a virtual frequency, indicating that the structure is unstable (**Figure R1**). We have revised the relevant statement in the revised manuscript. The detailed revisions are as follows:

Page 8: "However, the phonon spectrum for int model in which Te occupies interstitial position has virtual phonon mode, proving that its structure is unstable, as shown in Supplementary Fig. 6c-e. The virtual frequency may be caused by the large Te atom occupying the interstitial position, rendering huge structural distortion and profoundly affecting the normal arrangement of the surrounding atoms."

Supplementary Fig. 6 Phonon spectra for the structural models of (a) sub 1, (b) sub 3 and (c) int. d and e Enlarged view of the phonon spectra for int model. f The average energy per atom in the final state of different configuration structures.

Figure R1 a-d Referenced phonon spectra in reported literature (*Phys. Rev. B*, **2015**, 92, 245131; *Phys. Rev. B*, **2020**, 101, 235405; *Phys. Rev. B*, **2022**, 105, 174105; *Appl. Phys. Lett.*, **2010**, 97, 162907).

2: "From Supplementary Fig. 6a and 6b, it can be seen that the phonon spectra of sub 1 model with one Te atom replacing one Se atom and sub 3 model with

three Te atoms replacing three Se atoms have no virtual phonon mode, indicating that the models are mechanically and dynamically stable."

Our response: Thanks for your consideration. We have revised this sentence in the revised manuscript. The detailed revision is as follows:

Page 8: "From Supplementary Fig. 6a and 6b, it can be seen that the phonon spectra of sub 1 model with one Te atom replacing one Se atom and sub 3 model with three Te atoms replacing three Se atoms have no virtual phonon mode, indicating that the structures are dynamically stable."

3: Mechanical stability is usually depicted through the calculation of elastic stiffness constant. How authors conclude mechanical stability from phonon spectra should be prescribed in details. And mechanical stability of the structure should be analyzed in details.

Our response: We thank the reviewer for pointing it out. According to the reviewer's suggestions, we studied the mechanical stability of these structures (sub 1, sub 3, int) by calculating the elastic stiffness constant. The mechanical stability of these structures are analyzed in detail in the revised manuscript. The detailed revisions are as follows:

Supplementary information, Page 22-27: "Note 1: Calculations of elastic stiffness constant for the structural models of sub 1, sub 3 and int.

The sub 1 model with one Te atom replacing one Se atom belongs to the cubic crystal system and has only three independent constants, C_{11} , C_{12} and C_{44} .

$$C = \begin{pmatrix} C_{11} & C_{12} & C_{12} & 0 & 0 & 0 \\ C_{12} & C_{11} & C_{12} & 0 & 0 & 0 \\ C_{12} & C_{12} & C_{11} & 0 & 0 & 0 \\ 0 & 0 & 0 & C_{44} & 0 & 0 \\ 0 & 0 & 0 & 0 & C_{44} & 0 \\ 0 & 0 & 0 & 0 & 0 & C_{44} \end{pmatrix} \quad (7)$$

According to the mechanical stability criteria^{6,7}, mechanical stability requires the elastic constants of cubic crystal systems to meet the following requirements:

$$C_{44} > 0, \quad C_{11} - C_{12} > 0, \quad C_{11} + 2C_{12} > 0 \quad (8)$$

The elastic constant of sub 1 model can be obtained by theoretical calculation as follows:

$$C_{\text{sub 1}} = \begin{pmatrix} 92.196 & 56.159 & 56.159 & 0.000 & 0.000 & 0.000 \\ 56.159 & 92.196 & 56.159 & 0.000 & 0.000 & 0.000 \\ 56.159 & 56.159 & 92.196 & 0.000 & 0.000 & 0.000 \\ 0.000 & 0.000 & 0.000 & 39.666 & 0.000 & 0.000 \\ 0.000 & 0.000 & 0.000 & 0.000 & 39.666 & 0.000 \\ 0.000 & 0.000 & 0.000 & 0.000 & 0.000 & 39.666 \end{pmatrix} \quad (9)$$

The elastic constants of sub 1 model satisfy the criteria in formula (8), so sub 1 model is mechanically stable.

The sub 3 model with three Te atoms replacing three Se atoms belongs to the monoclinic crystal system, and has 13 independent elastic constants.

$$C = \begin{pmatrix} C_{11} & C_{12} & C_{13} & 0 & C_{15} & 0 \\ C_{12} & C_{22} & C_{23} & 0 & C_{25} & 0 \\ C_{13} & C_{23} & C_{33} & 0 & C_{35} & 0 \\ 0 & 0 & 0 & C_{44} & 0 & C_{46} \\ C_{15} & C_{25} & C_{35} & 0 & C_{55} & 0 \\ 0 & 0 & 0 & C_{46} & 0 & C_{66} \end{pmatrix} \quad (10)$$

Similarly, mechanical stability requires that the elastic constants of monoclinic crystal systems meet the following requirements:

$$\begin{aligned} C_{11} > 0, \quad C_{11}C_{22} - C_{12}^2 > 0, \quad C_{11}C_{22}C_{33} + 2C_{12}C_{13}C_{23} - C_{11}C_{23}^2 - C_{22}C_{13}^2 - C_{33}C_{12}^2 > 0, \\ C_{44} > 0, \quad C_{11}(C_{22}C_{33}C_{55} + 2C_{23}C_{25}C_{35} - C_{33}C_{25}^2 - C_{22}C_{35}^2 - C_{55}C_{23}^2) - C_{12}(C_{12}C_{33}C_{55} + \\ C_{23}C_{15}C_{35} + C_{13}C_{25}C_{35} - C_{33}C_{15}C_{25} - C_{12}C_{35}^2 - C_{13}C_{23}C_{55}) + C_{13}(C_{12}C_{23}C_{55} + \\ C_{22}C_{35}C_{15} + C_{13}C_{25}^2 - C_{23}C_{15}C_{25} - C_{12}C_{25}C_{35} - C_{22}C_{13}C_{55}) - C_{15}(C_{12}C_{23}C_{35} + \\ C_{22}C_{33}C_{15} + C_{13}C_{23}C_{25} - C_{15}C_{23}^2 - C_{12}C_{33}C_{25} - C_{22}C_{13}C_{35}) > 0, \quad C_{44}C_{66} - C_{46}^2 > 0 \end{aligned} \quad (11)$$

The elastic constants of sub 3 model can be obtained by theoretical calculation as follows:

$$C_{\text{sub 3}} = \begin{pmatrix} 91.476 & 54.352 & 54.458 & 0.000 & -0.310 & 0.000 \\ 54.352 & 91.439 & 54.476 & 0.000 & 0.172 & 0.000 \\ 54.458 & 54.476 & 90.803 & 0.000 & 0.350 & 0.000 \\ 0.000 & 0.000 & 0.000 & 39.089 & 0.000 & 0.697 \\ -0.310 & 0.172 & 0.350 & 0.000 & 39.135 & 0.000 \\ 0.000 & 0.000 & 0.000 & 0.697 & 0.000 & 39.550 \end{pmatrix} \quad (12)$$

The elastic constants of sub 3 model satisfy the criteria in formula (11), so sub 3 model is mechanically stable.

The int model with one Te atom occupying interstitial position belongs to the triclinic crystal system, and has 21 independent elastic constants.

$$C = \begin{pmatrix} C_{11} & C_{12} & C_{13} & C_{14} & C_{15} & C_{16} \\ C_{12} & C_{22} & C_{23} & C_{24} & C_{25} & C_{26} \\ C_{13} & C_{23} & C_{33} & C_{34} & C_{35} & C_{36} \\ C_{14} & C_{24} & C_{34} & C_{44} & C_{45} & C_{46} \\ C_{15} & C_{25} & C_{35} & C_{45} & C_{55} & C_{56} \\ C_{16} & C_{26} & C_{36} & C_{46} & C_{56} & C_{66} \end{pmatrix} \quad (13)$$

Similarly, mechanical stability requires that the elastic constants of triclinic crystal systems meet the following requirements:

$$\begin{aligned} & C_{11} > 0, \quad C_{11}C_{22} - C_{12}^2 > 0, \quad C_{11}C_{22}C_{33} + 2C_{12}C_{13}C_{23} - C_{11}C_{23}^2 - C_{22}C_{13}^2 - C_{33}C_{12}^2 > \\ & 0, \quad C_{11}(C_{22}C_{33}C_{44} + 2C_{23}C_{24}C_{34} - C_{33}C_{24}^2 - C_{22}C_{34}^2 - C_{44}C_{23}^2) - C_{12}(C_{12}C_{33}C_{44} + \\ & C_{23}C_{14}C_{34} + C_{13}C_{24}C_{34} - C_{33}C_{14}C_{24} - C_{12}C_{34}^2 - C_{13}C_{23}C_{44}) + C_{13}(C_{12}C_{23}C_{44} + \\ & C_{22}C_{14}C_{34} + C_{13}C_{24}^2 - C_{23}C_{14}C_{24} - C_{12}C_{24}C_{34} - C_{22}C_{13}C_{44}) - C_{14}(C_{12}C_{23}C_{34} + \\ & C_{22}C_{33}C_{14} + C_{13}C_{23}C_{24} - C_{14}C_{23}^2 - C_{12}C_{33}C_{24} - C_{22}C_{13}C_{34}) > 0, \\ & C_{11}(a - b + c - d) - C_{12}(e - f + g - h) + C_{13}(i - j + k - l) - C_{14}(m - n + o - p) + C_{15}(q - r \\ & + s - t) > 0, \quad \sum_{i=1}^{20} A_i B_i > 0 \end{aligned} \quad (14)$$

$$\text{where } a = C_{22}(C_{33}C_{44}C_{55} + 2C_{34}C_{35}C_{45} - C_{44}C_{35}^2 - C_{33}C_{45}^2 - C_{55}C_{34}^2),$$

$$\begin{aligned}
b &= C_{23}(C_{23}C_{44}C_{55} + C_{34}C_{25}C_{45} + C_{24}C_{35}C_{45} - C_{44}C_{25}C_{35} - C_{23}C_{45}^2 - C_{24}C_{34}C_{55}), \\
c &= C_{24}(C_{23}C_{34}C_{55} + C_{33}C_{45}C_{25} + C_{24}C_{35}^2 - C_{34}C_{25}C_{35} - C_{23}C_{35}C_{45} - C_{33}C_{24}C_{55}), \\
d &= C_{25}(C_{23}C_{34}C_{45} + C_{33}C_{44}C_{25} + C_{24}C_{34}C_{35} - C_{34}^2C_{25} - C_{23}C_{44}C_{35} - C_{33}C_{24}C_{45}), \\
e &= C_{12}(C_{33}C_{44}C_{55} + 2C_{34}C_{35}C_{45} - C_{44}C_{35}^2 - C_{33}C_{45}^2 - C_{55}C_{34}^2), \\
f &= C_{23}(C_{13}C_{44}C_{55} + C_{34}C_{15}C_{45} + C_{14}C_{35}C_{45} - C_{44}C_{15}C_{35} - C_{13}C_{45}^2 - C_{14}C_{34}C_{55}), \\
g &= C_{24}(C_{13}C_{34}C_{55} + C_{33}C_{45}C_{15} + C_{14}C_{35}^2 - C_{34}C_{15}C_{35} - C_{13}C_{35}C_{45} - C_{33}C_{14}C_{55}), \\
h &= C_{25}(C_{13}C_{34}C_{45} + C_{33}C_{44}C_{15} + C_{14}C_{34}C_{35} - C_{34}^2C_{15} - C_{13}C_{44}C_{35} - C_{33}C_{14}C_{45}), \\
i &= C_{12}(C_{23}C_{44}C_{55} + C_{34}C_{25}C_{45} + C_{24}C_{35}C_{45} - C_{44}C_{25}C_{35} - C_{23}C_{45}^2 - C_{24}C_{34}C_{55}), \\
j &= C_{22}(C_{13}C_{44}C_{55} + C_{34}C_{15}C_{45} + C_{14}C_{35}C_{45} - C_{44}C_{15}C_{35} - C_{13}C_{45}^2 - C_{14}C_{34}C_{55}), \\
k &= C_{24}(C_{13}C_{24}C_{55} + C_{23}C_{15}C_{45} + C_{14}C_{25}C_{35} - C_{24}C_{15}C_{35} - C_{13}C_{25}C_{45} - C_{23}C_{14}C_{55}), \\
l &= C_{25}(C_{13}C_{24}C_{45} + C_{23}C_{44}C_{15} + C_{14}C_{34}C_{25} - C_{24}C_{34}C_{15} - C_{13}C_{44}C_{25} - C_{23}C_{14}C_{45}), \\
m &= C_{12}(C_{23}C_{34}C_{55} + C_{33}C_{25}C_{45} + C_{24}C_{35}^2 - C_{34}C_{25}C_{35} - C_{23}C_{35}C_{45} - C_{33}C_{24}C_{55}), \\
n &= C_{22}(C_{13}C_{34}C_{55} + C_{33}C_{15}C_{45} + C_{14}C_{35}^2 - C_{34}C_{15}C_{35} - C_{13}C_{35}C_{45} - C_{33}C_{14}C_{55}), \\
o &= C_{23}(C_{13}C_{24}C_{55} + C_{23}C_{45}C_{15} + C_{14}C_{25}C_{35} - C_{24}C_{15}C_{35} - C_{13}C_{25}C_{45} - C_{23}C_{14}C_{55}), \\
p &= C_{25}(C_{13}C_{24}C_{35} + C_{23}C_{34}C_{15} + C_{33}C_{14}C_{25} - C_{33}C_{24}C_{15} - C_{13}C_{34}C_{25} - C_{23}C_{14}C_{35}), \\
q &= C_{12}(C_{23}C_{34}C_{45} + C_{33}C_{44}C_{25} + C_{24}C_{34}C_{35} - C_{25}C_{34}^2 - C_{23}C_{44}C_{35} - C_{24}C_{33}C_{45}), \\
r &= C_{22}(C_{13}C_{34}C_{45} + C_{33}C_{44}C_{15} + C_{14}C_{34}C_{35} - C_{15}C_{34}^2 - C_{13}C_{44}C_{35} - C_{33}C_{14}C_{45}), \\
s &= C_{23}(C_{13}C_{24}C_{45} + C_{23}C_{44}C_{15} + C_{14}C_{25}C_{34} - C_{34}C_{24}C_{15} - C_{13}C_{25}C_{44} - C_{14}C_{23}C_{45}), \\
t &= C_{24}(C_{13}C_{24}C_{35} + C_{23}C_{34}C_{15} + C_{14}C_{25}C_{33} - C_{33}C_{24}C_{15} - C_{13}C_{34}C_{25} - C_{23}C_{14}C_{35}), \\
B_1 &= C_{14}C_{25}C_{36} + C_{24}C_{35}C_{16} + C_{34}C_{15}C_{26} - C_{16}C_{25}C_{34} - C_{14}C_{35}C_{26} - C_{24}C_{15}C_{36}, \\
B_2 &= C_{14}C_{25}C_{46} + C_{24}C_{45}C_{16} + C_{44}C_{15}C_{26} - C_{44}C_{25}C_{16} - C_{14}C_{45}C_{26} - C_{24}C_{15}C_{46}, \\
B_3 &= C_{14}C_{25}C_{56} + C_{24}C_{55}C_{16} + C_{15}C_{45}C_{26} - C_{25}C_{45}C_{16} - C_{14}C_{55}C_{26} - C_{24}C_{15}C_{56}, \\
B_4 &= C_{14}C_{25}C_{66} + C_{24}C_{56}C_{16} + C_{15}C_{26}C_{46} - C_{16}C_{25}C_{46} - C_{15}C_{24}C_{66} - C_{14}C_{26}C_{56}, \\
B_5 &= C_{14}C_{35}C_{46} + C_{34}C_{45}C_{16} + C_{44}C_{15}C_{36} - C_{44}C_{35}C_{16} - C_{14}C_{45}C_{36} - C_{34}C_{15}C_{46}, \\
B_6 &= C_{14}C_{35}C_{56} + C_{34}C_{55}C_{16} + C_{15}C_{36}C_{45} - C_{45}C_{35}C_{16} - C_{14}C_{55}C_{36} - C_{34}C_{15}C_{56}, \\
B_7 &= C_{14}C_{35}C_{66} + C_{34}C_{56}C_{16} + C_{15}C_{36}C_{46} - C_{16}C_{35}C_{46} - C_{14}C_{36}C_{56} - C_{15}C_{34}C_{66}, \\
B_8 &= C_{14}C_{45}C_{56} + C_{44}C_{55}C_{16} + C_{15}C_{45}C_{46} - C_{16}C_{45}^2 - C_{14}C_{46}C_{55} - C_{15}C_{44}C_{56},
\end{aligned}$$

$$\begin{aligned}
B_9 &= C_{14}C_{45}C_{66} + C_{44}C_{56}C_{16} + C_{15}C_{46}^2 - C_{16}C_{45}C_{46} - C_{14}C_{46}C_{56} - C_{15}C_{44}C_{66}, \\
B_{10} &= C_{14}C_{55}C_{66} + C_{45}C_{16}C_{56} + C_{15}C_{46}C_{56} - C_{55}C_{16}C_{46} - C_{14}C_{56}^2 - C_{15}C_{45}C_{66}, \\
B_{11} &= C_{24}C_{35}C_{46} + C_{34}C_{45}C_{26} + C_{44}C_{25}C_{36} - C_{44}C_{35}C_{26} - C_{24}C_{45}C_{36} - C_{34}C_{25}C_{46}, \\
B_{12} &= C_{24}C_{35}C_{56} + C_{34}C_{55}C_{26} + C_{25}C_{36}C_{45} - C_{35}C_{45}C_{26} - C_{24}C_{55}C_{36} - C_{34}C_{25}C_{56}, \\
B_{13} &= C_{24}C_{35}C_{66} + C_{34}C_{26}C_{56} + C_{25}C_{36}C_{46} - C_{35}C_{26}C_{46} - C_{24}C_{36}C_{56} - C_{34}C_{25}C_{66}, \\
B_{14} &= C_{24}C_{45}C_{56} + C_{44}C_{55}C_{26} + C_{25}C_{46}C_{45} - C_{45}^2C_{26} - C_{24}C_{55}C_{46} - C_{25}C_{44}C_{56}, \\
B_{15} &= C_{24}C_{45}C_{66} + C_{44}C_{26}C_{56} + C_{25}C_{46}^2 - C_{45}C_{26}C_{46} - C_{24}C_{46}C_{56} - C_{44}C_{25}C_{66}, \\
B_{16} &= C_{24}C_{55}C_{66} + C_{45}C_{56}C_{26} + C_{25}C_{56}C_{46} - C_{26}C_{55}C_{46} - C_{24}C_{56}^2 - C_{25}C_{45}C_{66}, \\
B_{17} &= C_{34}C_{45}C_{56} + C_{44}C_{55}C_{36} + C_{35}C_{45}C_{46} - C_{36}C_{45}^2 - C_{44}C_{35}C_{56} - C_{34}C_{55}C_{46}, \\
B_{18} &= C_{34}C_{45}C_{66} + C_{44}C_{36}C_{56} + C_{35}C_{46}^2 - C_{45}C_{36}C_{46} - C_{34}C_{46}C_{56} - C_{44}C_{35}C_{66}, \\
B_{19} &= C_{34}C_{55}C_{66} + C_{45}C_{36}C_{56} + C_{35}C_{46}C_{56} - C_{55}C_{36}C_{46} - C_{34}C_{56}^2 - C_{35}C_{45}C_{66}, \\
B_{20} &= C_{44}C_{55}C_{66} + C_{45}C_{56}C_{46} + C_{45}C_{46}C_{56} - C_{55}C_{46}^2 - C_{66}C_{45}^2 - C_{44}C_{56}^2, \\
A_1 &= -C_{14}C_{25}C_{36} - C_{15}C_{34}C_{26} - C_{16}C_{24}C_{35} + C_{14}C_{35}C_{26} + C_{15}C_{24}C_{36} + C_{16}C_{34}C_{25}, \\
A_2 &= C_{13}C_{25}C_{36} + C_{15}C_{33}C_{26} + C_{16}C_{23}C_{35} - C_{13}C_{26}C_{35} - C_{15}C_{23}C_{36} - C_{16}C_{25}C_{33}, \\
A_3 &= -C_{13}C_{24}C_{36} - C_{14}C_{33}C_{26} - C_{16}C_{23}C_{34} + C_{13}C_{34}C_{26} + C_{14}C_{23}C_{36} + C_{16}C_{24}C_{33}, \\
A_4 &= C_{13}C_{24}C_{35} + C_{14}C_{33}C_{25} + C_{15}C_{23}C_{34} - C_{13}C_{34}C_{25} - C_{14}C_{23}C_{35} - C_{15}C_{33}C_{24}, \\
A_5 &= -C_{12}C_{25}C_{36} - C_{23}C_{15}C_{26} - C_{22}C_{35}C_{16} + C_{12}C_{35}C_{26} + C_{22}C_{15}C_{36} + C_{23}C_{25}C_{16}, \\
A_6 &= C_{12}C_{24}C_{36} + C_{23}C_{14}C_{26} + C_{22}C_{34}C_{16} - C_{12}C_{34}C_{26} - C_{22}C_{14}C_{36} - C_{23}C_{24}C_{16}, \\
A_7 &= -C_{12}C_{24}C_{35} - C_{23}C_{14}C_{25} - C_{22}C_{34}C_{15} + C_{12}C_{34}C_{25} + C_{22}C_{14}C_{35} + C_{23}C_{24}C_{15}, \\
A_8 &= -C_{12}C_{23}C_{36} - C_{13}C_{23}C_{26} - C_{22}C_{33}C_{16} + C_{12}C_{33}C_{26} + C_{22}C_{13}C_{36} + C_{16}C_{23}^2, \\
A_9 &= C_{12}C_{23}C_{35} + C_{13}C_{23}C_{25} + C_{22}C_{33}C_{15} - C_{12}C_{33}C_{25} - C_{22}C_{13}C_{35} - C_{15}C_{23}^2, \\
A_{10} &= -C_{12}C_{23}C_{34} - C_{13}C_{23}C_{24} - C_{22}C_{33}C_{14} + C_{12}C_{33}C_{24} + C_{22}C_{13}C_{34} + C_{14}C_{23}^2, \\
A_{11} &= C_{11}C_{25}C_{36} + C_{13}C_{15}C_{26} + C_{12}C_{35}C_{16} - C_{11}C_{35}C_{26} - C_{12}C_{15}C_{36} - C_{13}C_{25}C_{16}, \\
A_{12} &= -C_{11}C_{24}C_{36} - C_{13}C_{14}C_{26} - C_{12}C_{16}C_{34} + C_{11}C_{34}C_{26} + C_{12}C_{14}C_{36} + C_{13}C_{24}C_{16}, \\
A_{13} &= C_{11}C_{24}C_{35} + C_{13}C_{14}C_{25} + C_{12}C_{34}C_{15} - C_{12}C_{14}C_{35} - C_{13}C_{24}C_{15} - C_{11}C_{34}C_{25}, \\
A_{14} &= C_{11}C_{23}C_{36} + C_{26}C_{13}^2 + C_{12}C_{33}C_{16} - C_{11}C_{33}C_{26} - C_{12}C_{13}C_{36} - C_{13}C_{23}C_{16}, \\
A_{15} &= -C_{11}C_{23}C_{35} - C_{25}C_{13}^2 - C_{12}C_{33}C_{15} + C_{11}C_{33}C_{25} + C_{12}C_{13}C_{35} + C_{13}C_{23}C_{15},
\end{aligned}$$

$$A_{16} = C_{11}C_{23}C_{34} + C_{24}C_{13}^2 + C_{12}C_{33}C_{14} - C_{11}C_{33}C_{24} - C_{12}C_{13}C_{34} - C_{13}C_{23}C_{14},$$

$$A_{17} = -C_{11}C_{22}C_{36} - C_{12}C_{13}C_{26} - C_{12}C_{23}C_{16} + C_{11}C_{23}C_{26} + C_{36}C_{12}^2 + C_{22}C_{13}C_{16},$$

$$A_{18} = C_{11}C_{22}C_{35} + C_{12}C_{13}C_{25} + C_{12}C_{23}C_{15} - C_{11}C_{23}C_{25} - C_{35}C_{12}^2 - C_{22}C_{13}C_{15},$$

$$A_{19} = -C_{11}C_{22}C_{34} - C_{12}C_{13}C_{24} - C_{12}C_{23}C_{14} + C_{11}C_{23}C_{24} + C_{34}C_{12}^2 + C_{22}C_{13}C_{14},$$

$$A_{20} = C_{11}C_{22}C_{33} + C_{12}C_{13}C_{23} + C_{12}C_{13}C_{23} - C_{11}C_{23}^2 - C_{33}C_{12}^2 - C_{22}C_{13}^2.$$

The elastic constants of int model can be obtained by theoretical calculation as follows:

$$C_{\text{int}} = \begin{pmatrix} 86.138 & 53.252 & 54.007 & 1.614 & 0.267 & -2.713 \\ 53.252 & 88.327 & 54.647 & 2.124 & 2.529 & 0.184 \\ 54.007 & 54.647 & 83.672 & -3.625 & -3.672 & -1.728 \\ 1.614 & 2.124 & -3.625 & 32.738 & -1.386 & 5.226 \\ 0.267 & 2.529 & -3.672 & -1.386 & 37.284 & 1.442 \\ -2.713 & 0.184 & -1.728 & 5.226 & 1.442 & 32.056 \end{pmatrix} \quad (15)$$

The elastic constants of int model satisfy the criteria in formula (14), so int model is also mechanically stable.”

In addition, we have added relevant discussion about the mechanical stability of these structural models in the revised manuscript. The detailed revisions are as follows:

Page 8: “Furthermore, the elastic constants of sub 1, sub 3 and int models have been calculated, and the results indicate that these structural models are mechanically stable (The calculations and result details are in Note 1 in Supplementary information).”.

4: How chemical potential of Se and Te is calculated is not clear. It should be clearly stated as it involves in formation energy which is related to stability.

Our response: Thank you for your professional question. According to the suggestions of reviewers, we have supplemented the chemical potential calculation methods for Se and Te. These revisions have added in the revised manuscript. The detailed revision is as follows:

Page 23-24: “The defect formation energy of structural models was also calculated with the chemical potential of the components and the calculation formula was as follows:

$$\Delta H_D = E_D - E_h + \sum n_i \mu_i$$

$$\mu_i = E_i/N_i$$

where ΔH_D is the defect formation energy, E_D is the total energy of the supercell with the defect (D), E_h is the total energy of the ZnSe supercell, μ_i is the chemical potentials of element Te and Se. n_i is the numbers of Te and Se atoms that were removed or added to form the system. E_i is the total energy of solid structure of Se and Te. N_i is the total number of atoms of solid structure of Se and Te. The chemical potentials of Te and Se were obtained by the solid structure models of Se and Te after structure optimization, as shown in Supplementary Fig. 21.

$$E_{\text{per atom}} = E_{\text{total}}/N_{\text{atom}}$$

$E_{\text{per atom}}$ is the average energy per atom in the final state, E_{total} is the total energy of the structural model, and N_{atom} is the total number of atoms in the structural model.”.

Supplementary Fig. 21 Solid structure models of **a** Se and **b** Te after structure optimization.

5: Since the authors are showing the application of the system for Sodium Ion Batteries (SIBs), so they should also see how these structures behave in room temperature and some higher temperatures through the study of molecular dynamics.

Our response: Many thanks to the helpful comments and valuable suggestions. Based on your suggestion, we studied the behavior of sub 1 and sub 3 at 298 K and 313 K by molecular dynamics simulations, as shown in Supplementary Fig. 7. These revisions have been added in the revised manuscript. The detailed revision is as follows:

Page 8: “In addition, considering the application of the system for SIB, the ab-initio molecular dynamics simulations were used to detect the behaviors of sub 1 and sub 3 at different temperatures. As shown in Supplementary Fig. 7, the results show that, at 298 and 313 K, the sub 1 and sub 3 structures show no significant change with the increase of temperature.”.

Page 24: “Ab initio molecular dynamics (AIMD) simulation

The ab-initio molecular dynamics (AIMD) simulations were carried out via VASP⁵⁰,

and 300 eV cutoff energy and 10^{-4} eV energy convergence were used. Nose-Hoover thermostat⁵¹⁻⁵³ was employed in order to control the system at finite temperatures of 298 and 313 K. The time step was 2 fs and each simulation lasted for 20 ps.”.

Supplementary Fig. 7 The models of **a** sub 1 after 20 ps **b** at 298 K and **c** at 313 K. The models of **d** sub 3 after 20 ps **e** at 298 K and **f** at 313 K.

Page 29:

50. Kresse, G. & Furthmüller, J. Efficient iterative schemes for ab initio total-energy calculations using a plane-wave basis set. *Phys. Rev. B* **54**, 11169-11186 (1996).
51. Nosé, S. A unified formulation of the constant temperature molecular dynamics methods. *J. Chem. Phys.* **81**, 511 (1984).
52. Nosé, S. Constant temperature molecular dynamics methods. *Prog. Theor. Phys. Suppl.* **103**, 1-46 (1991).
53. Hoover, W. G. Canonical dynamics: equilibrium phase-space distributions *Phys. Rev. A* **31**, 1695 (1985).

6: In Fig 3, the authors may clearly mark the points which shows the presence of Te in sub 1 and sub 3. DFT methodology is used in this work but no reference of it. Original reference for DFT should be given.

Our response: We appreciate the reviewer’s suggestions. According to the reviewer’s suggestions, we have adjusted the Fig. 3a-3c in the revised manuscript. In addition, we have supplemented the original reference to the DFT method used in this work. The detailed revision is as follows:

Page 7: “In order to explain the formation mechanism of twin structure in ZnSe_{0.7}Te_{0.3}@C, we carried out the density functional theory (DFT)^{22,23} calculations to detect the underlying impetus”.

Page 26:

22. Hohenberg, P. & Kohn, W. Inhomogeneous electron gas. *Phys Rev.* **136**, B864-B871 (1964).
23. Kohn, W. & Sham, L. J. Self-consistent equations including exchange and

correlation effects. *Phys Rev.* **140**, A1133-A1138 (1965).

Fig. 3 | Calculations of defect formation energy and average energy per atom in the final state. a A phase model in which one Te atom replaces one Se atom (sub 1). **b** A phase model in which three Te atoms replace three Se atoms (sub 3). **c** A phase model with one Te atom occupying interstitial position (int). Interface models of ZnSe_{0.7}Te_{0.3} (**d**) with TBs and (**e**) without TBs. **f** Comparison diagram of defect formation energy of three models of sub 1, sub 3 and int. **g** Comparison diagram of average energy per atom in the final state of interface models.

7: The authors should provide the original references of DFT, software and the functional used in the calculation.

Our response: We appreciate the reviewer's suggestions. According to the reviewer's suggestions, we have provided the original references of DFT, software and the functional used in the calculation in the revised manuscript. The detailed revision is as

follows:

Page 23: "All calculations were performed on the Vienna ab initio simulation package (VASP)⁴⁴ within the frame of density functional theory (DFT)^{22,23} with a cutoff energy of 450 eV. The exchange correlation interaction of electrons was described by the generalized gradient approximation (GGA) of Perdew–Burke–Ernzerhof (PBE) functional⁴⁵, and the interaction between electrons and ions was described by the projected augmented wave (PAW) method⁴⁶. The Brillouin zone was sampled by a Monkhorst-Pack k-point mesh⁴⁷ of $2 \times 2 \times 2$ grid for bulk phase models and $2 \times 3 \times 1$ grid for interface models. In addition, DFT–D3 method^{48,49} was used to explain the presence of van der Waals force within the system. The structural optimizations were done and the total energy converged within 10^{-5} eV. The final force of each ions was below 0.02 eV/Å."

Page 26:

22. Hohenberg, P. & Kohn, W. Inhomogeneous electron gas. *Phys Rev.* **136**, B864-B871 (1964).

23. Kohn, W. & Sham, L. J. Self-consistent equations including exchange and correlation effects. *Phys Rev.* **140**, A1133-A1138 (1965).

Page 29:

44. Kresse, G. & Furthmüller, J. Efficient iterative schemes for ab initio total-energy calculations using a plane-wave basis set. *Phys. Rev. B* **54**, 11169-11186 (1996).

45. Perdew, J. P., Burke, K. & Ernzerhof, M. Generalized gradient approximation made simple. *Phys. Rev. Lett.* **77**, 3865–3868 (1996).

46. Kresse, G. & Joubert, D. From ultrasoft pseudopotentials to the projector augmented-wave method. *Phys. Rev. B* **59**, 1758 (1999).

47. Monkhorst, H. J. & Pack, J. D. Special points for Brillouin-zone integrations. *Phys. Rev. B* **13**, 5188–5192 (1976).

48. Grimme, S., Antony, J., Ehrlich, S. & Krieg, H. A consistent and accurate ab initio parametrization of density functional dispersion correction (DFT-D) for the 94 elements H-Pu. *J. Chem. Phys.* **132**, 154104 (2010).

49. Grimme, S., Ehrlich, S. & Goerigk, L. Effect of the damping function in dispersion corrected density functional theory. *J. Comput. Chem.* **32**, 1456–1465 (2011).

8: "The authors may also explain the reason behind the choice of Monkhorst-Pack k-point mesh of $2 \times 2 \times 2$ grid for bulk phase model and $2 \times 3 \times 1$ grid for interface model in the study". This is not addressed. I failed to understand why along Y direction, 3 points are used for bulk but 2 points for interface. Why? Have you done k point convergence test?

Our response: Thank you very much for your comments. In order to choose the

reasonable Monkhorst-Pack k-point mesh, we had carried out the K-point convergence tests on the bulk phase and interface models, respectively, as shown in the following Figure R2. In general, when the energy change of each atom is less than 0.02 eV, the computing needs can be met. According to our results, so Monkhorst-Pack k-point mesh of $2 \times 2 \times 2$ grid for bulk phase model and $2 \times 3 \times 1$ grid for interface model are adopted.

Figure R2 a The bulk phase model of one Te atom replacing one Se atom in ZnSe and **b** K-point convergence test results. **c** The interface model of $\text{ZnSe}_{0.7}\text{Te}_{0.3}$ with one TB and **d** K-point convergence test results.

Thanks for your comments again. If it could be considered for publication in *Nat. Commun.*, it would be greatly appreciated!

Reviewer #2:

General comments: The authors have adequately responded to each of the reviewer comments and put together a complete study on the effects of Tellurium doping into ZnSe as an anode material for Sodium-ion batteries. The only remaining comment the reviewer has is for the authors to alter the title to be more impactful-something along the lines of explaining the study was performed with the goal of tailoring the properties of the material for performance as a sodium-ion battery anode material. Once this task has been completed, this work is suitable for publication in Nature Communications.

Our response: Thank you very much for your comments and kind suggestions for further improving our manuscript. We have made changes to the title with “Engineering twin structures and substitutional dopants in ZnSe_{0.7}Te_{0.3} anode material for enhanced sodium storage performance” in the revised manuscript. Thanks a lot.

Thanks for your comments again. If it could be considered for publication in **Nat. Commun.**, it would be greatly appreciated!